# On the (linear) convergence of Generalized Newton Inexact ADMM

**Zachary Frangella**                                             *zfrangella@alumni.stanford.edu*
*Department of Management Science and Engineering*
*Stanford University*

**Theo Diamandis**                                                         *tdiamand@mit.edu*
*Department of Electrical Engineering and Computer Science*
*Massachusetts Institute of Technology*

**Bartolomeo Stellato**                                               *bstellato@princeton.edu*
*Department of Operations Research and Financial Engineering*
*Princeton University*

**Madeleine Udell**                                                        *udell@stanford.edu*
*Department of Management Science and Engineering*
*Stanford University*

**Reviewed on OpenReview:** *https://openreview.net/forum?id=GT3naIXBxK*

## Abstract

This paper presents GeNI-ADMM, a framework for large-scale composite convex optimization that facilitates theoretical analysis of both existing and new approximate ADMM schemes. GeNI-ADMM encompasses any ADMM algorithm that solves a first- or second-order approximation to the ADMM subproblem inexactly. GeNI-ADMM exhibits the usual $\mathcal{O}(1/t)$-convergence rate under standard hypotheses and converges linearly under additional hypotheses such as strong convexity. Further, the GeNI-ADMM framework provides explicit convergence rates for ADMM variants accelerated with randomized linear algebra, such as NysADMM and sketch-and-solve ADMM, resolving an important open question on the convergence of these methods. This analysis quantifies the benefit of improved approximations and can aid in the design of new ADMM variants with faster convergence.

## 1 Introduction

The Alternating Direction Method of Multipliers (ADMM) is one of the most popular methods for solving composite optimization problems, as it provides a general template for a wide swath of problems and converges to an acceptable solution within a moderate number of iterations (Boyd et al., 2011). Indeed, Boyd et al. (2011) implicitly promulgates the vision that ADMM provides a unified solver for various convex machine learning problems. Unfortunately, for the large-scale problem instances routinely encountered in the era of Big Data, ADMM scales poorly and cannot provide a unified machine learning solver for problems of all scales. The scaling issue arises as ADMM requires solving two subproblems at each iteration, whose cost can increase superlinearly with the problem size. As a concrete example, in the case of $\ell_1$-logistic regression with an $n \times d$ data matrix, ADMM requires solving an $\ell_2$-regularized logistic regression problem at each iteration (Boyd et al., 2011). With a fast-gradient method, the total complexity of solving the subproblem is $\tilde{\mathcal{O}}(nd\sqrt{\kappa})$ (Bubeck, 2015), where $\kappa$ is the condition number of the problem. When $n$ and $d$ are in the tens of thousands or larger—a moderate problem size by contemporary machine learning standards—and $\kappa$ is large, such a high per-iteration cost becomes unacceptable. Worse, ill-conditioning is ubiquitous in machine

learning problems; often $\kappa = \Omega(n)$, in which case the cost of the subproblem solve becomes superlinear in the problem size.

Randomized numerical linear algebra (RandNLA) offers promising tools to scale ADMM to larger problem sizes. Recently Zhao et al. (2022) proposed the algorithm NysADMM, which uses a randomized fast linear system solver to scale ADMM up to problems with tens of thousands of samples and hundreds of thousands of features. The results in Zhao et al. (2022) show that ADMM combined with the RandNLA primitive runs *3 to 15× faster* than state-of-the-art solvers on machine learning problems from LASSO to SVM to logistic regression. Unfortunately, the convergence of randomized or approximate ADMM solvers like NysADMM is not well understood. NysADMM approximates the $x$-subproblem using a linearization based on a second-order Taylor expansion, which transforms the $x$-subproblem into a Newton-step, *i.e.*, a linear system solve. It then solves this system approximately (and quickly) using a randomized linear system solver. The convergence of this scheme, which combines linearization and inexactness, is not covered by prior theory for approximate ADMM; prior theory covers either linearization (Ouyang et al., 2015) or inexact solves (Eckstein & Bertsekas, 1992) but not both.

In this work, we bridge the gap between theory and practice to explain the excellent performance of a large class of approximate linearized ADMM schemes, including NysADMM (Zhao et al., 2022) and many methods not previously proposed. We introduce a framework called Generalized Newton Inexact ADMM, which we refer to as GeNI-ADMM (pronounced *genie-ADMM*). GeNI-ADMM includes NysADMM and many other approximate ADMM schemes as special cases. The name is inspired by viewing the linearized $x$-subproblem in GeNI-ADMM as a generalized Newton-step. GeNI-ADMM allows for inexactness in both the $x$-subproblem and the $z$-subproblem. We show GeNI-ADMM exhibits the usual $\mathcal{O}(1/t)$-convergence rate under standard assumptions, with linear convergence under additional assumptions. Our analysis also clarifies the value of using curvature in the generalized Newton step: approximate ADMM schemes that take advantage of curvature converge faster than those that do not at a rate that depends on the conditioning of the subproblems. As the GeNI-ADMM framework covers any approximate ADMM scheme that replaces the $x$-subproblem by a linear system solve, our convergence theory covers any ADMM scheme that uses fast linear system solvers. Given the recent flurry of activity on fast linear system solvers within the (randomized) numerical linear algebra community (Lacotte & Pilanci, 2020; Meier & Nakatsukasa, 2022; Frangella et al., 2023), our results will help realize these benefits for optimization problems as well. To demonstrate the power of the GeNI-ADMM framework, we establish convergence of NysADMM and another RandNLA-inspired scheme, sketch-and-solve ADMM, whose convergence was left as an open problem in Buluc et al. (2022).

## 1.1 Contributions

Our contributions may be summarized concisely as follows:

1. We provide a general ADMM framework GeNI-ADMM, that encompasses prior approximate ADMM schemes as well as new ones. It can take advantage of second-order information and allows for inexact subproblem solves.

2. Our analysis shows the benefits of schemes that employ preconditioning and variable metrics over methods that do not.

3. We show that despite all the approximations it makes, GeNI-ADMM converges ergodically at the usual ergodic $\mathcal{O}(1/t)$ rate.

4. We apply our framework to show some RandNLA-based approximate ADMM schemes converge at the same rate as vanilla ADMM, answering some open questions regarding their convergence (Buluc et al., 2022; Zhao et al., 2022). In the case of sketch-and-solve ADMM, we show modifications to the naive scheme are required to ensure convergence.

## 1.2 Roadmap

In Section 2, we formally state the optimization problem that we focus on in this paper and briefly introduce ADMM. In Section 3 we introduce the Generalized Newton Inexact ADMM framework and review ADMM and its variants. Section 4 gives various technical backgrounds and assumptions needed for our analysis. Section 5 establishes that GeNI-ADMM converges at an $\mathcal{O}(1/t)$ rate in the convex setting. In Section 7, we apply our theory to establish convergence rates for two methods that naturally fit into our framework, and we illustrate these results numerically in Section 8.

## 1.3 Notation and preliminaries

We call a matrix psd if it is positive semidefinite. We denote the convex cone of $n \times n$ real symmetric psd matrices by $\mathbb{S}_n^+$. Similarly we use $\mathbb{S}_n^{++}$ to denote the convex of symmetric postive definite matrices. We denote the Loewner ordering on $\mathbb{S}_n^+$ and $\mathbb{S}_n^{++}$ by $\preceq$, that is $A \preceq B$ if and only if $B - A$ is psd. Given a matrix $H$, we denote its spectral norm by $\|H\|$. If $f$ is a smooth function we denote its smoothness constant by $L_f$. We say a positive sequence $\{\varepsilon^k\}_{k \geq 1}$ is summable if $\sum_{k=1}^{\infty} \varepsilon^k < \infty$.

## 2 Problem statement and ADMM

Let $\mathcal{X}, \mathcal{Z}, \mathcal{H}$ be finite-dimensional inner-product spaces with inner-product $\langle \cdot, \cdot \rangle$ and norm $\| \cdot \|$. We wish to solve the convex constrained optimization problem

$$
\begin{aligned}
\text{minimize} \quad & f(x) + g(z) \\
\text{subject to} \quad & Mx + Nz = 0 \\
& x \in X, \ z \in Z,
\end{aligned}
\tag{1}
$$

with variables $x$ and $z$, where $X \subseteq \mathcal{X}$ and $Z \subseteq \mathcal{Z}$ are closed convex sets, $f$ is a smooth convex function, $g$ is a convex proper lower-semicontinuous (lsc) function, and $M : \mathcal{X} \mapsto \mathcal{H}$ and $N : \mathcal{Z} \mapsto \mathcal{H}$ are bounded linear operators.

**Remark 1.** *Often, the constraint is presented as $Mx + Nz = c$ for some non-zero vector $c$ however, by increasing the dimension of $\mathcal{Z}$ by 1 and replacing $N$ by $[N \ c]$, we can make $c = 0$. Thus, our setting of $c = 0$ is without loss of generality.*

We can write problem (1) as the saddle point problem

$$
\underset{x \in X, z \in Z}{\text{minimize}} \ \underset{y \in Y}{\text{maximize}} \ f(x) + g(z) + \langle y, Mx + Nz \rangle,
\tag{2}
$$

where $Y \subseteq \mathcal{H}$ is a closed convex set. The saddle-point formulation will play an important role in our analysis. Perform the change of variables $u = y/\rho$ and define the Lagrangian

$$
L_\rho(x, z, u) := f(x) + g(z) + \langle \rho u, Mx + Nz \rangle.
$$

Then (2) may be written concisely as

$$
\underset{x \in X, z \in Z}{\text{minimize}} \ \underset{u \in U}{\text{maximize}} \ L_\rho(x, z, u).
\tag{3}
$$

The ADMM algorithm (Algorithm 1) is a popular method for solving (1). Our presentation uses the scaled form of ADMM (Boyd et al., 2011; Ryu & Yin, 2022), which uses the change of variables $u = y/\rho$, this simplifies the algorithms and analysis, so we maintain this convention throughout the paper.

---

**Algorithm 1** ADMM

---

**input:** convex proper lsc functions $f$ and $g$, constraint matrix $M$, stepsize $\rho$

  **repeat**

      $x^{k+1} = \underset{x \in X}{\mathrm{argmin}}\{f(x) + \frac{\rho}{2}\|Mx + Nz^k + u^k\|^2\}$

      $z^{k+1} = \underset{z \in Z}{\mathrm{argmin}}\{g(z) + \frac{\rho}{2}\|Mx^{k+1} + Nz + u^k\|^2\}$

      $u^{k+1} = u^k + Mx^{k+1} + Nz^{k+1}$

  **until** convergence

**output:** solution $(x^\star, z^\star)$ of problem (1)

---

## 3   Generalized Newton Inexact ADMM

As shown in Algorithm 1, at each iteration of ADMM, two subproblems are solved sequentially to update variables $x$ and $z$. ADMM is often the method of choice when the $z$-subproblem has a closed-form solution. For example, if $g(x) = \|x\|_1$, the $z$-subproblem is soft thresholding, and if $g(x) = 1_{\mathcal{S}}(x)$ is the indicator function of a convex set $\mathcal{S}$, the $z$-subproblem is projection onto $\mathcal{S}$ (Parikh & Boyd, 2014, §6). However, it may be expensive to compute even with a closed-form solution. For example, when $g(x) = 1_{\mathcal{S}}(x)$ and $\mathcal{S}$ is the psd cone, the $z$-subproblem requires an eigendecomposition to compute the projection, which is prohibitively expensive for large problems (Rontsis et al., 2022).

Let us consider the $x$-subproblem

$$x^{k+1} = \underset{x \in X}{\mathrm{argmin}}\left\{f(x) + \frac{\rho}{2}\|Mx + Nz^k + u^k\|^2\right\}. \tag{4}$$

In contrast to the $z$-subproblem, there is usually no closed-form solution for the $x$-subproblem. Instead, an iterative scheme is often used to solve it inaccurately, especially for large-scale applications. This solve can be very expensive when the problem is large. To reduce computational effort, many authors have suggested replacing this problem with a simplified subproblem that is easier to solve. We highlight several strategies to do so below.

**Augmented Lagrangian linearization.** One strategy is to linearize the augmented Lagrangian term $\frac{\rho}{2}\|Mx + Nz^k + u^k\|^2$ in the ADMM subproblem and replace it by the quadratic penalty $\frac{1}{2}\|x - x^k\|_P^2$ for some (carefully chosen) positive definite matrix $P$. More formally, the strategy adds a quadratic term to form a new subproblem

$$x^{k+1} = \underset{x \in X}{\mathrm{argmin}}\left\{f(x) + \frac{\rho}{2}\|Mx + Nz^k + u^k\|^2 + \frac{1}{2}\|x - x^k\|_P^2\right\},$$

which is substantially easier to solve for an appropriate choice of $P$. One canonical choice is $P = \eta I - \rho M^T M$, where $\eta > 0$ is a constant. For this choice, the quadratic terms involving $M$ cancel, and we may omit constants with respect to $x$, resulting in the subproblem

$$x^{k+1} = \underset{x \in X}{\mathrm{argmin}}\left\{f(x) + \rho\langle Mx, Mx^k - z^k + u^k\rangle + \frac{\eta}{2}\|x - x^k\|^2\right\}.$$

Here, an isotropic quadratic penalty replaces the augmented Lagrangian term in (4). This strategy allows the subproblem solve to be replaced by a proximal operator with a (possibly) closed-form solution. This strategy has been variously called *preconditioned ADMM*, proximal ADMM, and (confusingly) linearized ADMM (Deng & Yin, 2016; He & Yuan, 2012; Ouyang et al., 2015).

**Function approximation.** The second strategy to simplify the $x$-subproblem, is to approximate the function $f$ by a first- or second-order approximation (Ouyang et al., 2015; Ryu & Yin, 2022; Zhao et al., 2022), forming the new subproblem

$$x^{k+1} = \underset{x \in X}{\mathrm{argmin}}\left\{f(x^k) + \langle \nabla f(x^k), x - x^k\rangle + \frac{1}{2}\|x - x^k\|_H^2 + \frac{\rho}{2}\|Mx + Nz^k + u^k\|^2\right\} \tag{5}$$

where $H$ is the Hessian of $f$ at $x^k$. The resulting subproblem is quadratic and may be solved by solving a linear system or (for $M = I$) performing a linear update, as detailed below in Section 3.1.1.

**Inexact subproblem solve.** The third strategy is to solve the ADMM subproblems inexactly to achieve some target accuracy, either in absolute error or relative error. An absolute-error criterion chooses the subproblem error a priori (Eckstein & Bertsekas, 1992), while a relative error criterion requires the subproblem error to decrease as the algorithm nears convergence, for example, by setting the error target at each iteration proportional to $\|u^{k+1} - u^k - \rho(z^{k+1} - z^k)\|$ (Eckstein & Yao, 2018).

**Approximations used by GeNI-ADMM.** The GeNI-ADMM framework allows for any combination of the three strategies: augmented Lagrangian linearization, function approximation, and inexact subproblem solve. Consider the generalized second-order approximation to $f$

$$f(x) \approx f_1(x) + f_2(\tilde{x}^k) + \langle \nabla f_2(\tilde{x}^k), x - \tilde{x}^k \rangle + \frac{1}{2}\|x - \tilde{x}^k\|_{\Theta^k}^2, \tag{6}$$

where $f_1 + f_2 = f$ and $\{\Theta^k\}_{k \geq 1}$ is a sequence of psd matrices that approximate the Hessian of $f_2$. GeNI-ADMM uses this approximation in the $x$-subproblem, resulting in the new subproblem

$$\begin{aligned}
\tilde{x}^{k+1} = \underset{x \in X}{\operatorname{argmin}}\{&f_1(x) + \langle \nabla f_2(\tilde{x}^k), x - \tilde{x}^k \rangle + \frac{1}{2}\|x - \tilde{x}^k\|_{\Theta^k}^2 \\
&+ \frac{\rho}{2}\|Mx + N\tilde{z}^k + \tilde{u}^k\|^2\}.
\end{aligned} \tag{7}$$

We refer to (7) as a generalized Newton step. The intuition for the name is made plain when $X = \mathbb{R}^d$, $f_1 = 0, f_2 = f, N = -I$, in which case the update becomes

$$\left(\Theta^k + \rho M^T M\right)\tilde{x}^{k+1} = \Theta^k \tilde{x}^k - \nabla f(\tilde{x}^k) - \rho M^T(\tilde{u}^k - \tilde{z}^k). \tag{8}$$

Equation (8) shows that the $x$-subproblem reduces to a linear system solve, just like the Newton update. We can also interpret GeNI-ADMM as a linearized proximal augmented Lagrangian (P-ALM) method (Hermans et al., 2019; 2022). From this point of view, GeNI-ADMM replaces $f$ in the P-ALM step by its linearization and adds a specialized penalty defined by the $\Theta^k$-norm.

GeNI-ADMM also replaces the $z$-subproblem of ADMM with the following problem

$$\tilde{z}^{k+1} = \underset{z \in Z}{\operatorname{argmin}}\{g(z) + \frac{\rho}{2}\|M\tilde{x}^{k+1} + Nz + \tilde{u}^k\|^2 + \frac{1}{2}\|z - \tilde{z}^k\|_{\Psi^k}^2\}. \tag{9}$$

Incorporating the quadratic $\Psi^k$ term allows us to linearize the augmented Lagrangian term, which is useful when $N$ is very complicated.

However, even with the allowed approximations, it is unreasonable to assume that (7), (9) are solved exactly at each iteration. Indeed, the hallmark of the NysADMM scheme from Zhao et al. (2022) is that it solves (7) inexactly (but efficiently) using a randomized linear system solver. Thus, the GeNI-ADMM framework allows for inexactness in the $x$ and $z$-subproblems, as seen in Algorithm 2.

---

**Algorithm 2** Generalized Newton Inexact ADMM (GeNI-ADMM)

---

**input:** penalty parameter $\rho$, sequence of psd matrices $\{\Theta^k\}_{k \geq 1}, \{\Psi^k\}_{k \geq 1}$, forcing sequences $\{\varepsilon_x^k\}_{k \geq 1}, \{\varepsilon_z^k\}_{k \geq 1}$,
  **repeat**

$$\tilde{x}^{k+1} \overset{\varepsilon_x^k}{\approx} \underset{x \in X}{\operatorname{argmin}}\{f_1(x) + \langle \nabla f_2(\tilde{x}^k), x - \tilde{x}^k \rangle + \frac{1}{2}\|x - \tilde{x}^k\|_{\Theta^k}^2 + \frac{\rho}{2}\|Mx + N\tilde{z}^k + \tilde{u}^k\|^2\}$$

$$\tilde{z}^{k+1} \overset{\varepsilon_z^k}{\approx} \underset{z \in Z}{\operatorname{argmin}}\{g(z) + \frac{\rho}{2}\|M\tilde{x}^{k+1} + Nz + \tilde{u}^k\|^2 + \frac{1}{2}\|z - \tilde{z}^k\|_{\Psi^k}^2\}$$

$$\tilde{u}^{k+1} = \tilde{u}^k + M\tilde{x}^{k+1} + N\tilde{z}^{k+1}$$

  **until** convergence
**output:** solution $(x^\star, z^\star)$ of problem (1)

---

Algorithm 2 differs from ADMM (Algorithm 1) in that a) the $x$ and $z$-subproblems are now given by (7) and (9) and b) both subproblems may be solved inexactly. Given the inexactness and the use of the generalized Newton step in place of the original $x$-subproblem, we refer to Algorithm 2 as Generalized Newton Inexact ADMM (GeNI-ADMM). The inexactness schedule is controlled by the *forcing sequences* $\{\varepsilon_x^k\}_{k \geq 1}, \{\varepsilon_z^k\}_{k \geq 1}$ that specify how accurately the $x$ and $z$-subproblems are solved at each iteration. These subproblems have different structures that require different notions of accuracy. To distinguish between them, we make the following definition.

**Definition 1** ($\varepsilon$-minimizer and $\varepsilon$-minimum). *Let $h : \mathcal{T} \mapsto \mathbb{R}$ be strongly-convex and let $t^\star = \operatorname{argmin}_{t' \in \mathcal{T}} h(t')$.*

- *($\varepsilon$-minimizer) Given $t \in \mathcal{T}$, we say $t$ is an $\varepsilon$-minimizer of $minimize_{t \in \mathcal{T}} h(t)$ and write*

$$t \overset{\varepsilon}{\approx} \operatorname*{argmin}_{t' \in \mathcal{T}} h(t') \quad \text{if and only if} \quad \|t - t^\star\| \leq \varepsilon.$$

  *In words, $t$ is nearly equal to the argmin of $h(t)$ in set $\mathcal{T}$.*

- *($\varepsilon$-minimum) Given $t \in \mathcal{T}$, we say $t$ gives an $\varepsilon$-minimum of $minimize_{t \in \mathcal{T}} h(t)$ and write*

$$t \overset{\varepsilon}{\cong} \operatorname*{argmin}_{t' \in \mathcal{T}} h(t') \quad \text{if and only if} \quad h(t) - h(t^\star) \leq \varepsilon.$$

  *In words, $t$ produces nearly the same objective value as the argmin of $h(t)$ in set $\mathcal{T}$.*

Thus, from Definition 1 and Algorithm 2, we see for each iteration $k$ that $\tilde{x}^{k+1}$ is an $\varepsilon_x^k$-minimizer of the $x$-subproblem, while $\tilde{z}^{k+1}$ gives an $\varepsilon_z^k$-minimum of the $z$-subproblem.

## 3.1 Related work

The literature on the convergence of ADMM and its variants is vast, so we focus on prior work most relevant to our setting. Table 1 lists some of the prior work that developed and analyzed the approximation strategies described in Section 3. GeNI-ADMM differs from all prior work in Table 1 by allowing (almost) all these approximations and more. It also provides explicit rates of convergence to support choices between algorithms. Moreover, many of these algorithms can be recovered from GeNI-ADMM.

### 3.1.1 Algorithms recovered from GeNI-ADMM

Various ADMM schemes in the literature can be recovered by appropriately selecting the parameters in Algorithm 2. Let us consider a few special cases to highlight important prior work on approximate ADMM, and provide concrete intuition for the general framework provided by Algorithm 2.

**NysADMM** The NysADMM scheme (Zhao et al., 2022) assumes the problem is unconstrained $M = I, N = -I$. Diamandis et al. (2023) have extended NysADMM to the constrained setting with the GeNIOS solver, which we will discuss more in Section 7.1. For simplicity of exposition, we focus on the original NysADMM for unconstrained problems. GeNI-ADMM specializes to NysADMM taking $\Theta^k = \eta(H_f^k + \sigma I)$, and $\Psi^k = 0$, where $H_f^k$ denotes the Hessian of $f$ at the $k$th iteration. Unlike the original NysADMM scheme of Zhao et al. (2022), we include the regularization term $\sigma I$, where $\sigma > 0$. This inclusion is required for theoretical analysis but seems unnecessary in practice (see Section 7.1 for a detailed discussion). Substituting this information into (8) leads to the following update for the $x$-subproblem.

$$\tilde{x}^{k+1} = \tilde{x}^k - \left(\eta H_f^k + (\rho + \eta\sigma)I\right)^{-1} \left(\nabla f(\tilde{x}^k) + \rho(\tilde{x}^k - \tilde{z}^k + \tilde{u}^k)\right). \tag{10}$$

**Sketch-and-solve ADMM** If $M = I, N = -I$, $\Psi^k = 0$, and $\Theta^k$ is chosen to be a matrix such that $\Theta^k + \rho I$ is easy to factor, we call the resulting scheme *sketch-and-solve ADMM*. The name *sketch-and-solve ADMM* is motivated by the fact that such a $\Theta^k$ can often be obtained via sketching techniques. However, the method works for any $\Theta^k$, not only ones constructed by sketching methods. The update is given by

$$\tilde{x}^{k+1} = \tilde{x}^k - \left(\Theta^k + \rho I\right)^{-1} \left(\nabla f(\tilde{x}^k) + \rho(\tilde{x}^k - \tilde{z}^k + \tilde{u}^k)\right).$$

| Reference | Convergence rate | Problem class | Augmented Lagrangian Linearization | Function approximation | Subproblem inexactness |
|---|---|---|---|---|---|
| Eckstein & Bertsekas (1992) | ✗ | Convex | ✗ | ✗ | $x, z$ absolute error |
| He & Yuan (2012) | Ergodic $\mathcal{O}(1/t)$ | Convex | $x$ | ✗ | ✗ |
| Monteiro & Svaiter (2013) | Ergodic $\mathcal{O}(1/t)$ | Convex | ✗ | ✗ | ✗ |
| Ouyang et al. (2013) | $\mathcal{O}(1/\sqrt{t})$ | Stochastic convex | ✗ | $f$ stochastic first-order | ✗ |
| Ouyang et al. (2015) | Ergodic $\mathcal{O}(1/t)$ | Convex | $x$ | $f$ first-order | ✗ |
| Deng & Yin (2016) | Linear | Strongly convex | $x, z$ | ✗ | ✗ |
| Eckstein & Yao (2018) | ✗ | Convex | ✗ | ✗ | $x$ relative error |
| Hager & Zhang (2020) | Ergodic $\mathcal{O}(1/t)$, $\mathcal{O}(1/t^2)$ | Convex, Strongly convex | $x$ | ✗ | $x$ relative error |
| Yuan et al. (2020) | Locally linear | Convex | $x, z$ | ✗ | ✗ |
| Zhao et al. (2022) | ✗ Convergence for quadratic $f$ only | Convex | ✗ | $x$ | $x$ absolute error |
| Ryu & Yin (2022) | ✗ | Convex | $x, z$ | $f, g$ partial first-order | ✗ |
| **This work** | Ergodic $\mathcal{O}(1/t)$, Linear | Convex, Strongly convex | $x, z$ | $f$ partial generalized second-order | $x, z$ absolute error |

Table 1: A structured comparison of related work on the convergence of ADMM and its variants. The "$x$" ("$z$") in the table denotes that a paper uses the corresponding strategy of the column to simplify the $x$-subproblem ($z$-subproblem). In the "**Function approximation**" column, the "$f$" ("$g$") indicates that a paper approximates function "$f$" ("$g$") in the $x$-subproblem ($z$-subproblem). "stochastic first-order" means a paper uses first-order function approximation but replace the gradient term with a stochastic gradient. "partial generalized second-order" means a paper uses second-order function approximation as (7), but $H$ is not necessarily the Hessian.

We provide the full details of sketch-and-solve ADMM in Section 7.2. To our knowledge, sketch-and-solve ADMM has not been previously proposed or analyzed.

**Proximal/Generalized ADMM**  Set $f_1 = f, f_2 = 0$, and fix $\Theta^k = \Theta$ and $\Psi^k = \Psi$, where $\Theta$ and $\Psi$ are symmetric positive definite matrices. Then GeNI-ADMM simplifies to

$$\tilde{x}^{k+1} = \operatorname*{argmin}_{x \in X} \left\{ f(x) + \frac{\rho}{2} \| Mx + N\tilde{z}^k + \tilde{u}^k \|^2 + \frac{1}{2} \| x - \tilde{x}^k \|_\Theta^2 \right\},$$

$$\tilde{z}^{k+1} = \operatorname*{argmin}_{z \in Z} \left\{ g(z) + \frac{\rho}{2} \| M\tilde{x}^{k+1} + Nz + \tilde{u}^k \|^2 + \frac{1}{2} \| z - \tilde{z}^k \|_\Psi^2 \right\},$$

which is the Proximal/Generalized ADMM of Deng & Yin (2016).

**Linearized ADMM**  Set $\Theta^k = \alpha^{-1} I - \rho M^T M$ and $\Psi = \beta^{-1} I - \rho N^T N$, for all $k \geq 1$. Then GeNI-ADMM simplifies to

$$\tilde{x}^{k+1} = \operatorname{prox}_{\alpha f} \left( \tilde{x}^k - \alpha \rho M^T (M\tilde{x}^k + N\tilde{z}^k + \tilde{u}^k) \right),$$

$$\tilde{z}^{k+1} = \operatorname{prox}_{\beta g} \left( \tilde{z}^k - \beta \rho N^T (M\tilde{x}^{k+1} + N\tilde{z}^k + \tilde{u}^k) \right),$$

which is exactly Linearized ADMM (Parikh & Boyd, 2014, §4.4.2).

**Primal Dual Hybrid Gradient**  The Primal Dual Hybrid Gradient (PDHG) or Chambolle-Pock algorithm of Chambolle & Pock (2011) is a special case of GeNI-ADMM since PDHG is a special case of Linearized ADMM (see section 3.5 of Ryu & Yin (2022)).

**Gradient descent ADMM**  Consider two linearization schemes from Ouyang et al. (2015). The first scheme we call *gradient descent ADMM* (GD-ADMM) is useful when it is cheap to solve a least-squares problem with $M$. GD-ADMM is obtained from GeNI-ADMM by setting $f_1 = 0, f_2 = f$, and $\Theta^k = \eta I$ for all $k$. The $\tilde{x}^{k+1}$ update (8) for GD-ADMM simplifies to

$$\tilde{x}^{k+1} = \tilde{x}^k - (\rho M^T M + \eta I)^{-1} \left( \nabla f(\tilde{x}^k) + \rho M^T (M\tilde{x}^k + N\tilde{z}^k + \tilde{u}^k) \right). \tag{11}$$

The second scheme, *linearized-gradient descent ADMM* (LGD-ADMM), is useful when $M$ is not simple, so that the update (11) is no longer cheap. To make the $x$-subproblem update cheaper, it linearizes the augmented Lagrangian term by setting $\Theta^k = \alpha^{-1} I - \rho M^T M$ for all $k$ in addition to linearizing $f$. In this case, (8) yields the $\tilde{x}^{k+1}$ update

$$\tilde{x}^{k+1} = \tilde{x}^k - \alpha \left( \nabla f(\tilde{x}^k) + \rho M^T (M\tilde{x}^k + N\tilde{z}^k + \tilde{u}^k) \right). \tag{12}$$

Observe in the unconstrained case, when $M = I$, the updates (11) and (12) are equivalent and generate the same iterate sequences when initialized at the same point (Zhao et al., 2021). Indeed, they are both generated by performing a gradient step on the augmented Lagrangian (7), for suitable choices of the parameters. Notably, this terminology differs from Ouyang et al. (2015), who refer to (11) as "linearized ADMM" (L-ADMM) and (12) as "linearized preconditioned ADMM" (LP-ADMM). We choose our terminology to emphasize that GD-ADMM accesses $f$ via its gradient, as in the literature the term "linearized ADMM" is usually reserved for methods that access $f$ through its prox operator (He & Yuan, 2012; Parikh & Boyd, 2014; Deng & Yin, 2016).

In the remainder of this paper, we will prove convergence of Algorithm 2 under appropriate hypotheses on the sequences $\{\Theta^k\}_{k \geq 1}, \{\Psi^k\}_{k \geq 1}, \{\varepsilon_x^k\}_{k \geq 1}$, and $\{\varepsilon_z^k\}_{k \geq 1}$.

## 4  Technical preliminaries and assumptions

We introduce some important concepts that will be central to our analysis. The first is $\Theta$-*relative smoothness*, which is crucial to establish that GeNI-ADMM benefits from curvature information provided by the Hessian.

**Definition 2** (Θ-relative smoothness)**.** *Let* $\Theta : \mathcal{D} \rightarrow \mathbb{S}^n_{++}$ *be bounded. We say* $f : \mathcal{D} \rightarrow \mathbb{R}$ *is* Θ*-relatively smooth if there exists* $\hat{L}_\Theta > 0$ *such that for all* $x, y \in \mathcal{D}$

$$f(x) \leq f(y) + \langle \nabla f(y), x - y \rangle + \frac{\hat{L}_\Theta}{2} \|x - y\|^2_{\Theta(y)}. \tag{13}$$

That is, the function $f$ is smooth with respect to the Θ-norm. Definition 2 generalizes *relative smoothness*, introduced in Gower et al. (2019) to analyze Newton's method. The definition in Gower et al. (2019) takes $\Theta$ to be the Hessian of $f$, $H_f$. When $f$ belongs to the popular family of generalized linear models, then (13) holds with a value of $\hat{L}_{H_f}$ that is independent of the conditioning of the problem data (Gower et al., 2019). For instance, if $f$ is quadratic and $\Theta(y) = H_f$, then (13) holds with equality for $\hat{L}_{H_f} = 1$. Conversely, if we take $\Theta = I$, which corresponds to GD-ADMM, then $\hat{L}_\Theta = L_f$, the smoothness constant of $f$. Our theory uses the fact that $\hat{L}_\Theta$ is much smaller than the smoothness constant $L_f$ for methods that take advantage of curvature, and relies on $\hat{L}_\Theta$ to characterize the faster convergence speed of these methods.

The other important idea we need is the notion of an $\varepsilon$-*subgradient* (Bertsekas et al., 2003; Hiriart-Urruty & Lemaréchal, 1993).

**Definition 3** ($\varepsilon$-subgradient)**.** *Let* $r : \mathcal{D} \rightarrow \mathbb{R}$ *be a convex function and* $\varepsilon > 0$*. We say that* $s \in \mathcal{D}^*$ *is an* $\varepsilon$-*subgradient for* $r$ *at* $z \in \mathcal{D}$ *if, for every* $z' \in \mathcal{D}$*, we have*

$$r(z') - r(z) \geq \langle s, z' - z \rangle - \varepsilon.$$

*We denote the set of* $\varepsilon$-*subgradients for* $r$ *at* $z$ *by* $\partial_\varepsilon r(z)$*.*

Clearly, any subgradient is an $\varepsilon$-subgradient, so Definition 3 provides a natural weakening of a subgradient. The $\varepsilon$-subgradient is critical for analyzing $z$-subproblem inexactness, and our usage in this context is inspired by the convergence analysis of inexact proximal gradient methods (Schmidt et al., 2011). We shall need the following proposition whose proof may be found in (Bertsekas et al., 2003, Proposition 4.3.1).

**Proposition 1.** *Let* $r$*,* $r_1$*, and* $r_2$ *be convex functions. Then for any* $z$*, the following holds:*

1. $0 \in \partial_\varepsilon r(z)$ *if and only if* $r(z) \overset{\varepsilon}{\cong} \text{argmin}_{z'} r(z')$*, that is* $z$ *gives an* $\varepsilon$-*minimum of* $\text{minimize}_{z'} r(z')$*.*

2. $\partial_\varepsilon(r_1 + r_2)(z) \subset \partial_\varepsilon r_1(z) + \partial_\varepsilon r_2(z)$*.*

With Proposition 1 recorded, we prove the following lemma in Appendix B.1, which will play a critical role in establishing the convergence of GeNI-ADMM.

**Lemma 1.** *Let* $\tilde{z}^{k+1}$ *give an* $\varepsilon^k_z$-*minimum of the* $z$-*subproblem. Then there exists an* $\tilde{s}$ *with* $\|\tilde{s}\| \leq C\sqrt{\varepsilon^k_z}$ *such that*

$$-\rho N^T \tilde{u}^{k+1} + \Psi_k(\tilde{z}^k - \tilde{z}^{k+1}) + \tilde{s} \in \partial_{\varepsilon^k_z} g(\tilde{z}^{k+1}).$$

### 4.1 Assumptions

In this section, we present the main assumptions required by our analysis.

**Assumption 1** (Existence of saddle point)**.** *There exists an optimal primal solution* $(x^\star, z^\star) \in X \times Z$ *for* (1) *and an optimal dual solution* $u^\star \in U$ *such that* $(x^\star, z^\star, u^\star)$ *is a saddle point of* (2)*. Here,* $U \subset \mathcal{Z}$ *is a closed convex set and* $\rho U = Y$*. We denote the optimal objective value of* (1) *as* $p^\star$*.*

Assumption 1 is standard and merely assumes that (1) has a solution.

**Assumption 2** (Regularity of $f$ and $g$)**.** *The function* $f$ *is twice-continuously differentiable and is* 1-*relatively smooth with respect to* Θ*. The function* $g$ *is finite-valued, convex, and lower semi-continuous.*

Assumption 2 is also standard. It ensures that it makes sense to talk about the Hessian of $f$ and that $f$ is relatively smooth. Note assuming $\hat{L}_\Theta = 1$ is without loss of generality, for we can always redefine $\Theta' = \hat{L}_\Theta \Theta$, and $f$ will be 1-relatively smooth with respect to $\Theta'$.

**Assumption 3** (Forcing sequence summability and approximate subproblem oracles)**.** *Let* $\{\varepsilon_x^k\}_{k\geq 1}$ *and* $\{\varepsilon_z^k\}_{k\geq 1}$ *be given forcing sequences, we assume they satisfy*

$$\mathcal{E}_x = \sum_{k=1}^{\infty} \varepsilon_x^k < \infty, \quad \mathcal{E}_z = \sum_{k=1}^{\infty} \sqrt{\varepsilon_z^k} < \infty.$$

*Further we define the constants* $K_{\varepsilon_x} := \sup_{k\geq 1} \varepsilon_x^k$, $K_{\varepsilon_z} := \sup_{k\geq 1} \sqrt{\varepsilon_z^k}$. *Observe* $K_{\varepsilon_x}$ *and* $K_{\varepsilon_z}$ *are finite owing to the summability hypotheses. Moreover, we assume Algorithm 2 is equipped with oracles for solving the* $x$ *and* $z$-*subproblems, which at each iteration produce approximate solutions* $\tilde{x}^{k+1}$, $\tilde{z}^{k+1}$ *satisfying:*

$$\tilde{x}^{k+1} \overset{\varepsilon_x^k}{\approx} \underset{x\in X}{\operatorname{argmin}}\{f_1(x) + \langle \nabla f_2(\tilde{x}^k), x - \tilde{x}^k \rangle + \frac{1}{2}\|x - \tilde{x}^k\|_{\Theta^k}^2 + \frac{\rho}{2}\|Mx + N\tilde{z}^k + \tilde{u}^k\|^2\},$$

$$\tilde{z}^{k+1} \overset{\varepsilon_z^k}{\approx} \underset{z\in Z}{\operatorname{argmin}}\{g(z) + \frac{\rho}{2}\|M\tilde{x}^{k+1} + Nz + \tilde{u}^k\|^2 + \frac{1}{2}\|z - \tilde{z}^k\|_{\Psi^k}^2\}.$$

The conditions on the $x$ and $z$ subproblem oracles are consistent with those of Eckstein & Bertsekas (1992), which requires the sum of the errors in the subproblems to be summable. The approximate solution criteria of Assumption 3 are also easily met in practical applications, as the subproblems are either simple enough to solve exactly, or can be efficiently solved approximately via iterative algorithms. For instance, with LGD-ADMM the $x$-subproblem is simple to solve enough exactly, while for NysADMM the $x$-subproblem is efficiently solved via conjugate gradient with a randomized preconditioner with the standard stopping criteria. In general, the $x$ and $z$ subproblems are always strongly convex optimization problems, so classic criterion such as the norm of the gradient (subgradient) can be used to monitor convergence (Nesterov, 2018). Often the structure of the subproblem naturally suggests a certain algorithm, e.g. a Krylov method such as LOBPCG for projecting onto the psd cone (Rontsis et al., 2022), in which case the subproblem solver may come with a natural metric for determining when to terminate.

**Assumption 4** (Regularity of $\{\Theta^k\}_{k\geq 1}$ and $\{\Psi^k\}_{k\geq 1}$)**.** *We assume there exists constants* $\theta_{\max}$, $\theta_{\min}$, $\psi_{\max}$, *and* $\nu > 0$, *such that*

$$\theta_{\max}I \succeq \Theta^k \succeq \theta_{\min}I, \ \psi_{\max}I \succeq \Psi^k, \ \Psi^k + \rho N^T N \succeq \nu I, \quad \text{for all } k \geq 1. \tag{14}$$

*Moreover, we also assume that The sequences* $\{\Theta^k\}_{k\geq 1}, \{\Psi^k\}_{k\geq 1}$ *satisfy*

$$\|\tilde{x}^k - x^\star\|_{\Theta^k}^2 \leq (1 + \zeta^{k-1})\|\tilde{x}^k - x^\star\|_{\Theta^{k-1}}^2, \tag{15}$$
$$\|\tilde{z}^k - z^\star\|_{\Psi^k}^2 \leq (1 + \zeta^{k-1})\|\tilde{z}^k - z^\star\|_{\Psi^{k-1}}^2,$$

*where* $\{\zeta^k\}_{k\geq 1}$ *is a non-negative summable sequence, that is,* $\sum_{k=1}^{\infty} \zeta^k < \infty$.

The first half of Assumption 4 is standard, and essentially requires that $\Theta^k$ and $\Psi^k + \nu N^T N$ define norms. The strict positive-definiteness of the sequences $\{\Theta^k\}$ and $\{\Psi^k\}$ is crucial, as it ensures that the $x$ and $z$ subproblems in Algorithm 2 always have a solution. Indeed, classical ADMM, which is recovered from Algorithm 2 with $f_1 = f$ and $\Theta^k = \Psi^k = 0$ for all $k$, may not even be well-defined without further assumptions. Chen et al. (2017) showed that without additional assumptions on $f$, $g$, $M$, or $N$, there exists an objective for which the $x$ and $z$ subproblems fail to have solutions, so that ADMM cannot be executed. For most common choices of $\Theta^k$ and $\Psi^k$, the assumptions in (14) are satisfied; they can also be readily enforced by adding a small multiple of the identity. The addition of such regularization is common practice in popular ADMM solvers, such as OSQP (Stellato et al., 2020), as it avoids issues with degeneracy and increases numerical stability.

The second part of the assumption requires that $\Theta^k$ ($\Psi^k$) and $\Theta^{k-1}$ ($\Psi^{k-1}$) eventually not differ much on the distance of $\tilde{x}^k$ ($\tilde{z}^k$) to $x^\star$ ($z^\star$). Assumptions of this form are common in analyses of optimization algorithms that use variable metrics. For instance, He et al. (2002) which develops an alternating directions method for monotone variational inequalities, assumes their equivalents of the sequences $\{\Theta^k\}$ and $\{\Psi^k\}$ satisfy

$$(1 - \zeta^{k-1})\Theta^{k-1} \preceq \Theta^k \preceq (1 + \zeta^{k-1})\Theta^{k-1}, \ (1 - \zeta^{k-1})\Psi^{k-1} \preceq \Psi^k \preceq (1 + \zeta^{k-1})\Psi^{k-1}. \tag{16}$$

More recently, Rockafellar (2023) analyzed the proximal point method with variable metrics, under the assumption the variable metrics satisfy (16). Thus, Assumption 4 is consistent with the literature and is, in fact, weaker than prior work, as (16) implies (15). Assumption 4 may always be enforced by changing the $\Theta^k$'s and $\Psi^k$'s only finitely many times.

We also define the following constants which shall be useful in our analysis,

$$\tau_\zeta := \prod_{k \geq 2}(1 + \zeta^k), \quad \mathcal{E}_\zeta = \tau_\zeta \left(\sum_{k=1}^\infty \zeta^k\right) < \infty.$$

Note Assumption 4 implies $\tau_\zeta < \infty$. Moreover, for any $k \geq 1$ it holds that

$$\|\tilde{x}^k - x^\star\|_{\Theta^k}^2 \leq \tau_\zeta \|\tilde{x}^k - x^\star\|_{\Theta^1}^2, \quad \|\tilde{z}^k - z^\star\|_{\Psi^k}^2 \leq \tau_\zeta \|\tilde{z}^k - z^\star\|_{\Psi^1}^2.$$

## 5 Sublinear convergence of GeNI-ADMM

This section establishes our main result, Theorem 1, which shows that Algorithm 2 enjoys the same $\mathcal{O}(1/t)$ ergodic convergence rate as standard ADMM. Section 5.1 formally presents the theorem, Section 5.2 shows how better Hessian approximations enhance the convergence rate, Section 5.3 outlines our proof approach, Section 5.4 establishes key lemmas, and Section 5.5 completes the proof.

### 5.1 GeNI-ADMM: Ergodic sublinear convergence

Before presenting the convergence result, we must define several important constants that appear in the theorem statement. The first set of constants represent the distances of initial iterates of Algorithm 2 from their optimal values:

$$d_{x^\star,\Theta^1} := \|\tilde{x}^1 - x^\star\|_{\Theta^1}, \ d_{u^\star} := \|\tilde{u}^1 - u^\star\|, \ d_{z^\star,\Psi^1_{\rho,N}} := \|\tilde{z}^1 - z^\star\|_{\Psi^1 + \rho N^T N}.$$

The next constant we define aggregates the effects of the distance to the optimum, the subproblem errors, and the changes in the variable metric:

$$\Gamma := \frac{1}{2}d_{x^\star,\Theta^1}^2 + \frac{1}{2}d_{z^\star,\Psi^1_{\rho,N}}^2 + C_x\mathcal{E}_x + C_z\mathcal{E}_z + C_\zeta\mathcal{E}_\zeta,$$

where $C_x$ and $C_z$ are as in Lemma 3, Lemma 4, and $C_\zeta > 0$ is a constant no larger than $\tau_\zeta R\left(\lambda_1(\Theta^1) + \lambda_1(\Psi^1 + \rho N^T N)\right)$.

**Theorem 1** (Ergodic convergence)**.** *Instate Assumptions 1-2 and let $p^\star$ denote the optimum of* (1)*. Run Algorithm 2 with $\rho > 0$, $\{\varepsilon_x^k\}_{k \geq 1}, \{\varepsilon_x^k\}_{k \geq 1}$ satisfying Assumption 3, $\{\Theta^k\}_{k \geq 1}, \{\Psi^k\}_{k \geq 1}$ satisfying Assumption 4, and initialization $(\tilde{u}^1, \tilde{x}^1, \tilde{z}^1)$, where $\tilde{u}^1 = 0$ and $M\tilde{x}^1 = -N\tilde{z}^1$. Then, the suboptimality gap and feasibility gaps satisfy*

$$f(\bar{x}^{t+1}) + g(\bar{z}^{t+1}) - p^\star \leq \frac{\Gamma}{t}, \quad \|M\bar{x}^{t+1} + N\bar{z}^{t+1}\| \leq \frac{2}{t}\sqrt{\frac{\Gamma}{\rho} + d_{u^\star}^2},$$

*where*

$$\bar{x}^{t+1} = \frac{1}{t}\sum_{k=2}^{t+1}\tilde{x}^k, \quad \bar{z}^{t+1} = \frac{1}{t}\sum_{k=2}^{t+1}\tilde{z}^k.$$

*Consequently, after $\mathcal{O}(1/\epsilon)$ iterations,*

$$f(\bar{x}^{t+1}) + g(\bar{z}^{t+1}) - p^\star \leq \epsilon, \ and \ \|M\bar{x}^{t+1} + N\bar{z}^{t+1}\| \leq \epsilon.$$

The proof of Theorem 1 is given in Section 5.5.

Theorem 1 shows that with appropriate forcing sequences, the suboptimality gap and the feasibility residuals both go to zero at a rate of $\mathcal{O}(1/t)$. Hence, the overall convergence rate of ADMM is preserved despite all the approximations involved in GeNI-ADMM.

## 5.2 The benefits of using a better Hessian approximation

An important consequence of Theorem 1 is that not all approximate ADMM schemes yield the same performance. GeNI-ADMM schemes that use better approximations ot the Hessian can enjoy faster convergence. To see this, let us focus on the case when $f$ is a convex quadratic function and $M = I$. Let $H_f$ be the Hessian of $f$. Consider (a) NysADMM $\Theta^k = H_f$ (10) and (b) GD-ADMM $\Theta^k = I$ (11). Further, suppose that NysADMM and GD-ADMM are initialized at 0. The rates of convergence guaranteed by Theorem 1 for Algorithm 2 for both methods are outlined in Table 2. In the first case, the relative smoothness constant

Table 2: Convergence rate comparison of NysADMM and GD-ADMM when initialized at 0 for quadratic $f$.

| Method | NysADMM $\left(\Theta^k = H_f\right)$ | GD-ADMM $\left(\Theta^k = I\right)$ |
|---|---|---|
| Feasibility gap | $\mathcal{O}\left(\frac{1}{t}\left(\sqrt{\frac{2}{\rho}\|x^\star\|_{H_f}^2}\right)\right)$ | $\mathcal{O}\left(\frac{1}{t}\left(\sqrt{\frac{2}{\rho}\lambda_1\left(H_f\right)\|x^\star\|^2}\right)\right)$ |
| Suboptimality gap | $\mathcal{O}\left(\frac{1}{2t}\|x^\star\|_{H_f}^2\right)$ | $\mathcal{O}\left(\frac{1}{2t}\lambda_1\left(H_f\right)\|x^\star\|^2\right)$ |

satisfies $\hat{L}_\Theta = 1$, while in the second, $\hat{L}_\Theta = \lambda_1\left(H_f\right)$, which is the largest eigenvalue of $H_f$. Comparing the rates in Table 2, we see NysADMM improves over GD-ADMM whenever

$$\lambda_1\left(H_f\right)\|x^\star\|^2 \geq \|x^\star\|_{H_f}^2.$$

Hence NysADMM improves significantly over GD-ADMM when $H_f$ exhibits a decaying spectrum, provided $x^\star$ is not concentrated on the top eigenvector of $H_f$. We formalize the latter property in the following definition.

**Definition 4** ($\mu_{\mathbf{V}}$-incoherence)**.** *Let $\mathbf{V}$ be the eigenbasis of $H_f$. We say $x^\star$ is $\mu_{\mathbf{V}}$-incoherent if there exists $1 \leq \mu_{\mathbf{V}} < d$ such that:*

$$\sup_{1 \leq i \leq d} |\langle \mathbf{v}_i, x^\star\rangle|^2 \leq \mu_{\mathbf{V}}\frac{\|x^\star\|^2}{d}. \tag{17}$$

Definition 4 is a weak form of the incoherence condition from compressed sensing and matrix completion, which plays a key role in signal and low-rank matrix recovery (Candes & Romberg, 2007; Candes & Recht, 2012). In words, $x^\star$ is $\mu_{\mathbf{V}}$-incoherent if its energy is not solely concentrated on the top eigenvector of $H_f$ and can be expected to hold generically. The parameter $\mu_{\mathbf{v}}$ controls the allowable concentration. When $\mu_{\mathbf{v}} = 0$, $x^\star$ is orthogonal to $\mathbf{v}$, so its energy is distributed amongst the other eigenvectors. Conversely, the closer $\mu_{\mathbf{v}}$ is to 1, the more $x^\star$ is allowed to concentrate on $\mathbf{v}$.

Using $\mu_{\mathbf{v}}$-incoherence, we can say more about how NysADMM improves on GD-ADMM.

**Proposition 2.** *Suppose $x^\star$ is $\mu_{\mathbf{v}}$-incoherent and $\sigma = \tau\lambda_1(H_f)$ where $\tau \in (0,1)$. Then, the following bound holds*

$$\frac{\lambda_1(H_f)\|x^\star\|^2}{\|x^\star\|_{H_f}^2} \geq \frac{d}{\mu_{\mathbf{V}}\mathrm{intdim}(H_f)}$$

*Hence if $\mu_{\mathbf{v}} + \lambda_2(H_f)/\lambda_1(H_f) \leq (1-\tau)\alpha^{-1}$, where $\alpha \geq 1$, then*

$$\frac{\lambda_1(H_f)\|x^\star\|^2}{\|x^\star\|_{H_f}^2} \geq \alpha.$$

The proof of Proposition 2 is given in Appendix B.2. Proposition 2 shows when $x^\star$ is $\mu_{\mathbf{v}}$-incoherent and $H_f$ has a decaying spectrum, NysADMM yields a significant improvement over GD-ADMM. As a concrete example, consider when $\alpha = 2$, then Proposition 2 implies the ergodic convergence of NysADMM is twice as fast as GD-ADMM. We observe this performance improvement in practice; see Section 8 for corroborating numerical evidence. Just as Newton's method improves on gradient descent for ill-conditioned problems, NysADMM is less sensitive to ill-conditioning than GD-ADMM.

### 5.3 Our approach

To prove Theorem 1, we take the approach in Ouyang et al. (2015), and analyze GeNI-ADMM by viewing Eq. (1) through its formulation as saddle point problem Eq. (2). Let $W = X \times Z \times U$, $\hat{w} = (\hat{x}, \hat{z}, \hat{u})$, and $w = (x, z, u)$, where $w, \hat{w} \in W$. Define the *gap function*

$$Q(\hat{w}, w) \coloneqq [f(x) + g(z) + \langle \rho \hat{u}, Mx + Nz \rangle] - [f(\hat{x}) + g(\hat{z}) + \langle \rho u, M\hat{x} + N\hat{z} \rangle]. \tag{18}$$

The gap function naturally arises from the saddle-point formulation Eq. (2). By construction, it is concave in its first argument and convex in the second, and satisfies the important inequality

$$Q(w^\star, w) = L_\rho(x, z, u^\star) - L_\rho(x^\star, z^\star, u) \geq 0,$$

which follows by definition of $(x^\star, z^\star, u^\star)$ being a saddle-point. Hence the gap function may be viewed as measuring the distance to the saddle $w^\star$.

Further, given a closed set $U \subseteq \mathcal{Z}$ and $v \in \mathcal{Z}$ we define

$$\ell_U(v, w) \coloneqq \sup_{\hat{u} \in U} \{ Q(\hat{w}, w) + \langle v, \rho \hat{u} \rangle \mid \hat{w} = (\hat{x}, \hat{z}, \hat{u}), \ \hat{x} = x^\star, \ \hat{z} = z^\star \} \tag{19}$$

$$= f(x) + g(z) - p^\star + \sup_{\hat{u} \in U} \langle \rho \hat{u}, v - (Mx + Nz) \rangle.$$

The following lemma of Ouyang et al. (2015) relates $Q(\hat{w}, w)$ to the suboptimality and feasibility gaps.

**Lemma 2.** *For any $U \subseteq \mathcal{Z}$, suppose $\ell_U(Mx + Nz, w) \leq \epsilon < \infty$ and $\|Mx + Nz\| \leq \delta$, where $w = (x, z, u) \in W$. Then*

$$f(x) + g(z) - p^\star \leq \epsilon. \tag{20}$$

*In other words, $(x, z)$ is an approximate solution of* (1) *with suboptimality gap $\epsilon$ and feasibility gap $\delta$. Further, if $U = \mathcal{Z}$, for any $v$ such that $\ell_U(v, w) \leq \epsilon < \infty$ and $\|v\| \leq \delta$, we have $v = Mx - z$.*

Lemma 2 shows that if we can find $w$ such that $\ell_U(Mx + Nz, w) \leq \epsilon$ and $\|Mx + Nz\| \leq \delta$, then we have an approximate optimal solution to (1) with gaps $\epsilon$ and $\delta$, that is, $\ell_U(Mx + Nz, w)$ controls the suboptimality and feasibility gaps.

### 5.4 Controlling the gap function

Lemma 2 shows the key to establishing convergence of GeNI-ADMM is to achieve appropriate control over the gap function. To accomplish this, we use the optimality conditions of the $x$ and $z$ subproblems. However, as the subproblems are only solved approximately, the inexact solutions satisfy *perturbed* optimality conditions. To be able to reason about the optimality conditions under inexact solutions, the iterates must remain bounded. Indeed, if the iterates are unbounded, they can fail to satisfy the subproblem optimality conditions arbitrarily badly. Fortunately, the following proposition shows the iterates remain bounded.

**Proposition 3** (GeNI-ADMM iterates remain bound). *Let $\{\varepsilon_x^k\}_{k \geq 1}, \{\varepsilon_z^k\}_{k \geq 1}, \{\Theta^k\}_{k \geq 1}, \{\Psi^k\}_{k \geq 1}$, and $\rho > 0$ be given. Instate Assumptions 1-4. Run Algorithm 2, then the output sequences $\{\tilde{x}^k\}_{k \geq 1}, \{\tilde{z}^k\}_{k \geq 1}, \{\tilde{u}^k\}_{k \geq 1}$ are bounded. That is, there exists $R > 0$, such that*

$$\{\tilde{x}^k\}_{k \geq 1} \subset B(x^\star, R), \quad \{\tilde{z}^k\}_{k \geq 1} \subset B(z^\star, R), \quad \{\tilde{u}^k\}_{k \geq 1} \subset B(u^\star, R).$$

The proof of Proposition 3 is provided in Appendix B.2.

As the iterates remain bounded, we can show that the optimality conditions are approximately satisfied at each iteration. The precise form of these perturbed optimality conditions is given in Lemmas 3 and 4. Detailed proofs establishing these lemmas are given in Appendix B.2.

**Lemma 3** (Inexact $x$-optimality condition). *Instate Assumptions 1-4. Suppose $\tilde{x}^{k+1}$ is an $\varepsilon_x^k$-minimizer of the $x$-subproblem under Assumption 3. Then for some absolute constant $C_x > 0$ no larger than $\max\{L_{f_1}R, \ (\theta_{max} + \rho\lambda_1(M^T M))R, \ L_{f_1}K_{\varepsilon_x}, \ (\theta_{max} + \rho\lambda_1(M^T M))K_{\varepsilon_x}, \ \sup_k \|\nabla S_x(x^k)\|\}$, we have*

$$\langle \nabla f_1(\tilde{x}^{k+1}) + \nabla f_2(\tilde{x}^k), \tilde{x}^{k+1} - x^\star \rangle \leq \langle \Theta^k(\tilde{x}^{k+1} - \tilde{x}^k), x^\star - \tilde{x}^{k+1} \rangle$$
$$+ \rho \langle M\tilde{x}^{k+1} + N\tilde{z}^k + \tilde{u}^k, M(x^\star - \tilde{x}^{k+1}) \rangle + C_x \varepsilon_x^k. \tag{21}$$

**Lemma 4** (Inexact $z$-optimality condition). *Instate Assumptions 1-4. Suppose $\tilde{z}^{k+1}$ is an $\varepsilon_z^k$-minimum of the $z$-subproblem under Definition 1. Then for some absolute constant $C_z > 0$ no larger than $R/\sqrt{\psi_{min} + \rho\lambda_{min}(N^T N)} + K_{\varepsilon_z^k}$, we have*

$$g(z^\star) - g(\tilde{z}^{k+1}) \geq \langle -\rho N^T \tilde{u}^{k+1} + \Psi^k(\tilde{z}^k - \tilde{z}^{k+1}), z^\star - \tilde{z}^{k+1}\rangle - C_z\sqrt{\varepsilon_z^k}. \tag{22}$$

When $\varepsilon_x^k = \varepsilon_z^k = 0$ the approximate optimality conditions of Lemmas 3 and 4 collapse to the exact optimality conditions. We also note while Lemma 3 (Lemma 4) necessarily holds when $\tilde{x}^{k+1}$ ($\tilde{z}^{k+1}$) is an $\varepsilon_x^k$-approximate minimizer ($\varepsilon_z^k$-minimum), the converse does not hold. With Lemmas 3 and 4 in hand, we can establish control of the gap function for one iteration.

**Lemma 5.** *Instate Assumptions 1-4. Let $\tilde{w}^{k+1} = (\tilde{x}^{k+1}, \tilde{z}^{k+1}, \tilde{u}^{k+1})$ denote the iterates generated by Algorithm 2 at iteration $k$. Set $w = (x^\star, z^\star, u)$, then the gap function $Q$ satisfies*

$$Q(w, \tilde{w}^{k+1}) \leq \frac{1}{2}\left(\|\tilde{x}^k - x^\star\|_{\Theta^k}^2 - \|\tilde{x}^{k+1} - x^\star\|_{\Theta^k}^2\right) + \frac{1}{2}\left(\|\tilde{z}^k - z^\star\|_{\Psi^k + \rho N^T N}^2 - \|\tilde{z}^{k+1} - z^\star\|_{\Psi^k + \rho N^T N}^2\right)$$
$$+ \frac{\rho}{2}\left(\|\tilde{u}^k - u\|^2 - \|\tilde{u}^{k+1} - u\|^2\right) + C_x\varepsilon_x^k + C_z\sqrt{\varepsilon_z^k}.$$

*Proof.* From the definition of $Q$,

$$Q(w, \tilde{w}^{k+1}) = f(\tilde{x}^{k+1}) - f(x^\star) + g(\tilde{z}^{k+1}) - g(z^\star) + \langle \rho u, M\tilde{x}^{k+1} + N\tilde{z}^{k+1}\rangle - \langle \rho\tilde{u}^{k+1}, Mx^\star + Nz^\star\rangle.$$

Our goal is to upper bound $Q(w, \tilde{w}^{k+1})$. We start by bounding $f(\tilde{x}^{k+1}) - f(x^\star)$ as follows:

$$f(\tilde{x}^{k+1}) - f(x^\star) = \left(f_1(\tilde{x}^{k+1}) - f_1(x^\star)\right) + \left(f_2(\tilde{x}^{k+1}) - f_2(\tilde{x}^k)\right) + \left(f_2(\tilde{x}^k) - f_2(x^\star)\right)$$

$$\overset{(1)}{\leq} \langle \nabla f_1(\tilde{x}^{k+1}), \tilde{x}^{k+1} - \tilde{x}^\star\rangle + \langle \nabla f_2(\tilde{x}^k), \tilde{x}^{k+1} - \tilde{x}^k\rangle + \frac{1}{2}\|\tilde{x}^{k+1} - \tilde{x}^k\|_{\Theta^k}^2 + f_2(\tilde{x}^k) - f_2(x^\star)$$

$$\overset{(2)}{\leq} \langle \nabla f_1(\tilde{x}^{k+1}), \tilde{x}^{k+1} - \tilde{x}^\star\rangle + \langle \nabla f_2(\tilde{x}^k), \tilde{x}^{k+1} - \tilde{x}^k\rangle + \frac{1}{2}\|\tilde{x}^{k+1} - \tilde{x}^k\|_{\Theta^k}^2 + \langle \nabla f_2(\tilde{x}^k), \tilde{x}^k - x^\star\rangle$$

$$= \langle \nabla f_1(\tilde{x}^{k+1}) + \nabla f_2(\tilde{x}^k), \tilde{x}^{k+1} - x^\star\rangle + \frac{1}{2}\|\tilde{x}^{k+1} - \tilde{x}^k\|_{\Theta^k}^2,$$

where (1) uses convexity of $f_1$ and 1-relative smoothness of $f_2$, and (2) uses convexity of $f_2$. Inserting the upper bound on $f(\tilde{x}^{k+1}) - f(x)$ into the expression for $Q(w, \tilde{w}^{k+1})$, we find

$$Q(w, \tilde{w}^{k+1}) \leq \langle \nabla f_1(\tilde{x}^{k+1}) + \nabla f_2(\tilde{x}^k), \tilde{x}^{k+1} - x^\star\rangle + \frac{1}{2}\|\tilde{x}^{k+1} - \tilde{x}^k\|_{\Theta^k}^2 + g(\tilde{z}^{k+1}) - g(z^\star)$$
$$+ \langle \rho u, M\tilde{x}^{k+1} + N\tilde{z}^{k+1}\rangle - \langle \rho\tilde{u}^{k+1}, Mx^\star + Nz^\star\rangle. \tag{23}$$

Now, using the inexact optimality condition for the $x$-subproblem (Lemma 3), the above display becomes

$$Q(w, \tilde{w}^{k+1}) \leq \langle \Theta^k(\tilde{x}^{k+1} - \tilde{x}^k), x^\star - \tilde{x}^{k+1}\rangle + \rho\langle M\tilde{x}^{k+1} + N\tilde{z}^k + \tilde{u}^k, M(x^\star - \tilde{x}^{k+1})\rangle + C_x\varepsilon_x^k$$
$$+ \frac{1}{2}\|\tilde{x}^{k+1} - \tilde{x}^k\|_{\Theta^k}^2 + g(\tilde{z}^{k+1}) - g(z^\star) + \langle \rho u, M\tilde{x}^{k+1} + N\tilde{z}^{k+1}\rangle - \langle \rho\tilde{u}^{k+1}, Mx^\star + Nz^\star\rangle.$$

Similarly, applying the inexact optimality for the $z$-subproblem (Lemma 4), we further obtain

$$Q(w, \tilde{w}^{k+1}) \leq \langle \Theta^k(\tilde{x}^{k+1} - \tilde{x}^k), x^\star - \tilde{x}^{k+1}\rangle + \rho\langle M\tilde{x}^{k+1} + N\tilde{z}^k + \tilde{u}^k, M(x^\star - \tilde{x}^{k+1})\rangle + C_x\varepsilon_x^k$$
$$+ \frac{1}{2}\|\tilde{x}^{k+1} - \tilde{x}^k\|_{\Theta^k}^2 - \rho\langle N^T\tilde{u}^{k+1}, \tilde{z}^{k+1} - z^\star\rangle + \langle \Psi^k(\tilde{z}^{k+1} - \tilde{z}^k), z^\star - \tilde{z}^{k+1}\rangle + C_z\sqrt{\varepsilon_z^k} \tag{24}$$
$$+ \langle \rho u, M\tilde{x}^{k+1} + N\tilde{z}^{k+1}\rangle - \langle \rho\tilde{u}^{k+1}, Mx^\star + Nz^\star\rangle.$$

We now simplify (24) by combining terms. Some basic manipulations show the terms on line 2 of (24) may be rewritten as

$$-\rho\langle N^T\tilde{u}^{k+1}, \tilde{z}^{k+1} - z^\star\rangle + \rho\langle u, M\tilde{x}^{k+1} + N\tilde{z}^{k+1}\rangle - \rho\langle \tilde{u}^{k+1}, Mx^\star + Nz^\star\rangle$$
$$= \rho\langle u - \tilde{u}^{k+1}, \tilde{M}\tilde{x}^{k+1} + N\tilde{z}^{k+1}\rangle - \rho\langle \tilde{u}^{k+1}, M(x^\star - \tilde{x}^{k+1})\rangle.$$

We can combine the preceding display with the second term of line 1 in (24) to reach

$$
\begin{aligned}
&\rho\langle M\tilde{x}^{k+1} + N\tilde{z}^k + \tilde{u}^k, M(x^\star - \tilde{x}^{k+1})\rangle - \rho\langle\tilde{u}^{k+1}, M(x^\star - \tilde{x}^{k+1})\rangle + \rho\langle u - \tilde{u}^{k+1}, M\tilde{x}^{k+1} + N\tilde{z}^{k+1}\rangle \\
&= \rho\langle\tilde{u}^{k+1} + N(\tilde{z}^k - \tilde{z}^{k+1}), M(x^\star - \tilde{x}^{k+1})\rangle - \rho\langle\tilde{u}^{k+1}, M(x^\star - \tilde{x}^{k+1})\rangle + \rho\langle u - \tilde{u}^{k+1}, M\tilde{x}^{k+1} + N\tilde{z}^{k+1}\rangle \\
&= \rho\langle N(\tilde{z}^k - \tilde{z}^{k+1}), M(x^\star - \tilde{x}^{k+1})\rangle + \rho\langle u - \tilde{u}^{k+1}, M\tilde{x}^{k+1} + N\tilde{z}^{k+1}\rangle \\
&= \rho\langle N^-(\tilde{z}^{k+1} - \tilde{z}^k), M(x^\star - \tilde{x}^{k+1})\rangle + \rho\langle\tilde{u}^{k+1} - u, \tilde{u}^k - \tilde{u}^{k+1}\rangle,
\end{aligned}
$$

where $N^- = -N$. Inserting the preceding simplification into (24), we reach

$$
\begin{aligned}
Q(w, \tilde{w}^{k+1}) \leq{}& \langle\Theta^k(\tilde{x}^{k+1} - \tilde{x}^k), x^\star - \tilde{x}^{k+1}\rangle + \rho\langle\tilde{u}^{k+1} - u, \tilde{u}^k - \tilde{u}^{k+1}\rangle \\
&+ \langle\Psi^k(\tilde{z}^{k+1} - \tilde{z}^k), z^\star - \tilde{z}^{k+1}\rangle + \rho\langle N^-(\tilde{z}^{k+1} - \tilde{z}^k), M(x^\star - \tilde{x}^{k+1})\rangle \\
&+ \frac{1}{2}\|\tilde{x}^{k+1} - \tilde{x}^k\|_{\Theta^k}^2 + C_x\varepsilon_x^k + C_z\sqrt{\varepsilon_z^k}.
\end{aligned}
\tag{25}
$$

Now, we bound the first two leading terms in line 1 of (25) by invoking the identity $\langle a - b, \Upsilon(c - d)\rangle = 1/2\left(\|a - d\|_\Upsilon^2 - \|a - c\|_\Upsilon^2\right) + 1/2\left(\|c - b\|_\Upsilon^2 - \|d - b\|_\Upsilon^2\right)$ to obtain

$$
\begin{aligned}
&\langle\Theta^k(\tilde{x}^{k+1} - \tilde{x}^k), x^\star - \tilde{x}^{k+1}\rangle + \rho\langle\tilde{u}^{k+1} - u, \tilde{u}^k - \tilde{u}^{k+1}\rangle = \\
&\frac{1}{2}\left(\|x^\star - \tilde{x}^k\|_{\Theta^k}^2 - \|x^\star - \tilde{x}^{k+1}\|_{\Theta^k}^2\right) - \frac{1}{2}\|\tilde{x}^{k+1} - \tilde{x}^k\|_{\Theta^k}^2 + \frac{\rho}{2}\left(\|\tilde{u}^k - u\|^2 - \|\tilde{u}^{k+1} - u\|^2 - \|\tilde{u}^k - \tilde{u}^{k+1}\|^2\right).
\end{aligned}
$$

Similarly, to bound the third and fourth terms in (25), we again invoke $(a - b)^T\Upsilon(c - d) = 1/2\left(\|a - d\|_\Upsilon^2 - \|a - c\|_\Upsilon^2\right) + 1/2\left(\|c - b\|_\Upsilon^2 - \|d - b\|_\Upsilon^2\right)$ which yields

$$
\langle\Psi^k(\tilde{z}^{k+1} - \tilde{z}^k), z^\star - \tilde{z}^{k+1}\rangle = \frac{1}{2}\left(\|\tilde{z}^k - z^\star\|_{\Psi^k}^2 - \|\tilde{z}^{k+1} - z^\star\|_{\Psi^k}^2\right) - \frac{1}{2}\|\tilde{z}^{k+1} - \tilde{z}^k\|_{\Psi^k}^2
$$

and

$$
\begin{aligned}
&\rho\langle N^-(\tilde{z}^k - \tilde{z}^{k+1}), M(\tilde{x}^{k+1} - x^\star)\rangle \\
&= \frac{\rho}{2}\left(\|N^-\tilde{z}^k - Mx^\star\|^2 - \|\tilde{z}^{k+1} - Mx^\star\|^2 + \|N^-\tilde{z}^{k+1} - M\tilde{x}^{k+1}\|^2 - \|N^-\tilde{z}^k - M\tilde{x}^{k+1}\|^2\right) \\
&= \frac{\rho}{2}\left(\|N\tilde{z}^k - Nz^\star\|^2 - \|N\tilde{z}^{k+1} - Nz^\star\|^2 - \|M\tilde{x}^{k+1} + N\tilde{z}^k\|^2\right) + \frac{\rho}{2}\|\tilde{u}^k - \tilde{u}^{k+1}\|^2.
\end{aligned}
$$

Putting everything together, we conclude

$$
\begin{aligned}
Q(w, \tilde{w}^{k+1}) \leq{}& \frac{1}{2}\left(\|\tilde{x}^k - x^\star\|_{\Theta^k}^2 - \|\tilde{x}^{k+1} - x^\star\|_{\Theta^k}^2\right) + \frac{1}{2}\left(\|\tilde{z}^k - z^\star\|_{\Psi^k+\rho N^TN}^2 - \|\tilde{z}^{k+1} - z^\star\|_{\Psi^k+\rho N^TN}^2\right) \\
&+ \frac{\rho}{2}\left(\|\tilde{u}^k - u\|^2 - \|\tilde{u}^{k+1} - u\|^2\right) + C_x\varepsilon_x^k + C_z\sqrt{\varepsilon_z^k}.
\end{aligned}
$$

as desired. $\qquad\square$

### 5.5 Proof of Theorem 1

With Lemma 5 in hand, we are ready to prove Theorem 1.

*Proof.* From Lemma 5, we have for each $k$ that

$$
\begin{aligned}
Q(w, \tilde{w}^{k+1}) \leq{}& \frac{1}{2}\left(\|\tilde{x}^k - x^\star\|_{\Theta^k}^2 - \|\tilde{x}^{k+1} - x^\star\|_{\Theta^k}^2\right) + \frac{1}{2}\left(\|\tilde{z}^k - z^\star\|_{\Psi^k+\rho N^TN}^2 - \|\tilde{z}^{k+1} - z^\star\|_{\Psi^k+\rho N^TN}^2\right) \\
&+ \frac{\rho}{2}\left(\|\tilde{u}^k - u\|^2 - \|\tilde{u}^{k+1} - u\|^2\right) + C_x\varepsilon_x^k + C_z\sqrt{\varepsilon_z^k}.
\end{aligned}
$$

Now, summing up the preceding display from $k = 1$ to $t$ and using $w = (x^\star, z^\star, u)$, we obtain

$$
\sum_{k=2}^{t+1} Q(x^\star, z^\star, u; \tilde{w}^k) \leq \underbrace{\frac{1}{2} \sum_{k=1}^{t} \left( \|\tilde{x}^k - x^\star\|_{\Theta^k}^2 - \|\tilde{x}^{k+1} - x^\star\|_{\Theta^k}^2 \right)}_{T_1}
$$

$$
+ \underbrace{\frac{1}{2} \sum_{k=1}^{t} \left( \|\tilde{z}^k - z^\star\|_{\Psi^k + \rho N^T N}^2 - \|\tilde{z}^{k+1} - z^\star\|_{\Psi^k + \rho N^T N}^2 \right)}_{T_2}
$$

$$
+ \underbrace{\frac{\rho}{2} \sum_{k=1}^{t} \left( \|\tilde{u}^k - u\|^2 - \|\tilde{u}^{k+1} - u\|^2 \right)}_{T_3} + C_x \mathcal{E}_x + C_z \mathcal{E}_z.
$$

We now turn to bounding $T_1$. Using the definition of $T_1$, we find

$$
T_1 = \frac{1}{2} \sum_{k=1}^{t} \left( \|\tilde{x}^k - x^\star\|_{\Theta^k}^2 - \|\tilde{x}^{k+1} - x^\star\|_{\Theta^k}^2 \right)
$$

$$
= \frac{1}{2} \left( \|\tilde{x}^1 - x^\star\|_{\Theta^1}^2 - \|\tilde{x}^{t+1} - x^\star\|_{\Theta^t}^2 \right) + \frac{1}{2} \sum_{k=2}^{t} \left( \|\tilde{x}^k - x^\star\|_{\Theta^k}^2 - \|\tilde{x}^k - x^\star\|_{\Theta^{k-1}}^2 \right).
$$

Now using our hypotheses on the sequence $\{\Theta^k\}_k$ in (15), we obtain

$$
\|\tilde{x}^k - x^\star\|_{\Theta^k}^2 - \|\tilde{x}^k - x^\star\|_{\Theta^{k-1}}^2 = (\tilde{x}^k - x^\star)^T (\Theta^k - \Theta^{k-1})(\tilde{x}^k - x^\star) \leq \zeta^{k-1} \|\tilde{x}^k - x^\star\|_{\Theta^{k-1}}^2
$$

$$
\leq \tau_\zeta \zeta^{k-1} \|\tilde{x}^k - x^\star\|_{\Theta^1}^2.
$$

Inserting the previous bound into $T_1$, we reach

$$
T_1 \leq \frac{1}{2} \|\tilde{x}^1 - x^\star\|_{\Theta^1}^2 + \frac{1}{2} \tau_\zeta \sum_{k=1}^{t} \zeta^{k-1} \|\tilde{x}^k - x^\star\|_{\Theta^1}^2 - \frac{1}{2} \|\tilde{x}^{t+1} - x^\star\|_{\Theta^t}^2
$$

$$
\leq \frac{1}{2} \left( \|\tilde{x}^1 - x^\star\|_{\Theta^1}^2 + \tau_\zeta R \lambda_1(\Theta^1) \mathcal{E}_\zeta \right) - \frac{1}{2} \|\tilde{x}^{t+1} - x^\star\|_{\Theta^t}^2
$$

$$
\leq \frac{1}{2} \left( d_{x^\star, \Theta^1}^2 + \tau_\zeta R \lambda_1(\Theta^1) \mathcal{E}_\zeta \right).
$$

Next, we bound $T_2$.

$$
T_2 = \frac{1}{2} \sum_{k=1}^{t} \left( \|\tilde{z}^k - z^\star\|_{\Psi^k + \rho N^T N}^2 - \|\tilde{z}^{k+1} - z^\star\|_{\Psi^k + \rho N^T N}^2 \right)
$$

$$
= \frac{1}{2} \left( \|\tilde{z}^1 - z^\star\|_{\Psi^1 + \rho N^T N}^2 - \|\tilde{z}^{t+1} - z^\star\|_{\Psi^t + \rho N^T N}^2 \right) + \frac{1}{2} \sum_{k=2}^{t} \left( \|\tilde{z}^k - z^\star\|_{\Psi^k + \rho N^T N}^2 - \|\tilde{x}^k - x^\star\|_{\Psi^{k-1} + \rho N^T N}^2 \right)
$$

$$
\leq \frac{1}{2} \|\tilde{z}^1 - z^\star\|_{\Psi^1 + \rho N^T N}^2 + \frac{1}{2} \tau_\zeta \sum_{k=1}^{t} \zeta^{k-1} \|\tilde{z}^k - z^\star\|_{\Psi^1 + \rho N^T N}^2
$$

$$
\leq \frac{1}{2} \left( \|\tilde{z}^1 - z^\star\|_{\Psi^1 + \rho N^T N}^2 + \tau_\zeta R \lambda_1(\Psi^1 + \rho N^T N) \mathcal{E}_\zeta \right).
$$

$$
= \frac{1}{2} \left( d_{z^\star, \Psi_{\rho, N}^1} + \tau_\zeta R \lambda_1(\Psi^1 + \rho N^T N) \mathcal{E}_\zeta \right)
$$

Last, $T_3$ is a telescoping sum, hence

$$T_2 = \frac{\rho}{2} \sum_{k=1}^{t} \left( \|\tilde{u}^k - u\|^2 - \|\tilde{u}^{k+1} - u\|^2 \right) = \frac{\rho}{2} \left( \|\tilde{u}^1 - u\|^2 - \|\tilde{u}^{t+1} - u\|^2 \right).$$

Defining $C_\zeta = \tau_\zeta R \left( \lambda_1(\Theta^1) + \lambda_1(\Psi^1 + \rho N^T N) \right)$ and using our bounds on $T_1$ through $T_3$, we find

$$\sum_{k=2}^{t+1} Q(x^\star, z^\star, u; \tilde{w}^k) \leq \frac{1}{2} \left( d^2_{x^\star, \Theta^1} + C_\zeta \mathcal{E}_\zeta \right) + \frac{1}{2} d^2_{z^\star, \Psi^1_{\rho, N}} + \frac{1}{2} C_\zeta \mathcal{E}_\zeta + C_x \mathcal{E}_x + C_z \mathcal{E}_z$$

$$+ \frac{\rho}{2} \left( \|\tilde{u}^1 - u\|^2 - \|\tilde{u}^{t+1} - u\|^2 \right)$$

$$= \frac{1}{2} d^2_{x^\star, \Theta^1} + \frac{1}{2} d^2_{z^\star, \Psi^1_{\rho, N}} + C_x \mathcal{E}_x + C_z \mathcal{E}_z + C_\zeta \mathcal{E}_\zeta + \frac{\rho}{2} \left( \|\tilde{u}^1 - u\|^2 - \|\tilde{u}^{t+1} - u\|^2 \right),$$

Now, as $\bar{w}^{t+1} = \frac{1}{t} \sum_{k=2}^{t+1} \tilde{w}^k$, the convexity of $Q$ in its second argument yields

$$Q(x^\star, z^\star, u; \bar{w}^{t+1}) \leq \frac{1}{t} \sum_{k=2}^{t+1} Q(x^\star, z^\star, u; \tilde{w}^k)$$

$$\leq \frac{1}{t} \left( \frac{1}{2} d^2_{x^\star, \Theta^1} + \frac{\rho}{2} d^2_{z^\star, \Psi^1_{\rho, N}} + C_x \mathcal{E}_x + C_z \mathcal{E}_z + C_\zeta \mathcal{E}_\zeta + \frac{\rho}{2} \left( \|\tilde{u}^1 - u\|^2 - \|\tilde{u}^{t+1} - u\|^2 \right) \right). \tag{26}$$

Define $\Gamma := \frac{1}{2} d^2_{x^\star, \Theta^1} + \frac{1}{2} d^2_{z^\star, \Psi^1_{\rho, N}} + C_x \mathcal{E}_x + C_z \mathcal{E}_z + C_\zeta \mathcal{E}_\zeta$. Since $Q(w^\star, \bar{w}^{t+1}) \geq 0$, by (26) we reach

$$\|\tilde{u}^{t+1} - u^\star\|^2 \leq \frac{2}{\rho} \Gamma + d^2_{u^\star}.$$

Let $\tilde{v}^{t+1} = \frac{1}{t} \left( \tilde{u}^1 - \tilde{u}^{t+1} \right)$. Then we can bound $\|\tilde{v}_{t+1}\|^2$ as

$$\|\tilde{v}^{t+1}\|^2 \leq \frac{2}{t^2} \left( \|\tilde{u}^1 - u^\star\|^2 + \|\tilde{u}^{t+1} - u^\star\|^2 \right) \leq \frac{4}{t^2} \left( \frac{\Gamma}{\rho} + d^2_{u^\star} \right).$$

By (26), given the fact $\tilde{u}^1 = 0$, we also have

$$Q(x^\star, z^\star, u; \bar{w}^{t+1}) \leq \frac{\Gamma - \rho \langle \tilde{u}^1 - \tilde{u}^{t+1}, u \rangle}{t} = \frac{\Gamma}{t} - \rho \langle \tilde{v}^{t+1}, u \rangle,$$

where the equality follows from the definition of $\tilde{v}^{t+1}$. Hence for any $u$

$$Q(x^\star, z^\star, u; \bar{w}^{t+1}) + \langle \tilde{v}^{t+1}, \rho u \rangle \leq \frac{\Gamma}{t},$$

and therefore

$$\ell_U(\tilde{v}^{t+1}, \bar{w}^{t+1}) \leq \frac{1}{t} \left( \frac{1}{2} d^2_{x^\star, \Theta^1} + \frac{1}{2} d^2_{z^\star, \Psi^1_{\rho, N}} + C_x \mathcal{E}_x + C_z \mathcal{E}_z + C_\zeta \mathcal{E}_\zeta \right).$$

We finish the proof by invoking Lemma 2. $\square$

# 6 Linear convergence of GeNI-ADMM

In this section, we seek to establish linear convergence results for Algorithm 2. In general, the linear convergence of ADMM relies on strong convexity of the objective function (Boyd et al., 2011; Nishihara et al., 2015; Parikh & Boyd, 2014). Consistently, the linear convergence of GeNI-ADMM also requires strong convexity. Many applications of GeNI-ADMM fit into this setting, such as elastic net (Friedman et al., 2010). However, linear convergence is not restricted to strongly convex problems. It has been shown that local linear convergence of ADMM can be guaranteed even without strong convexity (Yuan et al., 2020).

Experiments in Section 8 show the same phenomenon for GeNI-ADMM: it converges linearly after a couple of iterations when the iterates reach some manifold containing the solution. The linear convergence theory of GeNI-ADMM provides a way to understand this phenomenon. We first list the additional assumptions required for linear convergence:

**Assumption 5** (Optimization is over the whole space). *The sets $X$ and $Z$ in (1) satisfy*

$$X = \mathcal{X}, \ and \ Z = \mathcal{Z}.$$

Assumption 5 states that the optimization problem in (1) is over the entire spaces $\mathcal{X}$ and $\mathcal{Z}$, not closed subsets. This assumption is met in many practical optimization problems of interest, where $\mathcal{X} = \mathcal{Z} = \mathcal{H} = \mathbb{R}^d$. Moreover, it is consistent with prior analyses such as Deng & Yin (2016), who specialize their analysis to the setting of the last sentence.

**Assumption 6** (Regularity of $f$). *The function $f$ is finite valued, strongly convex with parameter $\sigma_f$, and smooth with parameter $L_f$.*

Assumption 6 imposes standard regularity conditions on $f$, in addition to the conditions of Assumption 2.

**Assumption 7** (Non-degeneracy of constraint operators). *The linear operators $MM^T$ and $N^TN$ are invertible.*

Assumption 7 is consistent with prior analyses of ADMM-type schemes under strong convexity, such as Deng & Yin (2016), who make this assumption in their analysis of Generalized ADMM. Moreover, the assumption holds in many important application problems, especially those that arise from machine learning where, typically, $M = I$ and $N = -I$.

**Assumption 8** (Geometric decay of the forcing sequences). *There exists a constant $q > 0$ such that the forcing sequences $\{\varepsilon_x^k\}_{k\geq 1}$, $\{\varepsilon_z^k\}_{k\geq 1}$ satisfy*

$$\varepsilon_x^{k+1} \leq \varepsilon_x^k/(1+q), \ and \ \varepsilon_z^{k+1} \leq \varepsilon_z^k/(1+q)^2. \tag{27}$$

*Moreover, we assume Algorithm 2 is equipped with oracles for solving the $x$ and $z$-subproblems, which at each iteration produce approximate solutions $\tilde{x}^{k+1}$, $\tilde{z}^{k+1}$ satisfying:*

$$\tilde{x}^{k+1} \overset{\varepsilon_x^k}{\approx} \underset{x \in X}{\arg\min}\{f_1(x) + \langle \nabla f_2(\tilde{x}^k), x - \tilde{x}^k \rangle + \frac{1}{2}\|x - \tilde{x}^k\|_{\Theta^k}^2 + \frac{\rho}{2}\|Mx + N\tilde{z}^k + \tilde{u}^k\|^2\},$$

$$\tilde{z}^{k+1} \overset{\varepsilon_z^k}{\approx} \underset{z \in Z}{\arg\min}\{g(z) + \frac{\rho}{2}\|M\tilde{x}^{k+1} + Nz + \tilde{u}^k\|^2 + \frac{1}{2}\|z - \tilde{z}^k\|_{\Psi^k}^2\}.$$

Assumption 8 replaces Assumption 3 and requires the inexactness sequences to decay geometrically. Compared with the sublinear convergence result, linear convergence requires more accurate solutions to the subproblems. Again, since the $z$-subproblem inexactness is weaker than the $x$-subproblem inexactness, $\{\varepsilon_z^k\}_{k\geq 1}$ should have a faster decay rate $(1+q)^2$ than the decay rate $(1+q)$ of $\{\varepsilon_x^k\}_{k\geq 1}$.

The requirement that the forcing sequences decay geometrically is somewhat burdensome, as it leads to the subproblems needing to be solved to higher accuracy sooner than if the forcing sequences were only summable. Fortunately, this condition seems to be an artifact of the analysis; our numerical experiments with strongly convex $f$ (Section 8.2) only use summable forcing sequences but show linear convergence of GeNI-ADMM.

## 6.1 Our approach

Inspired by Deng & Yin (2016), we take a *Lyapunov function* approach to proving linear convergence. Let $\tilde{w} = (\tilde{x}, \tilde{z}, \tilde{u})$, and $w^\star = (x^\star, z^\star, u^\star)$. We define the Lyapunov function:

$$\Phi^k = \frac{1}{\rho}\|\tilde{x}^k - x^\star\|_{\Theta^k}^2 + \frac{1}{\rho}\|\tilde{z}^k - \tilde{z}^\star\|_{\Psi^k + \rho N^T N}^2 + \|\tilde{u}^k - u^\star\|^2 = \|\tilde{w}^k - w^\star\|_{G^k}^2,$$

where

$$G^k := \begin{pmatrix} \frac{1}{\rho}\Theta^k & 0 & 0 \\ 0 & \frac{1}{\rho}\Psi^k + N^T N & 0 \\ 0 & 0 & I \end{pmatrix}.$$

Our main result in this section is the following theorem, which shows the Lyapunov function converges linearly to 0.

**Theorem 2.** *Instate Assumptions 1-2, and Assumptions 4-8. Moreover, suppose that $\theta_{\min}$ in (14) satisfies $\theta_{\min} > L_f^2/\sigma_f$. Then there exist constants $\delta$ and $S > 0$ such that if $q > \delta$,*

$$(1+\delta)^k \Phi^k \leq \tau_\zeta \Phi^1 + S. \tag{28}$$

*Hence after $k = \mathcal{O}\left(\frac{1}{\delta}\log\left(\frac{\tau_\zeta \Phi^1}{\epsilon}\right)\right)$ iterations,*

$$\Phi^k \leq \epsilon.$$

The proof of Theorem 2 is deferred to Section 6.3. As $G^k \succ 0$ for all $k$, Theorem 2 implies the iterates $(\tilde{x}^k, \tilde{z}^k, \tilde{u}^k)$ converge linearly to optimum $(x^\star, z^\star, u^\star)$. Thus, despite inexactly solving approximations of the original ADMM subproblems, GeNI-ADMM still enjoys linear convergence when the objective is strongly convex. The requirement in Theorem 2 that $\theta_{\min} > L_f^2/\sigma_f$ is not present in Deng & Yin (2016). Recall the Generalized ADMM scheme analyzed in Deng & Yin (2016) does not employ function linearization and only uses fixed metrics. The additional condition on $\theta_{\min}$ arises from the linearization of $f$. In particular, it stems from lower bounding the term $-\langle \nabla f_2(\tilde{x}^{k+1}) - \nabla f_2(\tilde{x}^k), \tilde{x}^{k+1} - x^\star \rangle$ in Lemma 7, which vanishes when $f$ is not linearized as $f_2 = 0$. Given GeNI-ADMM allows for this further approximation in the $x$-subproblem, it is perhaps unsurprising that an additional condition is required to ensure convergence.

The new condition on $\theta_{\min}$ can always be enforced by adding a damping term $\sigma I$ to $\Theta^k$. However, this could be undesirable as a large regularization term might lead GeNI-ADMM to converge slower. Empirically in Section 8, we find that GeNI-ADMM converges linearly without enforcing this condition. It is an interesting direction for future work to see if this condition can be removed while guaranteeing linear convergence.

### 6.2 Sufficient descent

From Theorem 2, to establish linear convergence of GeNI-ADMM, it suffices to show $\Phi^k$ decreases geometrically. We take two steps to achieve this. First, we show that $\|\tilde{w}^k - w^\star\|_{G^k}^2 - \|\tilde{w}^{k+1} - w^\star\|_{G^k}^2$ decreases for every iteration $k$ (Lemma 7). Second, we show that $\Phi^{k+1}$ decreases geometrically by a factor of $1/(1+\delta)$ with respect to $\Phi^k$ and some small error terms that stem from inexactness (Lemma 8).

As in the convex case, the optimality conditions of the subproblems play a vital role in the analysis. Since the subproblems are only solved approximately, we must again consider the inexactness of the solutions in these two steps. For the first step, we use strong convexity of $f$ and convexity of $g$ with appropriate perturbations to account for the inexactness. We call these conditions *perturbed convexity* conditions, as outlined in Lemma 6.

**Lemma 6** (Perturbed convexity)**.** *Instate the assumptions of Theorem 2. Let $\tilde{x}^{k+1}$ and $\tilde{z}^{k+1}$ be the inexact solutions of $x$ and $z$-subproblems under Definition 1. Recall $(x^\star, z^\star, u^\star)$ is a saddle point of (1). Then for some constant $C \geq 0$, the following inequalities are satisfied:*

1. (Semi-inexact $f$-strong convexity)

$$\langle \tilde{x}^k - \tilde{x}^{k+1}, \tilde{x}^{k+1} - x^\star \rangle_{\Theta^k} + \rho \langle N(\tilde{z}^{k+1} - \tilde{z}^k) + u^\star - \tilde{u}^{k+1}, M(\tilde{x}^{k+1} - x^\star) \rangle + C\varepsilon_x^k \tag{29}$$
$$\geq \sigma_f \|\tilde{x}^{k+1} - x^\star\|^2 - \langle \nabla f_2(\tilde{x}^{k+1}) - \nabla f_2(\tilde{x}^k), \tilde{x}^{k+1} - x^\star \rangle,$$

2. (Semi-inexact $g$-convexity)

$$\langle \tilde{z}^{k+1} - z^\star, \rho N^T(u^\star - \tilde{u}^{k+1}) + \Psi^k(\tilde{z}^k - \tilde{z}^{k+1}) \rangle \geq -C\sqrt{\varepsilon_z^k}. \tag{30}$$

A detailed proof of Lemma 6 is presented in Appendix B.3. In Lemma 6, we call (29) and (30) *semi-inexact* (strong) convexity because $x^\star$ ($z^\star$) is part of the exact saddle point but $\tilde{x}^{k+1}$ ($\tilde{z}^{k+1}$) is an inexact subproblem solution. With Lemma 6, we establish a descent-type inequality, which takes into inexactness.

**Lemma 7** (Inexact sufficient descent). *Define $\varepsilon_w^k = \varepsilon_x^k + \sqrt{\varepsilon_z^k}$, then the following descent condition holds.*

$$\|\tilde{w}^k - w^\star\|_{G^k}^2 - \|\tilde{w}^{k+1} - w^\star\|_{G^k}^2 + C\varepsilon_w^k \geq \frac{1}{2}\|\tilde{w}^k - \tilde{w}^{k+1}\|_{G^k}^2 + \frac{2\sigma_f}{\rho}\|\tilde{x}^{k+1} - x^\star\|^2 \tag{31}$$

$$- \frac{2}{\rho}\langle \nabla f_2(\tilde{x}^{k+1}) - \nabla f_2(\tilde{x}^k), \tilde{x}^{k+1} - x^\star\rangle.$$

*Proof.* Adding the inequalities (29) and (30) together, and using the relation $M(\tilde{x}^{k+1} - x^\star) = \tilde{u}^{k+1} - \tilde{u}^k + N(z^\star - \tilde{z}^{k+1})$, we reach

$$\langle \tilde{x}^k - \tilde{x}^{k+1}, \tilde{x}^{k+1} - x^\star\rangle_{\Theta^k} + \langle \tilde{z}^k - \tilde{z}^{k+1}, \tilde{z}^{k+1} - z^\star\rangle_{\Psi^k + \rho N^T N} + \rho\langle \tilde{u}^k - \tilde{u}^{k+1}, \tilde{u}^{k+1} - u^\star\rangle + C\varepsilon_x^k$$

$$\geq \sigma_f\|\tilde{x}^{k+1} - x^\star\|^2 - \langle \nabla f_2(\tilde{x}^{k+1}) - \nabla f_2(\tilde{x}^k), \tilde{x}^{k+1} - x^\star\rangle - C\sqrt{\varepsilon_z^k} + \rho\langle \tilde{u}^k - \tilde{u}^{k+1}, N(\tilde{z}^{k+1} - \tilde{z}^k)\rangle$$

Recalling the definitions of $\tilde{w}$, $\varepsilon_w^k$, and $G^k$, and using the identity $\langle a - b, \Upsilon(c - d)\rangle = 1/2\left(\|a - d\|_\Upsilon^2 - \|a - c\|_\Upsilon^2\right) + 1/2\left(\|c - b\|_\Upsilon^2 - \|d - b\|_\Upsilon^2\right)$, we arrive at

$$\|\tilde{w}^k - w^\star\|_{G^k}^2 - \|\tilde{w}^{k+1} - w^\star\|_{G^k}^2 + \varepsilon_w^k \geq \frac{2\sigma_f}{\rho}\|\tilde{x}^{k+1} - x^\star\|^2 + \|\tilde{w}^{k+1} - \tilde{w}^k\|_{G^k}^2$$

$$+ \langle \tilde{u}^k - \tilde{u}^{k+1}, N(\tilde{z}^{k+1} - \tilde{z}^k)\rangle - \frac{2}{\rho}\langle \nabla f_2(\tilde{x}^{k+1}) - \nabla f_2(\tilde{x}^k), \tilde{x}^{k+1} - x^\star\rangle.$$

Now, for the term $\|\tilde{w}^{k+1} - \tilde{w}^k\|_{G^k}^2 + \langle \tilde{u}^k - \tilde{u}^{k+1}, N(\tilde{z}^{k+1} - \tilde{z}^k)\rangle$, Cauchy-Schwarz implies

$$\|\tilde{w}^{k+1} - \tilde{w}^k\|_{G^k}^2 + \langle \tilde{u}^k - \tilde{u}^{k+1}, N(\tilde{z}^{k+1} - \tilde{z}^k)\rangle$$

$$\geq \|\tilde{w}^{k+1} - \tilde{w}^k\|_{G^k}^2 - \frac{1}{2}\|\tilde{u}^{k+1} - \tilde{u}^k\|^2 - \frac{1}{2}\|\tilde{z}^{k+1} - \tilde{z}^k\|_{N^T N}^2$$

$$\geq \|\tilde{w}^{k+1} - \tilde{w}^k\|_{G^k}^2 - \frac{1}{2}\|\tilde{u}^{k+1} - \tilde{u}^k\|^2 - \frac{1}{2}\|\tilde{z}^{k+1} - \tilde{z}^k\|_{1/\rho\Psi^k + N^T N}^2$$

$$= \frac{1}{2}\|\tilde{w}^{k+1} - \tilde{w}^k\|_{G^k}^2.$$

Hence we obtain

$$\|\tilde{w}^k - w^\star\|_{G^k}^2 - \|\tilde{w}^{k+1} - w^\star\|_{G^k}^2 + C\varepsilon_w^k \geq \frac{2\sigma_f}{\rho}\|\tilde{x}^{k+1} - x^\star\|^2 + \frac{1}{2}\|\tilde{w}^{k+1} - \tilde{w}^k\|_{G^k}^2$$

$$- \frac{2}{\rho}\langle \nabla f_2(\tilde{x}^{k+1}) - \nabla f_2(\tilde{x}^k), \tilde{x}^{k+1} - x^\star\rangle,$$

as desired. □ □

Given the inexact sufficient descent condition (31), the next step in proving linear convergence is to show (31) leads to a contraction relation between $\Phi^{k+1}$ and $\Phi^k$.

**Lemma 8** (Inexact Contraction Lemma). *Under the assumptions of Theorem 2, there exists constants $\delta > 0$, and $C \geq 0$ such that*

$$(1 + \delta)\Phi^{k+1} \leq (1 + \zeta^k)\left(\Phi^k + C\varepsilon_w^k\right). \tag{32}$$

The proof of Lemma 8, and an explicit expression for $\delta$, appears in Appendix B.3. As in Theorem 2, the constant $\delta$ gives the rate of linear convergence and depends on the conditioning of $f$ and the constraint matrices $M$ and $N$. The better the conditioning, the faster the convergence, with the opposite holding true as the conditioning worsens. With Lemma 8 in hand, we now prove Theorem 2.

### 6.3 Proof of Theorem 2

*Proof.* By induction on (32), we have

$$(1+\delta)^k \Phi^k \leq \left(\prod_{j=1}^{k}(1+\zeta^j)\right)\Phi^1 + C\sum_{j=1}^{k}\left(\prod_{i=j}^{k}(1+\zeta^i)\right)(1+\delta)^{j-1}\varepsilon_w^j$$

$$\leq \tau_\zeta \Phi^1 + C\tau_\zeta \sum_{j=1}^{k}(1+\delta)^{j-1}\varepsilon_w^j \leq \tau_\zeta \Phi^1 + C\tau_\zeta \sum_{j=1}^{k}\left(\frac{1+\delta}{1+q}\right)^{j-1}$$

$$\leq \tau_\zeta \Phi^1 + \frac{C\tau_\zeta}{1-\frac{1+\delta}{1+q}} = \tau_\zeta \Phi^1 + S,$$

where the second inequality uses Assumption 8 to reach $\varepsilon_w^j \leq \frac{C}{(1+q)^j}$, and the third inequality uses $q > \delta$ to bound the sum by the sum of the geometric series. Hence, we have shown the first claim. The second claim follows immediately from the first via a routine calculation. $\square$

## 7 Applications

This section applies our theory to establish convergence rates for NysADMM and sketch-and-solve ADMM.

### 7.1 Convergence of NysADMM

---
**Algorithm 3** NysADMM

**input:** penalty parameter $\rho$, step-size $\eta$, regularization $\sigma \geq 0$, forcing sequences $\{\varepsilon_x^k\}_{k\geq 1}$, $\{\varepsilon_z^k\}_{k\geq 1}$
  **repeat**
    Find $\varepsilon_x^k$-approximate solution $\tilde{x}^{k+1}$ of

$$(\eta H_f^k + (\rho + \eta\sigma)I)x = \eta(H_f^k + \sigma I)\tilde{x}^k - \nabla f(\tilde{x}^k) + \rho(\tilde{z}^k - \tilde{u}^k)$$

    $\tilde{z}^{k+1} \overset{\varepsilon_z^k}{\approx} \underset{z\in Z}{\text{argmin}}\{g(z) + \frac{\rho}{2}\|M\tilde{x}^{k+1} - z + \tilde{u}^k\|^2\}$
    $\tilde{u}^{k+1} = \tilde{u}^k + M\tilde{x}^{k+1} - \tilde{z}^{k+1}$
  **until** convergence
**output:** solution $(x^\star, z^\star)$ of problem (1)

---

We begin with the NysADMM scheme from Zhao et al. (2022). The defining aspect of NysADMM is that it linearizes $f$ to turn the $x$-subproblem into a linear system. NysADMM solves this system inexactly using the Nyström PCG method from Frangella et al. (2023).

Previously Theorem 4.3 of Zhao et al. (2022), only established convergence of NysADMM in the case where $f$ is quadratic and without providing an explicit rate of convergence. In the discussion in the paragraph below Theorem 4.3, Zhao et al. (2022) left open the question of convergence for general $f$, while conjecturing that strong convexity could ensure a linear convergence rate. In this subsection, we address convergence for non-quadratic $f$ by establishing a $\mathcal{O}(1/t)$ ergodic convergence rate, when $f$ is smooth and convex. This significantly strengthens the result of Zhao et al. (2022), by showing NysADMM enjoys the same rate as classical ADMM in this setting.

Recall NysADMM is obtained from Algorithm 2 by setting $f_1 = 0, f_2 = f$, using the exact Hessian $\Theta^k = H_f(\tilde{x}^k) = \eta(H_f^k + \sigma I)$, and setting $\Psi^k = 0$. Instantiating these selections into Algorithm 2, we obtain NysADMM, presented as Algorithm 3. Compared to the original NysADMM, Algorithm 3 adds a regularization term $\sigma I$ to the Hessian. In theory, when $f$ is only convex, this regularization term is required to ensure the condition $\Theta^k \succeq \theta_{\min}$ of Assumption 2, as the Hessian along the optimization path may fail

to be uniformly bounded below. The addition of the $\sigma I$ term removes this issue. However, empirically this seems to be unnecessary as as Zhao et al. (2022) runs Algorithm 3 on non-strongly convex objectives with $\sigma = 0$, and convergence is still achieved. We observe the same behavior in all our experiments, where we use $\sigma = 0$ throughout, and convergence consistent with our theory is obtained. Hence, in practice, we recommend setting some small value, say $\sigma = 10^{-8}$, so that convergence isn't slowed by unneeded regularization. This already consistent with best practices in ADMM solvers like OSQP (Stellato et al., 2020), which adds a small regularization term to ensure stability.

We obtain the following convergence guarantee by substituting the parameters defining NysADMM (with the added $\sigma I$ term) into Theorem 1.

**Corollary 1** (Convergence of NysADMM). *Instate the assumptions of Theorem 1. Let $\sigma > 0$. Set $f_1 = 0, f_2 = f$, $\eta = \hat{L}_f$, $\Theta^k = \eta(H_f^k + \sigma I)$, and $\Psi^k = 0$ in Algorithm 2. Then*

$$f(\bar{x}^{t+1}) + g(\bar{z}^{t+1}) - p^\star \leq \frac{\Gamma}{t}, \qquad \|M\bar{x}^{t+1} + N\bar{z}^{t+1}\| \leq \frac{2}{t}\sqrt{\frac{\Gamma}{\rho} + d_{u^\star}^2}.$$

*Here $\Gamma$ and $d_{u^\star}^2$ are the same as in Theorem 1.*

NysADMM converges at the same $\mathcal{O}(1/t)$-rate as standard ADMM, despite all the approximations it makes. Thus, NysADMM offers the same level of performance as ADMM, but is much faster due to its use of inexactness. This result is empirically verified in Section 8, where NysADMM converges almost identically to ADMM. Corollary 1 supports the empirical choice of a constant step-size $\eta = 1$, which was shown to have excellent performance uniformly across tasks in Zhao et al. (2022): the theorem sets $\eta = \hat{L}_f$ and $\hat{L}_f = 1$ for quadratic functions, and satisfies $\hat{L}_f = \mathcal{O}(1)$ for loss functions such as the logistic loss. We recommend setting $\eta = 1$ as the default value for GeNI-ADMM. Given NysADMM's superb empirical performance in Zhao et al. (2022) and the firm theoretical grounding given by Corollary 1, we conclude that NysADMM provides a reliable framework for solving large-scale machine learning problems.

## 7.2 Convergence of Sketch-and-solve ADMM

Sketch-and-solve ADMM is obtained from GeNI-ADMM by taking $\Theta^k$ to be an approximation of the Hessian of $f$. We focus on the case when the approximate Hessian is computed by a sketching procedure, though we note that $\Theta^k$ need not be obtained in this manner. The two most popular sketch-and-solve methods are the Newton sketch (Pilanci & Wainwright, 2017; Lacotte et al., 2021; Derezinski et al., 2021) and Nyström sketch-and-solve (Bach, 2013; Alaoui & Mahoney, 2015; Musco & Musco, 2017; Frangella et al., 2023; Chen et al., 2025). We briefly review these ideas, before discussing Sketch-and-solve ADMM in detail.

**Newton sketch** If the Hessian $H_f^k$ can be written in the form $H_f^k = L^T L$ for some $L \in \mathbb{R}^{n \times d}$, which occurs for many important problems (Pilanci & Wainwright, 2017). The Newton sketch uses the approximation:

$$\hat{H}^k = L^T S^T S L, \quad S \in \mathbb{R}^{s \times n},$$

where $S$ is a random sketching matrix. The sketching matrix $S$ could be a Gaussian embedding, a fast structured random embedding, or a row sketching matrix (Pilanci & Wainwright, 2017; Lacotte et al., 2021; Derezinski et al., 2021). If $n \gg d$, then typically $s = \tilde{\mathcal{O}}(d)$ and $\gamma = 0$, otherwise if $n \sim d$, $s \ll d$ is typically used, with $s$ being selected on the order of the effective degrees of freedom of $\hat{H}^k$. Using a structured sketching matrix allows for $\hat{H}^k$ to be constructed and factored in $\tilde{\mathcal{O}}(nd + ds^2)$ time (Pilanci & Wainwright, 2017; Lacotte & Pilanci, 2020). For more details on the Newton Sketch, we refer the reader to Pilanci & Wainwright (2017); Lacotte et al. (2021); Derezinski et al. (2021).

**Nyström sketch-and-solve** Nyström sketch-and-solve (Bach, 2013; Alaoui & Mahoney, 2015; Musco & Musco, 2017; Frangella et al., 2023) uses a randomized Nyström approximation (Gittens & Mahoney, 2016; Tropp et al., 2017) to $H_f^k$,

$$\hat{H}^k = (H_k^f S)(S^T H_k^f S)^\dagger (H_k^f S)^T,$$

where $S \in \mathbb{R}^{d \times s}$ is a random sketching matrix. Similar to the Newton Sketch, $S$ can be a standard Gaussian random matrix, a structured random embedding, or it could correspond to a column selection scheme such as uniform column sampling, leverage score sampling, or adaptive randomized pivoting (Bach, 2013; Alaoui & Mahoney, 2015; Tropp et al., 2017; Musco & Musco, 2017; Chen et al., 2025). The sketch size $s$ is taken to be much smaller than $d$, so that $\hat{H}^k$ is always a low-rank approximation to $H_f^k$. Similar to the Newton Sketch, if $S$ is a structured random embedding, then $\hat{H}^k$ can be constructed in $\tilde{\mathcal{O}}(d^2 + ds^2)$ (Tropp et al., 2017). For details on Nyström sketch-and-solve and its implementation we refer the reader to the references Bach (2013); Alaoui & Mahoney (2015); Musco & Musco (2017); Frangella et al. (2023); Chen et al. (2025).

### 7.3 Sketch-and-solve ADMM algorithm

We now present the Sketch-and-solve ADMM algorithm. To the best of our knowledge, this is the first time using a sketch-and-solve approach with ADMM has been formally developed and analyzed.

Our consideration of this scheme is motivated by the recent survey Buluc et al. (2022), which discusses in Section 2.3 the possibility for randomized algorithms to accelerate the solution of large-scale inverse problems. Specifically, (Buluc et al., 2022, pg.13) proposes using sketching to approximate expensive linear system solves in the ADMM subproblem associated with the data fidelity term. The idea is natural given the effectiveness of algorithms that combine sketching with Newton-type methods (Pilanci & Wainwright, 2017; Roosta-Khorasani & Mahoney, 2019; Na et al., 2023; Frangella et al., 2024; Jiang et al., 2024). However, no theoretical or empirical support for combining ADMM with sketching is presented. This omission is significant, as even in the simple case of a quadratic smooth term, so that no linearization of $f$ in the $x$-subproblem is needed, standard ADMM theory does not guarantee convergence. Indeed, classic results like those of Eckstein & Bertsekas (1992) assume the subproblems are solved to progressively better accuracy, an assumption that is not satisfied when solving the subproblem via sketching. Convergence is even less obvious when $f$ is non-quadratic, as linearizing $f$ is required to reduce the $x$-subproblem to solving a linear system.

In this section, we resolve this problem by providing the first convergence guarantees for sketch-and-solve ADMM, extending classical ADMM theory to accommodate inexact sketched solves.

---

**Algorithm 4** Sketch-and-solve ADMM

---

**input:** penalty parameter $\rho$, step-size $\eta$, $\{\varepsilon_z^k\}_{k \geq 1}$
  **repeat**
    Construct approximation $\hat{H}_k$ of $H_k$.
    Find solution $\tilde{x}^{k+1}$ of

$$(\eta \hat{H}^k + (\rho + \eta\gamma^k)I)x = \eta \left(\hat{H}^k + \gamma^k I\right)\tilde{x}^k - \nabla f(\tilde{x}^k) + \rho(\tilde{z}^k - \tilde{u}^k)$$

$$\tilde{z}^{k+1} \stackrel{\varepsilon_z^k}{\approx} \underset{z \in Z}{\text{argmin}}\{g(z) + \tfrac{\rho}{2}\|M\tilde{x}^{k+1} - z + \tilde{u}^k\|^2\}$$
    $\tilde{u}^{k+1} = \tilde{u}^k + M\tilde{x}^{k+1} - \tilde{z}^{k+1}$
  **until** convergence
**output:** solution $(x^\star, z^\star)$ of problem (1)

---

Sketch-and-solve ADMM presented in Algorithm 4 is obtained from Algorithm 2 by setting

$$\Theta^k = \eta \left(\hat{H}^k + \gamma^k I\right), \tag{33}$$

where $\hat{H}^k$ is an approximation to the Hessian $H_f(\tilde{x}^k)$ at the $k$th iteration, and $\gamma^k \geq 0$ is a constant chosen to ensure convergence. The term $\gamma^k I$ ensures that the approximate linearization satisfies the $\Theta$-relative smoothness condition when $\gamma^k$ is chosen appropriately, as in the following lemma:

**Lemma 9.** *Suppose $f$ is $\hat{L}_f$-relatively smooth with respect to its Hessian $H_f$. Construct $\{\Theta^k\}_{k \geq 1}$ as in (33) with $\eta = 1$. is such that one of the following conditions is satisfied:*

*1. Suppose that $\gamma^k > 0$ satisfies $\gamma^k \geq \|E^k\| = \|H_f(\tilde{x}^k) - \hat{H}^k\|$ for every $k$. Then*

$$f(x) \leq f(\tilde{x}^k) + \langle \nabla f(\tilde{x}^k), x - \tilde{x}^k \rangle + \frac{\hat{L}_f}{2}\|x - \tilde{x}^k\|_{\Theta^k}^2.$$

*2. For all $k > 0$ there exists $0 < \tau < 1$, such that*

$$(1 - \tau)(H_f^k + \gamma^k I) \preceq \hat{H}^k + \gamma^k I \preceq (1 + \tau)(H_f^k + \gamma^k I).$$

*Then*

$$f(x) \leq f(\tilde{x}^k) + \langle \nabla f(\tilde{x}^k), x - \tilde{x}^k \rangle + \frac{\hat{L}_f/(1 - \tau)}{2}\|x - \tilde{x}^k\|_{\Theta^k}^2.$$

For the proof of Lemma 9, please see Appendix C.

Lemma 9 shows we can ensure relative smoothness by selecting $\gamma^k > 0$ appropriately. Assuming relative smoothness, we may invoke Theorem 1 to guarantee that sketch-and-solve ADMM converges. Unlike with NysADMM, we find it is necessary to select $\gamma^k$ carefully (such as in Lemma 9) to ensure the relative smoothness condition holds, otherwise sketch-and-solve ADMM will diverge, see Section 8.3 for numerical demonstration. The need for this condition is different from prior sketching schemes in optimization, such as the Newton Sketch (Pilanci & Wainwright, 2017; Lacotte et al., 2021), where convergence is guaranteed as long as the Hessian approximation is invertible.

## 7.4 Quantifying $\gamma^k$ required for convergence

Lemma 9 shows that $\gamma^k$ must be set appropriately to achieve convergence. We now discuss how results from the sketching literature allow us to quantify this parameter. We focus on item 2 of Lemma 9, as it encompasses both the Newton Sketch and Nyström sketch-and-solve. Item 1 is only of interest in the case of Nyström sketch-and-solve, as the Newton Sketch provides an unbiased approximation of $H_f^k$. Consequently, it exhibits Monte Carlo-style errors that are independent of the eigenvalue decay of $H_k^f$ (Tropp, 2015), hence $\|E^k\|$ can be quite large. In contrast, error guarantees for the Nyström approximation depend directly upon the spectrum of $H_k^f$ (Gittens & Mahoney, 2016; Tropp et al., 2017; Frangella et al., 2023), so that $\|E^k\|$ is small when $H_k^f$ is approximately low-rank.

Now, if $\hat{H}_k$ is a randomized Nyström approximation or a Newton Sketch constructed via leverage score sampling or random embeddings with a sketch size of $s = \tilde{\mathcal{O}}(r)$, with $r < d$, then the following is known to hold with high probability (Musco & Musco, 2017; Dereziński et al., 2025; Dereziński & Sidford, 2025):

$$\frac{1}{2}\left(H_f^k + \frac{1}{r}\sum_{j>r}\lambda_j(H_f^k)I\right) \preceq \left(\hat{H}^k + \frac{1}{r}\sum_{j>r}\lambda_j(H_f^k)I\right) \preceq 2\left(H_f^k + \frac{1}{r}\sum_{j>r}\lambda_j(H_f^k)I\right).$$

Thus, item 2 of Lemma 9 holds with $\tau = 1/2$ and $\gamma^k = \frac{1}{r}\sum_{j>r}\lambda_j(H_f^k)$. Hence, when $H_f^k$ is approximately low-rank, only small or moderate damping is needed to ensure convergence. However, if $H_f^k$ has a slowly decaying spectrum, larger regularization is required to ensure convergence. As we shall see below, the size of the damping directly impacts the convergence rate.

## 7.5 Sketch-and-solve ADMM convergence guarantee

We now formally state our convergence guarantee for Sketch-and-solve ADMM.

**Corollary 2.** *Suppose $f$ is $\hat{L}_f$-relatively smooth with respect to its Hessian $H_f$ and instate the assumptions of Theorem 1. In Algorithm 2, set $\Theta^k = \eta(\hat{H}^k + \gamma^k I)$ with $\eta = 1$, $\Psi^k = 0$, and suppose $\gamma^k$ satisfies the*

*conditions of item 2 in Lemma 9. Then*

$$f(\bar{x}^{t+1}) + g(\bar{z}^{t+1}) - p^\star \leq \frac{1}{t} \left( \frac{\hat{L}_f}{1-\tau} \left( d^2_{x^\star, \hat{H}^1} + \gamma^1 d^2_{x^\star} \right) + \frac{\rho}{2} d^2_{z^\star} + C_x \mathcal{E}_x + C_z \mathcal{E}_z + C_\zeta \mathcal{E}_\zeta \right) =: \frac{1}{t} \hat{\Gamma}$$

*and*

$$\|M\bar{x}^{t+1} - \bar{z}^{t+1}\| \leq \frac{2}{t} \sqrt{\frac{\hat{\Gamma}}{\rho} + d^2_{u^\star}}.$$

*Here variables $\bar{x}^{t+1}$ and $\bar{z}^{t+1}$, diameters $d^2_{x^\star, \hat{H}^1}$, $d^2_{z^\star}$, and $d^2_{u^\star}$, and constants $C_x$, $C_z$, $\mathcal{E}_x$, $\mathcal{E}_z$, $\mathcal{E}_\zeta$, and $p^\star$ are all defined as in Theorem 1.*

Corollary 2 shows sketch-and-solve ADMM obtains an $\mathcal{O}(1/t)$-convergence rate. Compared to NysADMM, $\hat{L}_f$ has been inflated by $1/(1-\tau)$ and a new error term $\gamma^1 d^2_{x^\star}$ appears, both of which stem from using an approximate Hessian $\hat{H}^k$. In particular, the $\gamma^1$ term introduces a tradeoff. $\hat{H}^k$ can be constructed with a small sketch size $s$ to enable very fast inversion, but as Section 7.4 shows, $\gamma^1$ may have to be quite large to ensure convergence, especially when the spectrum of $H_f^k$ decays slowly. Corollary 2 shows that larger $\gamma^1$ implies Sketch-and-solve ADMM suffers from a slower convergence rate relative to using the exact Hessian as in NysADMM. Conversely, $\gamma^1$ can be made small by using a large sketch size, but at the cost of more expensive iterations. Given this tradeoff, Sketch-and-solve ADMM is expected to be most useful in two scenarios: 1) problems that are highly over-determined, so that the Newton Sketch can be applied with sketch size $s = \tilde{\mathcal{O}}(d)$ and $\gamma^k = 0$, or 2) problems for which $H_f^k$ is approximately low-rank so that Nyström sketch-and-solve or Newton Sketch can be applied with a sketch size much smaller than $d$ with a small value of $\gamma^k$.

The preceding considerations lead to an important observation: Sketch-and-solve ADMM can be regarded as a compromise between NysADMM ($\Theta^k = \eta(H_f(\tilde{x}^k) + \sigma I)$) and gradient descent ADMM ($\Theta^k = \eta I$), with the convergence rate improving as the accuracy of the Hessian approximation increases. In particular, if $\hat{H}^k$ provides a poor approximation (i.e., when $\gamma^k$ is large), Sketch-and-solve ADMM may not perform much better than gradient descent ADMM. Conversely, when $\hat{H}_f^k$ is a good approximation to $H_f^k$, the method can approach the performance of NysADMM while remaining computationally efficient. This hybrid behavior predicted by Corollary 2 is confirmed in our experiments in Section 8.

## 8 Numerical experiments

In this section, we numerically illustrate the convergence results developed in Section 5 for several methods highlighted in Section 3.1.1 that fit into the GeNI-ADMM framework: sketch-and-solve ADMM (Algorithm 4), NysADMM (Algorithm 3), and "gradient descent" ADMM (GD-ADMM) (11). As a baseline, we also compare to exact ADMM (Algorithm 1) to see how various approximations or inexactness impact the convergence rate. We conduct three sets of experiments that verify different aspects of the theory:

- Section 8.1 verifies that NysADMM, GD-ADMM and sketch-and-solve ADMM converge sublinearly for convex problems. Moreover, we observe a fast transition to linear convergence, after which all methods but GD-ADMM converge quickly to high accuracy.

- Section 8.2 verifies that NysADMM, GD-ADMM, and sketch-and-solve ADMM converge linearly for strongly convex problems.

- Section 8.3 verifies that, without the correction term, sketch-and-solve ADMM diverges, showing the necessity of the correction term in Section 7.2.

We consider three common problems in machine learning and statistics in our experiments: lasso (Tibshirani, 1996), elastic net regression (Zou & Hastie, 2005), and $\ell_1$-logistic regression (Hastie et al., 2015). All experiments use the `realsim` dataset from LIBSVM (Chang & Lin, 2011), accessed through OpenML (Vanschoren et al., 2013), which has $72,309$ samples and $20,958$ features. Our experiments use a subsample of

`realsim`, consisting of $10,000$ random samples, which ensures the objective is not strongly convex for lasso and $\ell_1$-logistic regression.

All methods solve the $z$-subproblem exactly, and every method, except NysADMM, solves their $x$-subproblem exactly. NysADMM solves the linear system in Algorithm 3 inexactly using Nyström PCG. At each iteration, the following absolute tolerance on the PCG residual is used for determining the termination of Nyström PCG:

$$\varepsilon_x^k = \min\left\{\frac{\sqrt{r_p^{k-1}r_d^{k-1}}}{k^{1.5}}, 1\right\},$$

where $r_p^{k-1}$ and $r_d^{k-1}$ are the ADMM primal and dual residuals at the previous iteration. We multiply a summable sequence by the geometric mean of the primal and dual residuals at the previous iteration. The geometric mean of the primal and dual residuals is a commonly used tolerance for PCG in practical ADMM implementations such as OSQP (Schubiger et al., 2020)—despite this sequence not being apriori summable.

For NysADMM, a sketch size of 50 is used to construct the Nyström preconditioner, and for sketch-and-solve ADMM, we use a sketch size 500 to form the Hessian approximation. For sketch-and-solve ADMM, the parameter $\gamma^k$ in (33) is chosen by estimating the error of the Nyström sketch using power iteration.

All experiments are performed in the Julia programming language (Bezanson et al., 2017) on a MacBook Pro with a M1 Max processor and 64GB of RAM. To compute the "true" optimal values, we use the commercial solver Mosek (ApS, 2022) (with tolerances set low for high accuracy and presolve turned off to preserve the problem scaling) and the modeling language JuMP (Dunning et al., 2017; Legat et al., 2021). The code to reproduce all experiments may be found in the anonymous repository:

https://github.com/tjdiamandis/GeNIADMM.jl

### 8.1 GeNI-ADMM converges sublinearly and locally linearly on convex problems

To illustrate the global sublinear convergence of GeNI-ADMM methods on convex objectives, we look at the performance of ADMM, NysADMM, GD-ADMM, and sketch-and-solve ADMM on solving a lasso and $\ell_1$-logistic regression problem with the `realsim` dataset. Note, as the number of samples is smaller than the number of features, the corresponding optimization problems are convex but not strongly convex.

**Lasso regression** The lasso regression problem is to minimize the $\ell_2$ error of a linear model with an $\ell_1$ penalty on the weights:

$$\text{minimize} \quad (1/2)\|Ax - b\|_2^2 + \gamma\|x\|_1.$$

This can be easily transformed into the form (1) by taking $f(x) = (1/2)\|Ax - b\|_2^2$, $g(z) = \gamma\|z\|_1$, $M = I$, $N = -I$, and $X = Z = \mathbb{R}^n$. We set $\gamma = 0.05 \cdot \gamma_{\max}$, where $\gamma_{\max} = \|A^T b\|_\infty$ is the value above which the all zeros vector is optimal Hastie et al. (2015). We stop the algorithm when the gap is less than $10^{-4}$, or after 500 iterations.

The results of lasso regression are illustrated in Figure 1. Figure 1 shows all methods initially converge at sublinear rate, but quickly transition to a linear rate of convergence after reaching some manifold containing the solution. ADMM, NysADMM, and sketch-and-solve ADMM (which use curvature information) all converge converge much faster then GD-ADMM, confirming the predictions of Section 5 that methods which use curvature information will converge faster than methods that do not. Moreover, the difference in convergence between NysADMM and ADMM is negligible, despite the former having a much cheaper iteration complexity due to the use of inexact linear system solves.

**L1-Logistic Regression** We set $\gamma = 0.05 \cdot \gamma_{\max}$, where $\gamma_{\max} = (1/2)\|A^T \mathbf{1}\|_\infty$ is the value above which the all zeros vector is optimal. For NysADMM, the preconditioner is re-constructed after every 20 iterations. For sketch-and-solve ADMM, we re-construct the approximate Hessian at every iteration. Since the $x$-subproblem is not a quadratic program, we use the L-BFGS (Liu & Nocedal, 1989) implementation from the `Optim.jl` package (Mogensen & Riseth, 2018) to solve the $x$-subproblem in exact ADMM. We use the

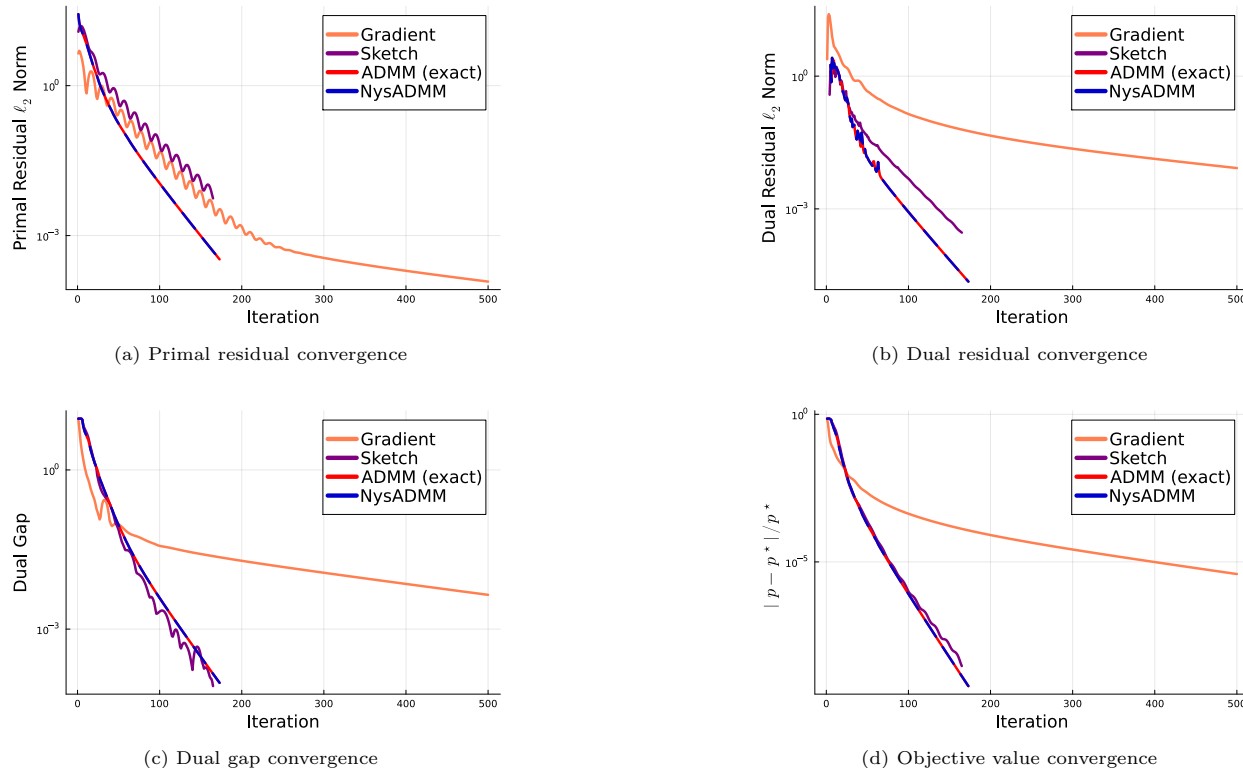

Figure 1: Convergence of lasso regression for NysADMM, sketch-and-solve ADMM, and gradient descent ADMM.

default hyperparamter settings in Optim.jl for L-BFGS. In particular, the $x$-subproblem is considered solved, if the infinity norm of the gradient of the $x$-subproblem satisfies: $\|g\|_\infty \leq 10^{-8}$.

Figure 2 presents the results for logistic regression. The results are consistent with the lasso experiment—all methods initially converge sublinearly before quickly transitioning to linear convergence, and methods using better curvature information, converge faster. In particular, although sketch-and-solve-ADMM converges slightly faster than GD-ADMM, its convergence is much slower than NysADMM, which more accurately captures the curvature due to using the exact Hessian. The convergence of NysADMM and ADMM is essentially identical, despite the former having a much cheaper iteration cost due to approximating the $x$-subproblem and using inexact linear system solves.

## 8.2 GeNI-ADMM converges linearly on strongly convex problems

To verify the linear convergence of GeNI-ADMM methods in the presence of strong convexity, we experiment with the elastic net problem:

$$\text{minimize} \quad (1/2)\|Ax - b\|_2^2 + \gamma\|x\|_1 + (\mu/2)\|x\|_2^2.$$

We set $\mu = 1$, and use the same problem data and value of $\gamma$ as the lasso experiment. The results of the elastic-net experiment are presented in Figure 3. Comparing Figures 1 and 3, we clearly observe the linear convergence guaranteed by the theory in Section 6. Although in Figure 1, ADMM and NysADMM quickly exhibit linear convergence, Figure 3 clearly shows strong convexity leads to an improvement in the number of iterations required to converge. Moreover, we see methods that make better use of curvature information converge faster than methods that do not, consistent with the results of the lasso and logistic regression experiments.

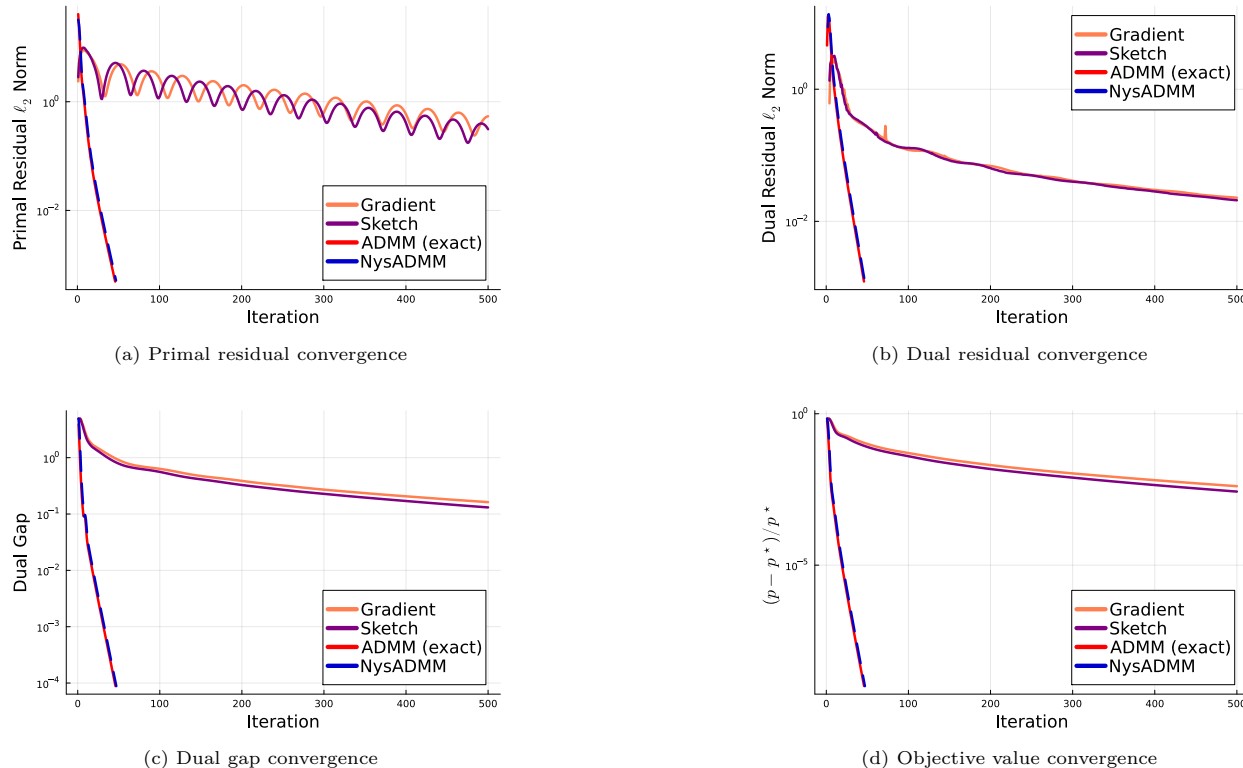

(a) Primal residual convergence

(b) Dual residual convergence

(c) Dual gap convergence

(d) Objective value convergence

Figure 2: Convergence of logistic regression for NysADMM, sketch-and-solve ADMM, and gradient descent ADMM.

### 8.3 Sketch-and-solve ADMM fails to converge without the correction term

To demonstrate the necessity of the correction term in Section 7.2, we run sketch-and-solve ADMM on lasso and $\ell_1$-logistic regression without the correction term. Figure 4 presents the results of these simulations. Without the correction term in (33), sketch-and-solve ADMM quickly diverges on the lasso problem. For the logistic regression problem, it oscillates and fails to converge. These results highlight the importance of selecting this term appropriately as discussed in Section 7.2.

## 9 Conclusion

In this paper, we have developed a framework, GeNI-ADMM, that facilitates efficient theoretical analysis of approximate ADMM schemes and can aid in the design of new, practical approximate ADMM methods. GeNI-ADMM encompasses prior approximate ADMM schemes as special cases by allowing various approximations to the $x$ and $z$ subproblems, which can be solved inexactly. We have established the usual ergodic $\mathcal{O}(1/t)$ convergence rate for GeNI-ADMM under standard hypotheses, and linear convergence under strong convexity. We have shown how to derive explicit convergence rates for ADMM variants that exploit randomized numerical linear algebra using the GeNI-ADMM framework. Specifically, we have established a general convergence result with an explicit rate for NysADMM from Zhao et al. (2022), which previously was only known to enjoy rate-free convergence for quadratic $f$. We also formally introduce sketch-and-solve ADMM, which was proposed informally in Buluc et al. (2022) without an explicit algorithm and without theoretical or empirical support. Our convergence analysis shows that sketch-and-solve ADMM enjoys the standard $\mathcal{O}(1/t)$ rate, and also quantifies how a better (or worse) approximation to the Hessian improves (or worsens) the convergence rate. Numerical experiments on real-world data generally show an initial sublinear phase followed by linear convergence, validating the theory we developed.

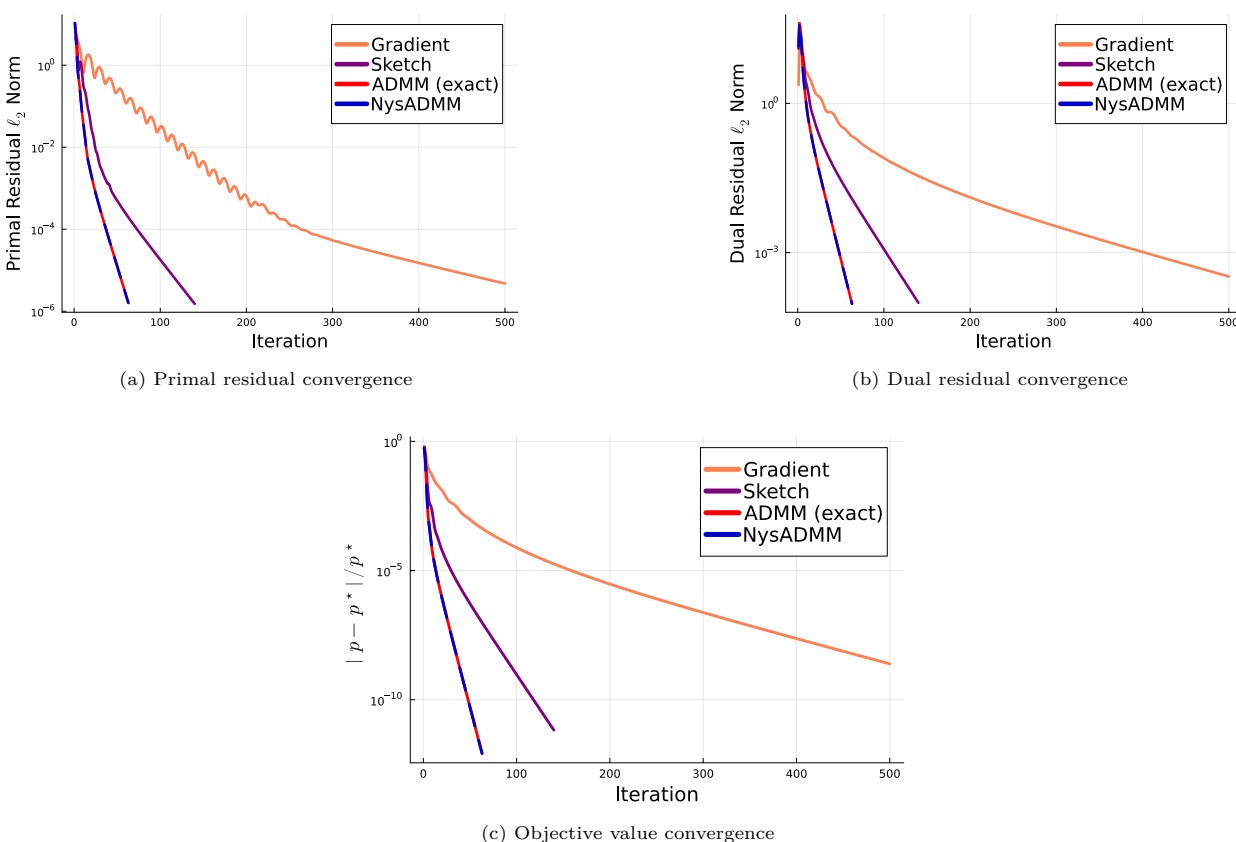

(a) Primal residual convergence

(b) Dual residual convergence

(c) Objective value convergence

Figure 3: Convergence of elastic net regression for NysADMM, sketch-and-solve ADMM, and gradient descent ADMM.

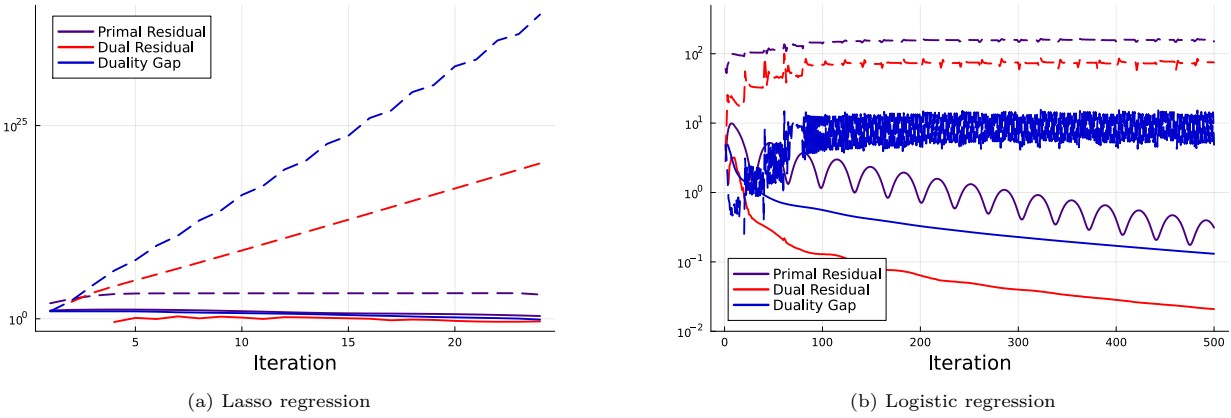

(a) Lasso regression

(b) Logistic regression

Figure 4: We show the convergence or lack thereof for sketch-and-solve ADMM with (solid lines) and without (dashed lines) the correction term required by the theoretical results (see section 7.2). When this term is not included, the algorithm does not converge.

# Acknowledgments

The authors thank Liubomir Baicev for providing helpful comments. We also thank the AE Ahmet Alacaoglu and the anonymous reviewers for their careful reading of the manuscript and insightful comments, which helped to improve the presentation of the paper.

Z. Frangella, and M. Udell gratefully acknowledge support from the National Science Foundation (NSF) Award IIS-2233762, the Office of Naval Research (ONR) Award N000142212825 and N000142312203, and the Alfred P. Sloan Foundation. T. Diamandis is supported by the Department of Defense (DoD) through the National Defense Science & Engineering Graduate (NDSEG) Fellowship Program. B. Stellato is supported by the NSF CAREER Award ECCS-2239771.

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

# A  Appendix

# B  Proofs not appearing in the main paper

## B.1  Proofs for Section 4

We begin with the following lemma, which plays a key role in the proof of Lemma 1. For a proof, see Example 1.2.2. in Chapter XI of Hiriart-Urruty & Lemaréchal (1993).

**Lemma 10** ($\varepsilon$-subdifferential of a quadratic function). *Let $h(z) = \langle b, z \rangle + \frac{1}{2}\|z\|_A^2$, where $A$ is a symmetric positive definite linear operator. Then*

$$\partial_\varepsilon h(z) = \left\{ b + A(z + v) \,\middle|\, \frac{\|v\|_A^2}{2} \le \varepsilon \right\}.$$

With Lemma 10 in hand, we now prove Lemma 1.

**Proof of Lemma 1**

*Proof.* Observe the function defining the $z$-subproblem may be decomposed as $G(z) = g(z) + h(z)$ with

$$h(z) = \frac{\rho}{2}\|M\tilde{x}^{k+1} + Nz + \tilde{u}^k\|_2^2 + \frac{1}{2}\|z - \tilde{z}^k\|_{\Psi^k}^2.$$

Now, by hypothesis $\tilde{z}^{k+1}$ gives an $\varepsilon_z^k$-minimum of $G(z)$. Hence by Proposition 1,

$$0 \in \partial_{\varepsilon_z^k} G(\tilde{z}^{k+1}) \text{ and } \partial_{\varepsilon_z^k} G(\tilde{z}^{k+1}) \subset \partial_{\varepsilon_z^k} g(\tilde{z}^{k+1}) + \partial_{\varepsilon_z^k} h(\tilde{z}^{k+1}).$$

Thus, we have $0 = s_g + s_h$ where $s \in \partial_{\varepsilon_z^k} g(\tilde{z}^{k+1})$ and $s_h \in \partial_{\varepsilon_z^k} h(\tilde{z}^{k+1})$. Applying Lemma 10, with $A = \Psi^k + \rho N^T N$ and $b = \rho N^T(M\tilde{x}^{k+1} + \tilde{u}^k) - \Psi^k \tilde{z}^k$, we reach

$$s_h = \rho N^T(M\tilde{x}^{k+1} + N\tilde{z}^{k+1} + \tilde{u}^k) + \Psi^k(\tilde{z}^{k+1} - \tilde{z}^k) + v.$$

The desired claim now immediately follows from using $s = -s_h$. □

### B.2 Proofs for Section 5

**Proof of Proposition 2**

*Proof.* Using the eigendecomposition, we may decompose $H_f$ as

$$H_f = \lambda_1(H_f)\mathbf{v}\mathbf{v}^T + \sum_{i=2}^{d}\lambda_i(H_f)\mathbf{v}_i\mathbf{v}_i^T.$$

So

$$\|x^\star\|_{H_f}^2 = \lambda_1(H_f)\langle\mathbf{v}, x^\star\rangle^2 + \sum_{i=2}^{d}\lambda_i(H_f)\langle\mathbf{v}_i, x_\star\rangle^2$$

$$\leq \lambda_1(H_f)\mu_\mathbf{v}\|x^\star\|^2 + \lambda_2(H_f)\sum_{i=2}^{d}\langle\mathbf{v}_i, x^\star\rangle^2$$

$$\leq \left(\mu_\mathbf{v} + \frac{\lambda_2(H_f)}{\lambda_1(H_f)}\right)\lambda_1(H_f)\|x^\star\|^2.$$

Recalling that $\sigma = \tau\lambda_1(H_f)$ with $\tau \in (0,1)$, we may put everything together to conclude

$$\frac{(\lambda_1(H_f) - \sigma)\|x^\star\|^2}{\|x^\star\|_{H_f}^2} \geq (1-\tau)\frac{\lambda_1(H_f)\|x^\star\|^2}{\|x^\star\|_{H_f}^2} \geq (1-\tau)\left(\mu_\mathbf{v} + \frac{\lambda_2(H_f)}{\lambda_1(H_f)}\right)^{-1},$$

proving the first claim. The second claim follows immediately from the first. $\qquad\square$

**Proof of Proposition 3** To prove Proposition 3, we require the following lemma, which is the analogue of Lemma 5 when the subproblems in Algorithm 2 at iteration $k+1$ are solved *exactly*.

**Lemma 11** (Exact Gap function 1-step bound). *Let $w^{k+1} = (x^{k+1}, z^{k+1}, u^{k+1})$ denote the iterates generated by Algorithm 2 at iteration $k+1$, when we solve the subproblems exactly. Set $w = (x^\star, z^\star, u)$, then the gap function $Q$ satisfies*

$$Q(w, w^{k+1}) \leq \frac{1}{2}\left(\|z - \tilde{x}^k\|_{\Theta^k}^2 - \|x - x^{k+1}\|_{\Theta^k}^2\right) + \frac{1}{2}\left(\|z - \tilde{z}^k\|_{\Psi^k+\rho N^T N}^2 - \|z - z^{k+1}\|_{\Psi^k+\rho N^T N}^2\right)$$
$$\frac{\rho}{2}\left(\|\tilde{z}^k - Mx\|^2 - \|z^{k+1} - Mx\|^2\right) + \frac{\rho}{2}\left(\|\tilde{u}^k - u\|^2 - \|u^{k+1} - u\|^2\right).$$

*Proof.* As $x^{k+1}$ and $z^{k+1}$ are the exact solutions to the $x$ and $z$ subproblems at iteration $k+1$, they satisfy the subproblem optimality conditions exactly. Thus, the following inequalities hold

$$\langle\nabla f_1(x^{k+1}) + \nabla f_2(\tilde{x}^k), x^{k+1} - x^\star\rangle \leq \langle\Theta^k(x^{k+1} - \tilde{x}^k), x^\star - x^{k+1}\rangle + \rho\langle Mx^{k+1} + N\tilde{z}^k + \tilde{u}^k, M(x^\star - x^{k+1})\rangle,$$
$$g(z^\star) - g(z^{k+1}) \geq \langle-\rho N^T u^{k+1} + \Psi^k(\tilde{z}^k - z^{k+1}), z^\star - z^{k+1}\rangle.$$

These are the same inequalities in Lemma 3 and Lemma 4, only $\tilde{x}^{k+1}$ and $\tilde{z}^{k+1}$ have been replaced by the exact solution $x^{k+1}$ and $z^{k+1}$, so that $\varepsilon_x^k = 0$ and $\varepsilon_z^k = 0$.

Observe the desired inequality is exactly the inequality in the conclusion of Lemma 5. The only changes are that there are no error terms present as $\tilde{x}^{k+1}$, $\tilde{z}^{k+1}$, and $\tilde{u}^{k+1}$ have been replaced with $x^{k+1}$, $z^{k+1}$, and $u^{k+1}$. As Lemma 3 and Lemma 4 are what were used to establish Lemma 5, we can apply the exact same argument used there using the exact optimality conditions, so that there are no error terms to conclude

$$Q(w, w^{k+1}) \leq \frac{1}{2}\left(\|z - \tilde{x}^k\|_{\Theta^k}^2 - \|x - x^{k+1}\|_{\Theta^k}^2\right) + \frac{1}{2}\left(\|z - \tilde{z}^k\|_{\Psi^k+\rho N^T N}^2 - \|z - z^{k+1}\|_{\Psi^k+\rho N^T N}^2\right)$$
$$\frac{\rho}{2}\left(\|\tilde{z}^k - Mx\|^2 - \|z^{k+1} - Mx\|^2\right) + \frac{\rho}{2}\left(\|\tilde{u}^k - u\|^2 - \|u^{k+1} - u\|^2\right).$$

$\qquad\square$

Having established Lemma 11, we now prove Proposition 3.

*Proof.* First, observe that item 2. is an immediate consequence of item 1, so it suffices to show item 1. To this end, plugging in $w = w^\star$ into the inequality of Lemma 11, using $Q(w^\star, w^{k+1}) \geq 0$, and rearranging, we reach

$$\|x^{k+1} - x^\star\|_{\Theta^k}^2 + \|z^{k+1} - z^\star\|_{\Psi^k + \rho N^T N}^2 + \rho\|u^{k+1} - u^\star\|^2$$
$$\leq \|\tilde{x}^k - x^\star\|_{\Theta^k}^2 + \|\tilde{z}^k - z^\star\|_{\Psi^k + \rho N^T N}^2 + \rho\|\tilde{u}^k - u^\star\|^2.$$

Defining the norm $\|w\|_{W,k} = \sqrt{\|x\|_{\Theta^k}^2 + \|z\|_{\Psi^k + \rho N^T N}^2 + \rho\|u\|^2}$, the preceding inequality may be rewritten as

$$\|w^{k+1} - w^\star\|_{W,k} \leq \|\tilde{w}^k - w^\star\|_{W,k}.$$

Now, our inexactness hypothesis (Assumption 3) along with $\nu$-strong convexity of the $z$-subproblem (which follows by Assumption 2) implies

$$\|\tilde{x}^{k+1} - x^{k+1}\| \leq \varepsilon_x^k, \quad \|\tilde{z}^{k+1} - z^{k+1}\| \leq \sqrt{\frac{\varepsilon_z^k}{\nu}}.$$

Using $\tilde{u}^{k+1} = M\tilde{x}^{k+1} + N\tilde{z}^{k+1} + \tilde{u}^k$ and $u^{k+1} = Mx^{k+1} + Nz^{k+1} + \tilde{u}^k$, we find

$$\rho\|\tilde{u}^{k+1} - u^{k+1}\| \leq \varepsilon_x^k + C\sqrt{\varepsilon_z^k},$$

where $C$ is some constant. Hence we have,

$$\|\tilde{w}^{k+1} - w^\star\|_{W,k} \leq \|w^{k+1} - w^\star\|_{W,k} + \|\tilde{w}^{k+1} - w^{k+1}\|_{W,k} \leq \|\tilde{w}^k - w^\star\|_{W,k} + C\left(\varepsilon_x^k + \sqrt{\varepsilon_z^k}\right).$$

Defining the summable sequence $\varepsilon_w^k = C(\varepsilon_x^k + \sqrt{\varepsilon_z^k})$, the preceding display may be written as

$$\|\tilde{w}^{k+1} - w^\star\|_{W,k} \leq \|\tilde{w}^k - w^\star\|_{W,k} + \varepsilon_w^k \quad (\triangle),$$

Now, Assumption 4 implies that $\|\tilde{w}^k - w^\star\|_{W,k} \leq \sqrt{1 + \zeta^k}\|\tilde{w}^k - w^\star\|_{W,k-1}$. Combining this inequality with induction on $(\triangle)$, we find

$$\|\tilde{w}^{k+1} - w^\star\|_{W,k} \leq \left(\prod_{j=2}^k \sqrt{1 + \zeta^j}\right)\|\tilde{w}^1 - w^\star\|_{W,1} + \left(\prod_{j=2}^k \sqrt{1 + \zeta^j}\right)\sum_{j=1}^k \varepsilon_w^j.$$

Hence,
$$\sqrt{\min\{\theta_{\min}, \nu\}}\|\tilde{w}^{k+1} - w^\star\| \leq \|\tilde{w}^{k+1} - w^\star\|_{W,k} \leq \sqrt{\tau_\zeta}\left(\|\tilde{w}^1 - w^\star\|_{W,1} + \mathcal{E}_w\right).$$

It follows immediately that:

$$\sup_{k \geq 1}\|\tilde{w}^{k+1} - w^\star\| \leq \max\left\{\frac{\sqrt{\tau_\zeta}\left(\|\tilde{w}^1 - w^\star\|_{W,1} + \mathcal{E}_w\right)}{\sqrt{\min\{\theta_{\min}, \nu\}}}, \|\tilde{w}^0 - w^\star\|\right\} < \infty,$$

and so the sequence $\{\tilde{w}^k\}_{k \geq 0}$ is bounded, which in turn implies $\{\tilde{x}^k\}_{k \geq 0}, \{\tilde{z}^k\}_{k \geq 0}$, and $\{\tilde{u}^k\}_{k \geq 0}$ are bounded. $\square$

## Proof of Lemma 3

*Proof.* Throughout the proof, we shall denote by $S_x^k(x)$, the function defining the $x$-subproblem at iteration $k$. The exact solution of the $x$-subproblem shall be denoted by $x^{k+1}$. To begin, observe that:

$$\nabla S_x^k(x) = \nabla f_1(x) + \nabla f_2(\tilde{x}^k) + (\Theta^k + \rho M^T M)(x - \tilde{x}^k) + \rho M^T(M\tilde{x}^k + N\tilde{z}^k + \tilde{u}^k).$$

Now,

$$
\begin{aligned}
\langle \nabla S_x^k(\tilde{x}^{k+1}), x^\star - \tilde{x}^{k+1} \rangle &= \langle \nabla S_x^k(\tilde{x}^{k+1}) - \nabla S_x^k(x^{k+1}) + \nabla S_x^k(x^{k+1}), x^\star - x^{k+1} + x^{k+1} - \tilde{x}^{k+1} \rangle \\
&= \langle \nabla S_x^k(\tilde{x}^{k+1}) - \nabla S_x^k(x^{k+1}), x^\star - x^{k+1} \rangle \\
&\quad + \langle \nabla S_x^k(\tilde{x}^{k+1}) - \nabla S_x^k(x^{k+1}), x^{k+1} - \tilde{x}^{k+1} \rangle \\
&\quad + \langle \nabla S_x^k(x^{k+1}), x^\star - x^{k+1} \rangle + \langle \nabla S_x^k(x^{k+1}), x^{k+1} - \tilde{x}^{k+1} \rangle \\
&\overset{(1)}{\geq} \langle \nabla f_1(\tilde{x}^{k+1}) - \nabla f_1(x^{k+1}), x^\star - \tilde{x}^{k+1} \rangle + \langle \tilde{x}^{k+1} - x^{k+1}, x^\star - x^{k+1} \rangle_{\Theta^k + \rho M^T M} \\
&\quad + \langle \nabla f_1(\tilde{x}^{k+1}) - \nabla f_1(x^{k+1}), \tilde{x}^{k+1} - x^{k+1} \rangle + \langle \tilde{x}^{k+1} - x^{k+1}, \tilde{x}^{k+1} - x^{k+1} \rangle_{\Theta^k + \rho M^T M} \\
&\quad + \langle \nabla S_x^k(x^{k+1}), x^{k+1} - \tilde{x}^{k+1} \rangle.
\end{aligned}
$$

Here the last inequality uses $\nabla S_x^k(x) - \nabla S_x^k(y) = \nabla f_1(x) - \nabla f_1(y) + (\Theta^k + \rho M^T M)(x - y)$, and that $x^{k+1}$ is the exact solution of the $x$-subproblem,

Let $R$ be the radius in Proposition 3, and $K_x = \sup_k \|\nabla S_x(x^k)\|$. Then Cauchy-Schwarz combined with $f_1$ having a Lipschitz continuous gradient yields

$$
\begin{aligned}
&\langle \nabla f_1(\tilde{x}^{k+1}) - \nabla f_1(x^{k+1}), x^\star - \tilde{x}^{k+1} \rangle + \langle \tilde{x}^{k+1} - x^{k+1}, x^\star - x^{k+1} \rangle_{\Theta^k + \rho M^T M} \\
&+ \langle \nabla f_1(\tilde{x}^{k+1}) - \nabla f_1(x^{k+1}), \tilde{x}^{k+1} - x^{k+1} \rangle + \langle \tilde{x}^{k+1} - x^{k+1}, \tilde{x}^{k+1} - x^{k+1} \rangle_{\Theta^k + \rho M^T M} + \langle \nabla S_x^k(x^{k+1}), x^{k+1} - \tilde{x}^{k+1} \rangle \\
&\geq -L_{f_1} R \varepsilon_x^k - (\theta_{\max} + \rho \lambda_1(M^T M)) R \epsilon_x^k - L_{f_1} \epsilon_0^k \varepsilon_x^k - L_{f_1} R \varepsilon_x^k - (\theta_{\max} + \rho \lambda_1(M^T M)) \varepsilon_x^0 \varepsilon_x^k - K_x \varepsilon_x^k.
\end{aligned}
$$

Defining

$$
C_x = \max\{L_{f_1} R, (\theta_{\max} + \rho \lambda_1(M^T M)) R, L_{f_1} \varepsilon_x^0, (\theta_{\max} + \rho \lambda_1(M^T M)) \epsilon_x^0 + \sup_k \|\nabla S_x(x^k)\|\},
$$

we conclude that

$$
\langle \nabla S_x^k(\tilde{x}^{k+1}), x^\star - \tilde{x}^{k+1} \rangle \geq -C_x \varepsilon_x^k.
$$

The desired claim now follows by plugging $\tilde{x}^{k+1}$ into the expression for $\nabla S_x^k(x)$ in the preceding inequality.

$\square$

**Proof of Lemma 4**

*Proof.* By hypothesis, $\tilde{z}^{k+1}$ is an $\varepsilon_z^k$-approximate minimizer of the $z$-subproblem, so Lemma 1 shows there exists a vector $\tilde{s}$ with $\|\tilde{s}\|_{\Psi^k + \rho N^T N} \leq \sqrt{\varepsilon_z^k}$, such that

$$
g(z^\star) - g(\tilde{z}^{k+1}) \geq \rho \langle -N^T \tilde{u}^{k+1} + \Psi^k(\tilde{z}^k - \tilde{z}^{k+1}) + \tilde{s}, z^\star - \tilde{z}^{k+1} \rangle - \varepsilon_z^k.
$$

Rearranging, using Cauchy-Schwarz, along with the boundedness of the iterates, the preceding display becomes

$$
\begin{aligned}
g(z^\star) - g(\tilde{z}^{k+1}) - \rho \langle N^T \tilde{u}^{k+1} + \Psi^k(\tilde{z}^k - \tilde{z}^{k+1}), z^\star - \tilde{z}^{k+1} \rangle &\geq -\|\tilde{s}\|\|z^{k+1} - z^\star\| - \varepsilon_z^k \\
&\geq -\frac{1}{\sqrt{\psi_{\min} + \rho \lambda_{\min}(N^T N)}} \|\tilde{s}\|_{\Psi^k + \rho N^T N} R - \varepsilon_z^k \geq -\frac{R}{\sqrt{\psi_{\min} + \rho \lambda_{\min}(N^T N)}} \sqrt{\varepsilon_z^k} - \varepsilon_z^k \\
&\geq -\left( \frac{R}{\sqrt{\psi_{\min} + \rho \lambda_{\min}(N^T N)}} + K_{\varepsilon_z^k} \right) \sqrt{\varepsilon_z^k} = -C_z \sqrt{\varepsilon_z^k}.
\end{aligned}
$$

where the last display uses $\varepsilon_z^k \leq K_{\varepsilon_z} \sqrt{\varepsilon_x^k}$ and we have defined $C_z = \frac{R}{\sqrt{\psi_{\min} + \rho \lambda_{\min}(N^T N)}} + K_{\varepsilon_z^k}$. $\square$

## B.3 Proofs for Section 6

**Proof of Lemma 6**

*Proof.* 1. We begin by observing that strong convexity of $f$ implies

$$\langle \nabla f(\tilde{x}^{k+1}) - \nabla f(x^\star), \tilde{x}^{k+1} - x^\star \rangle \geq \sigma_f \|\tilde{x}^{k+1} - x^\star\|^2.$$

Decomposing $f$ as $f = f_1 + f_2$, the preceding display may be rewritten as

$$\langle \nabla f_1(\tilde{x}^{k+1}) + \nabla f_2(\tilde{x}^k) - \nabla f(x^\star), \tilde{x}^{k+1} - x^\star \rangle + \langle \nabla f_2(\tilde{x}^{k+1}) - \nabla f_2(\tilde{x}^k), \tilde{x}^{k+1} - x^\star \rangle$$
$$\geq \sigma_f \|\tilde{x}^{k+1} - x^\star\|^2.$$

Now, the exact solution $x^{k+1}$ of the $x$-subproblem at iteration $k$ satisfies $\nabla S_x^k(x^{k+1}) = 0$. Moreover, $\tilde{x}^{k+1}$ is an $\varepsilon_x^k$ minimizer of the $x$-subproblem, and $\nabla S_x^k(x)$ is Lipschitz continuous. Combining these three properties, it follows there exists a vector $\text{Err}_\nabla^k$ satisfying:

$$\nabla S_x^k(\tilde{x}^{k+1}) = \nabla S_x^k(x^{k+1}) + \text{Err}_\nabla^k = \text{Err}_\nabla^k, \quad \text{where } \|\text{Err}_\nabla^k\| \leq C\varepsilon_x^k.$$

Consequently, we have

$$\nabla f_1(\tilde{x}^{k+1}) + \nabla f_2(\tilde{x}^k) = \Theta^k(\tilde{x}^k - \tilde{x}^{k+1}) + \rho M^T N(\tilde{z}^{k+1} - \tilde{z}^k) - \rho M^T \tilde{u}^{k+1} + \text{Err}_\nabla^k.$$

Utilizing the preceding relation, along with the fact that the iterates are bounded, we reach

$$\langle \tilde{x}^k - \tilde{x}^{k+1}, \tilde{x}^{k+1} - x^\star \rangle_{\Theta^k} + \rho\langle N(\tilde{z}^{k+1} - \tilde{z}^k) + u^\star - \tilde{u}^{k+1}, M(\tilde{x}^{k+1} - x^\star) \rangle + C\varepsilon_x^k$$
$$\geq \sigma_f \|\tilde{x}^{k+1} - x^\star\|^2 - \langle \nabla f_2(\tilde{x}^{k+1}) - \nabla f_2(\tilde{x}^k), \tilde{x}^{k+1} - x^\star \rangle. \quad \square$$

2. As $-\rho N^T u^\star \in \partial g(z^\star)$, and $-\rho N^T \tilde{u}^{k+1} + \Psi^k(\tilde{z}^k - \tilde{z}^{k+1}) + \tilde{s} \in \partial_{\varepsilon_z^k} g(\tilde{z}^{k+1})$, we have

$$g(\tilde{z}^{k+1}) - g(z^\star) \geq \langle -\rho N^T u^\star, \tilde{z}^{k+1} - z^\star \rangle,$$
$$g(z^\star) - g(\tilde{z}^{k+1}) \geq \langle -\rho N^T \tilde{u}^{k+1} + \Psi^k(\tilde{z}^k - \tilde{z}^{k+1}), z^\star - \tilde{z}^{k+1} \rangle - \varepsilon_z^k.$$

So, adding together the two inequalities and rearranging, we reach

$$\langle \tilde{z}^{k+1} - z^\star, \rho N^T(u^\star - \tilde{u}^{k+1}) + \Psi^k(\tilde{z}^k - \tilde{z}^{k+1}) + \tilde{s} \rangle \geq -\varepsilon_z^k.$$

Now using boundedness of the iterates and $\|\tilde{s}\| \leq C\sqrt{\varepsilon_z^k}$, Cauchy-Schwarz yields

$$\langle \tilde{z}^{k+1} - z^\star, \rho N^T(\tilde{u}^{k+1} - u^\star) + \Psi^k(\tilde{z}^k - \tilde{z}^{k+1}) \rangle \geq -C\sqrt{\varepsilon_z^k}.$$

$\square$

**Proof of Lemma 8**  We wish to show for some $\delta > 0$, that the following inequality holds

$$\|w^k - w^\star\|_{G^k}^2 - \|w^{k+1} - w^\star\|_{G^k}^2 + \varepsilon_w^k \geq \delta\|w^{k+1} - w^\star\|_{G^k}^2.$$

To establish this result, it suffices to show the inequality

$$\|w^{k+1} - w^k\|_{G^k}^2 + \frac{2\sigma_f}{\rho}\|x^{k+1} - x^\star\|^2 - \frac{2}{\rho}\langle \nabla f_2(x^{k+1}) - \nabla f_2(x^k), x^{k+1} - x^\star \rangle \geq \delta\|w^{k+1} - w_\star\|_{G^k}^2.$$

We accomplish this by upper bounding each term that appears in $\|w^{k+1} - w_\star\|_{G^k}^2$, by terms that appear in the left-hand side of the preceding equality. To that end, we have the following result.

**Lemma 12** (Coupling Lemma). *Under the assumptions of Theorem 2, the following statements hold:*

1. *Let $\mu_1 \geq 2$, $c_1 = \frac{12L_f^2}{\rho^2 \lambda_{\min}(MM^T)}, c_2 = \frac{4\left(2L_f^2 + \theta_{\max}^2\right)}{\rho^2 \lambda_{\min}(MM^T)}, c_3 = \frac{8\|M\|^2}{\lambda_{\min}(MM^T)}$. Then for some constant $C \geq 0$, we have*

$$\|u^{k+1} - u^\star\|^2 \leq c_1 \|\tilde{x}^{k+1} - x^\star\|^2 + c_2 \|\tilde{x}^k - \tilde{x}^{k+1}\|^2 + c_3 \|\tilde{z}^k - \tilde{z}^{k+1}\|_{1/\rho \Psi^k + N^T N}^2 + C\varepsilon_w^k.$$

2. *There exists constants $c_4 = 2\left(\frac{\|N\|^2 + \psi_{\max}/\rho}{\sigma_{min}^2(N)}\right)\|M\|^2$, $c_5 = c_4/\|M\|^2$, such that*

$$\|z^{k+1} - z^\star\|_{1/\rho \Psi^k + N^T N}^2 \leq c_4 \|x^{k+1} - x^\star\|^2 + c_5 \|u^{k+1} - u^k\|^2.$$

3. *For all $\mu > 0$, we have*

$$\langle \nabla f_2(x^{k+1}) - \nabla f_2(x^k), x^{k+1} - x^\star \rangle \leq \mu/2\|x^{k+1} - x^\star\|^2 + \frac{L_f^2}{2\mu}\|x^{k+1} - x^k\|^2.$$

Taking Lemma 12 as given, let us prove Lemma 8.

*Proof.* Observe Lemma 12 implies

$$\frac{1}{2}\|w^{k+1} - w^k\|_{G^k}^2 + \frac{2\sigma_f}{\rho}\|x^{k+1} - x^\star\|^2 - \frac{2}{\rho}\langle \nabla f_2(x^{k+1}) - \nabla f_2(x^k), x^{k+1} - x^\star \rangle - \delta\|w^{k+1} - w_\star\|_{G_k}^2$$

$$\geq \left(\frac{2\sigma_f - \mu_2}{\rho} - \delta(c_1 + c_4)\right)\|\tilde{x}^{k+1} - x^\star\|^2 + \left(\frac{\theta_{\min}}{2\rho} - \frac{L_f^2}{2\rho\mu_2} - \delta c_2\right)\|\tilde{x}^k - \tilde{x}^{k+1}\|^2$$

$$+ (1/2 - \delta c_3)\|\tilde{z}^k - \tilde{z}^{k+1}\|_{1/\rho \Psi^k + N^T N}^2 + (1/2 - \delta c_5)\|\tilde{u}^k - \tilde{u}^{k+1}\|^2 - C\varepsilon_w^k.$$

Setting $\mu_2 = \sigma_f$, and using $\theta_{\min} > L_f^2/\sigma_f = \kappa_f L_f$, we find by setting

$$\delta = \min\left\{\frac{\sigma_f}{\rho(c_1 + c_4)}, \frac{\theta_{\min} - \kappa_f L_f}{2\rho c_2}, \frac{1}{2c_3}, \frac{1}{2c_5}\right\} > 0,$$

that

$$\|\tilde{w}^k - w^\star\|_{G^k}^2 - \|\tilde{w}^{k+1} - w^\star\|_{G^k}^2 + C\varepsilon_w^k \geq \delta\|\tilde{w}^{k+1} - w^\star\|_{G^k}^2,$$

as desired. $\square$

We now turn to to the proof of Lemma 12.

**Proof of Lemma 12**

*Proof.*   1. Observe by $L_f$-smoothness of $f$, that

$$2L_f^2\left(\|\tilde{x}^{k+1} - x^\star\|^2 + \|\tilde{x}^k - x^\star\|^2\right) \geq \|\nabla f_1(\tilde{x}^{k+1}) - \nabla f_1(x^\star) + \nabla f_2(\tilde{x}^k) - \nabla f_2(x^\star)\|^2$$

$$= \|\Theta^k(\tilde{x}^k - \tilde{x}^{k+1}) + \rho M^T N(\tilde{z}^k - \tilde{z}^{k+1}) + \rho M^T(u^\star - \tilde{u}^{k+1}) + \mathrm{Err}_\nabla^k\|^2$$

$$\geq \frac{1}{2}\rho^2\|M^T(\tilde{u}^{k+1} - u^\star)\|^2 - \|\Theta^k(\tilde{x}^k - \tilde{x}^{k+1}) + \rho M^T N(\tilde{z}^k - \tilde{z}^{k+1}) + \mathrm{Err}_\nabla^k\|^2.$$

Here the last inequality uses with $\mu = 2$, the identity (valid for all $\mu > 1$)

$$\|a + b\|^2 \geq (1 - \mu^{-1})\|a\|^2 + (1 - \mu)\|b\|^2.$$

So, rearranging and using $\|a + b\|^2 \leq 2\|a\|^2 + 2\|b\|^2$, we reach

$$\frac{\rho^2}{2}\|M^T(\tilde{u}^{k+1} - u^\star)\|^2 \leq 6L_f^2\|\tilde{x}^{k+1} - x^\star\|^2 + 2\left(2L_f^2 + \|\Theta^k\|^2\right)\|\tilde{x}^k - \tilde{x}^{k+1}\|^2$$

$$+ 4\rho^2\|M\|^2\|N(\tilde{z}^k - \tilde{z}^{k+1})\|^2 + 4\|\mathrm{Err}_\nabla^k\|^2.$$

Now, using $\|\mathrm{Err}_\nabla^k\|^2 \le C\varepsilon_x^k \le C\varepsilon_w^k$ and $\|v\|_{N^T N} \le \|v\|_{1/\rho\Psi^k + N^T N}$, we reach

$$\|\tilde{u}^{k+1} - u^\star\|^2 \le \frac{12L_f^2}{\rho^2 \lambda_{\min}(MM^T)}\|\tilde{x}^{k+1} - x^\star\|^2 + \frac{4\left(2L_f^2 + \theta_{\max}^2\right)}{\rho^2 \lambda_{\min}(MM^T)}\|\tilde{x}^k - \tilde{x}^{k+1}\|^2$$

$$\frac{8\|M\|^2}{\lambda_{\min}(MM^T)}\|\tilde{z}^k - \tilde{z}^{k+1}\|_{1/\rho\Psi^k + N^T N}^2 + C\varepsilon_w^k. \ \square$$

2. This inequality is a straightforward consequence of the relation

$$M(\tilde{x}^{k+1} - x^\star) = \tilde{u}^{k+1} - \tilde{u}^k + N(z^\star - \tilde{z}^{k+1}).$$

Indeed, using the identity $\|a + b\|^2 \le 2\|a\|^2 + 2\|b\|^2$, we reach

$$\|N(\tilde{z}^{k+1} - z^\star)\|^2 \le 2\|M\|^2\|\tilde{x}^{k+1} - x^\star\| + 2\|\tilde{u}^k - \tilde{u}^{k+1}\|^2.$$

Consequently

$$\|\tilde{z}^{k+1} - z^\star\|_{1/\rho\Psi^k + N^T N}^2 \le 2\left(\frac{\|N\|^2 + \psi_{\max}/\rho}{\sigma_{\min}^2(N)}\right)\|M\|^2\|\tilde{x}^{k+1} - x^\star\|$$

$$+ 2\left(\frac{\|N\|^2 + \psi_{\max}/\rho}{\sigma_{\min}^2(N)}\right)\|\tilde{u}^k - \tilde{u}^{k+1}\|^2,$$

which is precisely the desired claim. $\square$

3. Young's inequality implies for all $\mu > 0$, that

$$\langle \nabla f_2(\tilde{x}^{k+1}) - \nabla f_2(\tilde{x}^k), \tilde{x}^{k+1} - x^\star \rangle \le \frac{1}{2\mu}\|\nabla f_2(\tilde{x}^{k+1}) - \nabla f_2(\tilde{x}^k)\|^2 + \frac{\mu}{2}\|\tilde{x}^{k+1} - x^\star\|^2.$$

The desired claim now follows from $L_f$-smoothness of $f_2$. $\square$

## C  Proofs for Section 7

### C.1  Proof of Lemma 9

*Proof.*    1. By $\hat{L}_f$ relative smoothness we have,

$$f(x) \le f(\tilde{x}^k) + \langle \nabla f(\tilde{x}^k), x - \tilde{x}^k \rangle + \frac{\hat{L}_f}{2}\|x - \tilde{x}^k\|_{H_f(\tilde{x}^k)}^2.$$

$$\begin{aligned}
\|x - \tilde{x}^k\|_{H_f(\tilde{x}^k)}^2 &= \|x - \tilde{x}^k\|_{\hat{H}^k}^2 + (x - \tilde{x}^k)^T(H_f(\tilde{x}^k) - \hat{H}^k)(x - \tilde{x}^k) \\
&\le \|x - \tilde{x}^k\|_{\hat{H}^k}^2 + \|E^k\|\|x - \tilde{x}^k\|^2 \le \|x - \tilde{x}^k\|_{\hat{H}^k}^2 + \gamma^k\|x - \tilde{x}^k\|^2 \\
&= \|x - \tilde{x}^k\|_{\Theta^k}^2.
\end{aligned}$$

The claimed inequality immediately follows.

2. By our hypothesis on $H_f^k$ and $\hat{H}^k$, we have

$$\|x - \tilde{x}^k\|_{H_f(\tilde{x}^k)}^2 \le \|x - \tilde{x}^k\|_{H_f(\tilde{x}^k) + \gamma I} \le \frac{1}{1-\tau}\|x - \tilde{x}^k\|_{\hat{H}^k + \gamma^k I}^2 = \frac{1}{1-\tau}\|x - \tilde{x}^k\|_{\Theta^k}^2.$$

The desired inequality now follows immediately from the last display.

$\square$

