# OpenReview forum: "On the (linear) convergence of Generalized Newton Inexact ADMM"
_TMLR — Accepted by TMLR_

### Review · Reviewer_N9oM · 2025-10-13

**Summary Of Contributions:**

The paper provided comprehensive convergence analysis for the inexact ADMM for solving large-scale composite convex optimization problems. The papers proposed GENI-ADMM that solves second-order approximation of the associated subproblems, which covers many existing algorithmic framework. Overall, the paper is well-written, and the contributions are significant.

**Additional Comments:**

NA

**Audience:**

Yes

**Audience Explanation:**

Large-scale composite convex optimization represents one of the most fundamental models in the literature and ADMM stands the most widely used first-order methods for solving such problems with numerous real-world applications. How to develop efficient and scalable ADMM-type algorithms is of significant interest.

**Claims And Evidence:**

Yes

**Claims Explanation:**

1. The proofs conducted in the paper are overall correct, to the best of my knowledge.
2. The paper provided numerical experiments to support the theoretical findings.
3. The paper is well-organized.

**Requested Changes:**

1. Can you clarify the motivation of using Newton-type methods for solving ADMM subproblem instead of apply them to the augmented Lagrangian subproblem directly. The ALM shares faster convergence properties and if one can afford to apply second-order-type methods, what is the reason of using ADMM?
2. Can you explain how the inexactness conditions are implementable, i.e., how to verify them in practical computation?
3. Can you also comment on the convergence Issus of ADMM without ensuring the solvability of the subproblems? It is expected that when your algorithm reduced to the classical ADMM, some additional assumptions are needed. A related paper is provided here:
    - Chen et al. A note on the convergence of ADMM for linearly constrained convex optimization problems.

---

> ### Author Response · Authors · 2025-11-04
> **Response to reviewer N9oM**
>
> We thank the reviewer you for their positive assessment and constructive feedback. We have carefully considered your comments and questions in our revision.
>
> **Requested Changes:**
> 1. > "Can you clarify the motivation of using Newton-type methods for solving ADMM subproblem instead of apply them to the augmented Lagrangian subproblem directly. The ALM shares faster convergence properties and if one can afford to apply second-order-type methods, what is the reason of using ADMM?
>
>  The purpose of GeNI-ADMM is to develop a useful convergence framework for ADMM schemes that use approximations to make the ADMM subproblems \emph{easier to solve}.
>
> We want to clarify a potential point of confusion: GeNI-ADMM does not employ Newton-type methods for solving the subproblems. The user can solve the subproblems with whatever method they wish. The ``Newton'' in GeNI-ADMM (see the discussion in the second half of page 4, we forgot to mention $N = - I$, and will fix this in the revision.) refers to the fact that it covers approximate ADMM schemes which replace $f(x)$ at each iteration with a preconditioned linearization:
>
> $f(x) \approx f(\tilde{x}^k) + \langle \nabla f(\tilde{x}^k), x-\tilde{x}^k\rangle + \frac{L_\Theta\|x-\tilde{x}^k\|^2_{\Theta^k}}{2}.$
>
> This approximation of $f$ leads to a linear system solve for the $x$-subproblem, where $\Theta^k$ is generally selected so that the resulting system can be solved exactly at low cost or rapidly by preconditioned Conjugate Gradient.
> For instance, NysADMM from Zhao et al. (2022) selects $\Theta^k$ to be the Hessian at $\tilde{x}^k$, exploiting the fact that data matrices in machine learning problems have approximate low-rank structure, which enables fast solution via conjugate gradient with a randomized preconditioner.
> Thus, this approach is quite different from applying Newton's method to the $x$-subproblem.
> Indeed, the core purpose of GeNI-ADMM is to avoid the need for expensive iterative methods when solving the subproblems.
>
> The standard augmented Lagrangian method (ALM) does not accomplish our  goal of cheaper iterations, as it does not decouple the non-smooth term $g$ from the smooth term $f$.
> At each iteration, ALM still needs to solve an expensive non-smooth optimization problem that is not much easier than the original problem.
> To handle non-smoothness, one would need to apply semi-smooth augmented Lagrangian (SSNAL) schemes such as the method in Li et al. (2018).
> However, the semismooth-Newton approach is orthogonal to the spirit of GeNI-ADMM, as it involves dealing with a much more complicated subproblem at each iteration.
> The SSNAL approach also often needs to be tailored to the specific problem class at hand; for example, the scheme in Li et al. (2018a) is specific to the lasso.
>
> GeNI-ADMM was motivated by efficient approximate ADMM schemes for machine learning problems like NysADMM, which lacked rigorous convergence guarantees. For such problems, high-accuracy solutions do little to improve generalization error, which is the ultimate quantity of interest in machine learning. Thus, methods achieving high accuracy may yield little practical benefit for these problems. Finally, even in settings where SSNAL methods are used for high-accuracy solutions, the SSNAL procedure is often warm-started with a solution from a first-order algorithm like ADMM. For instance, the QSDPNAL solver (Li et al. (2018b)) for SDPs first computes a reasonably good solution with an ADMM scheme, which it then uses to initialize the SSNAL solver to obtain a high-accuracy solution.
>
> **References**
>
> Xudong Li, Defeng Sun, and Kim-Chuan Toh. A highly efficient semismooth newton augmented
> lagrangian method for solving lasso problems. SIAM Journal on Optimization, 28(1):433–458,
> 2018a.
>
> Xudong Li, Defeng Sun, and Kim-Chuan Toh. Qsdpnal: A two-phase augmented lagrangian method
> for convex quadratic semidefinite programming. Mathematical Programming Computation, 10(4):
> 703–743, 2018b.

---

> ### Author Response · Authors · 2025-11-04
> **Response to reviewer N9oM contd.**
>
> 2. > "Can you explain how the inexactness conditions are implementable, i.e., how to verify them in practical computation?
>
> The $x$-subproblem in our framework is always a smooth strongly convex optimization problem. Therefore, subproblem solvers can use standard stopping criteria (e.g. monitoring the norm of the gradient) that satisfy our inexactness assumptions to determine whether the solution produced is of sufficient accuracy.
>
> However, the most practical schemes derived from GeNI-ADMM either reduce the $x$-subproblem to solving a symmetric positive definite linear system (e.g., NysADMM) or result in a subproblem with a closed-form solution (e.g., Sketch-and-Solve ADMM). In the latter case, verifying $x$-subproblem inexactness is trivial.
> In the former case, standard Conjugate Gradient stopping criteria such as monitoring the residual norm with a tolerance that decays to match our inexactness assumptions can be used.
>
> Verifying that the $z$-subproblem has been solved to sufficient accuracy is more challenging.
> In the most general case, a subproblem solver can use the norm of the subgradient as a stopping criterion.
> However, in most practical machine learning applications (the motivating setting for this work), the $z$-subproblem reduces to a proximal operator or projection with a simple closed-form solution.
>
> Even when the proximal operator or projection is expensive to compute (e.g., projection onto the PSD cone), the subproblem is often structured.
> Thus, solvers that exploit this structure and come equipped with natural stopping criteria for determining termination can be employed.
> In the case of the PSD cone, the projection can be solved approximately via Krylov methods like LOBPCG, which possess stopping criteria for determining termination (Rontsis et al. (2022)).
> Similar statements hold for other proximal operators used in machine learning that lack closed-form solutions (Schmidt et al. (2011)).
> Our treatment of $z$-subproblem inexactness was motivated by these practical concerns.
>
> In the revision, we have an included more discussion under Assumption on implementing the inexactness conditions in practice.
>
> **References**
>
> Nikitas Rontsis, Paul Goulart, and Yuji Nakatsukasa. Efficient semidefinite programming with
> approximate admm. Journal of Optimization Theory and Applications, 192(1):292–320, 2022
>
> Mark Schmidt, Nicolas Roux, and Francis Bach. Convergence rates of inexact proximal-gradient
> methods for convex optimization. Advances in Neural Information Processing Systems, 24, 2011

---

> ### Author Response · Authors · 2025-11-04
> **Response to reviewer N9oM contd.**
>
> 3. > "Can you also comment on the convergence Issus of ADMM without ensuring the solvability of the subproblems? It is expected that when your algorithm reduced to the classical ADMM, some additional assumptions are needed. A related paper is provided here:
> Chen et al. A note on the convergence of ADMM for linearly constrained convex optimization problems."
>
> Assumption 4 on the $\Theta^k$ and $\Psi^k$ sequences requires these matrices to be positive definite for all $k$. Therefore, the subproblems always involve minimizing lower semicontinuous strongly convex functions over closed convex sets, ensuring that both subproblems are always solvable.
>
> Vanilla ADMM is only recovered when $f_1 = f$, $f_2 = 0$, and $\Theta^k, \Psi^k = 0$ for all $k$, which would violate Assumption 4. In this case, one would either need to add the assumption that the subproblems are both solvable or adopt the regularity assumptions proposed in Chen et al.
>
> We have added a discussion of the importance of this assumption for convergence and the result of Chen et al. after Assumption 4.

---

> > ### Comment · Reviewer_N9oM · 2025-11-25
> > **Reponse to the revision**
> >
> > Thank you for providing the detailed revision. My previous concerns are addressed suitably and I don't have other comments.

---

> > > ### Author Response · Authors · 2025-12-01
> > >
> > > We are please to hear we addressed your concerns. We thank the reviewer again for their comments which helped to improve the manuscript.

---

### Review · Reviewer_NGmV · 2025-10-26

**Summary Of Contributions:**

This paper revisits ADMM for minimizing a convex, separable function $f(x)+g(z)$, $x\in\mathcal{X}$, $z\in\mathcal{Z}$ subject to coupled linear constraints $Mx + Nz=0$. Since many ADMM variants have been proposed in the literature, the authors propose a unified framework, called GeNI-ADMM, that inlcludes them as special cases. This ADMM framework follows the classical Gauss–Seidel way of updating $x$ and $z$ sequentially.

Specifically, at each iteration, the GeNI-ADMM assumes that $f$ has a separable structure, i.e., $f=f_1+f_2$, and forumuate the $x$-subproblem as
\begin{equation*}
\min_{x\in X}\ f_1(x) + f_2(\tilde{x}^k) + \nabla f_2(\tilde{x}^k)^T(x - \tilde{x}^k) + \frac{1}{2}\|\|x - \tilde{x}^k\|\|_{\Theta^k}^2 + \frac{\rho}{2}\|\|Mx + N\tilde{z}^k + \tilde{u}^k\|\|^2,
\end{equation*}
where $f_1$ is kept and $f_2$ is approximately locally with a quadtratic model. This scheme unifies the function approximation appraoch and AL linearization approach.

Next, this method solves a $z$-subproblem
\begin{equation*}
\min_{z\in Z}\ g(z) + \frac{\rho}{2}\|\|M\tilde{x}^{k+1} + Nz + \tilde{u}^k\|\|^2 + \frac{1}{2}\|\|z - \tilde{z}^k\|\|_{\Psi^k}^2,
\end{equation*}
where $\Psi^k$ helps linearize the AL quadratic term when $N$ is a complicated matrix.

The (scaled) Lagrangian multiplier is updated
\begin{equation*}
\tilde{u}^{k+1} = \tilde{u}^k + M\tilde{x}^{k+1} + N\tilde{z}^{k+1}.
\end{equation*}

A practical advantage of this method is it allows inexact solutions to the $x$- and $z$-subproblems. Theoretically, sublinear and linear convergence for convex and strongly convex problems are established, respectively. Its applications to specific ADMM variants are discussed. Finally, numerical experiments are conducted to validate the theoretical results

**Audience:**

Yes

**Audience Explanation:**

This paper provides a complete analysis of a practical ADMM framework for convex and strongly convex optimization problems. It allows flexible strategies for implementation and inexact solutions to the subproblems. I think the audience in the field of ADMM may be interested in these results.

**Claims And Evidence:**

Yes

**Claims Explanation:**

This paper develops a general ADMM framework that covers many existing ADMM variants. It fills the gap that some ADMM variants do not have theoretical guarantees. The paper’s fit within the existing ADMM literature is clearly discussed. The literature review and motivation sections are well written and easy to follow. Although the analysis is a bit technical, it somewhat lacks details supporting the derivations, which could be improved further.

**Requested Changes:**

Major Comments:
Please add more explanations to the analysis. For example on Page 30, first paragraph in the proof of Proposition 3. Please give more detailed explanations on how did you derive the first display. If one plugs in $w=w^\star$ $Mx^\star+Nz^*=0$ into Lemma 11, it follows that


\begin{equation*}
... + \rho \|\|z^{k+1} - Nz^\star\|\|^2 + ... \le ... +\rho  \|\|\tilde{z}^k - Nz^\star\|\|^2 + ...
\end{equation*}

But how you obtain $\|\|...\|\|_{\Psi^k + \rho N^TN}^2$? Also, please add plugging in $w=w^*$ into Lemma 11




Minor Comments:
1. Page 3, After Eq. (2). Should be $Y\subset\mathcal{H}$?
2. Page 4, After Eq. (6). Should be psd matrices that approximate the Hessian of $f_2$?
3. Page 3. After Eq. (1). If later in Section 5, the case $X=\mathcal{X}$ and $Z=\mathcal{Z}$ is considered, should $X\subset \mathcal{X}$ be replaced by $X\subseteq \mathcal{X}$? This question also applies to $y$ and $Z$.
4. Page 8, Definition 2. Do you mean $\hat{L}_{\Theta}$-relatively smooth with respect to bounded $\Theta$? It sounds a bit odd to repeat $\Theta$ twice here.
5. Page 12, Lemma 5. Delete a dot after Assumptions 1-4
6. Page 13, Second display in the proof of Lemma 5. $\Theta_k$ should be $\Theta^k$.
7. Page 13, Second display in the proof of Lemma 5. The $1$-relative smoothness of $f_2$ should give you $\nabla f_2(\tilde{x}^k)^T(\tilde{x}^{k+1} - \tilde{x}^k)$?
9. Page 13, Proof of Lemma 5. Delete the equation number (24), and also (25) if they are not used.
10. Page 14. Second display. Please adjust the space for the two identities. You may add "and" between them.

---

> ### Author Response · Authors · 2025-11-04
> **Response to reviewer NGmV**
>
> We thank the reviewer for their positive assessment, close reading of the paper and their suggestions for improving clarity. We have tried to take into account many of your comments in our revision.
>
> **Major Changes**
>
> > "Please add more explanations to the analysis. For example on Page 30, first paragraph in the proof of Proposition 3. Please give more detailed explanations on how did you derive the first display. If one plugs in $w=w^\star$ $Mx^\star+Nz^*=0$ into Lemma 11, it follows that
> > \begin{equation*} ... + \rho ||z^{k+1} - Nz^\star||^2 + ... \le ... +\rho ||\tilde{z}^k - Nz^\star||^2 + ... \end{equation*}
> >But how you obtain $||...||_{\Psi^k + \rho N^TN}^2$? Also, please add plugging in $w=w^*$ into Lemma 11"
>
> First, please let us know if after the revision, there are any parts you still find unclear, and we will clarify them for you.
>
> As to your specific questions, we first note that there was a typo in the claimed inequality in the statement of Lemma 11.
> The correct inequality is:
>
> \begin{align*}
> Q(w, w^{k+1}) & \le \frac{1}{2}\left(\|z-\tilde x^k\|^2_{\Theta^k}-\|x-x^{k+1}\|^2_{\Theta^k}\right)+\frac{1}{2}\left(\|z-\tilde z^k\|^2_{\Psi^k +\rho N^TN}-\|z-z^{k+1}\|^2_{\Psi^k + \rho N^TN}\right) \\ & + \frac{\rho}{2} \left(\|\tilde z^k - Mx\|^2 - \|z^{k+1} - Mx\|^2 \right)+\frac{\rho}{2}\left(\|\tilde u^k-u\|^2-\|u^{k+1}-u\|^2\right).
> \end{align*}
>
> Note, the $\rho N^{T}N$ that was missing in the definition of the norms in the second term, is now present.
> In the revision we now provide a more detailed argument for the proof of Proposition 3.
> The key observation is that the preceding inequality, is the result guaranteed by Lemma 5, only with no subproblem errors, as we are using the exact subproblem solutions.
> Lemma 5 is proved using the perturbed optimality conditions.
> When the subproblems are solved exactly, the exact optimality conditions are satisfied.
> Therefore to conclude Lemma 11, we can use the exact same argument used to establish Lemma 5, only this time using the exact optimality conditions instead of the perturbed ones.
>
> Lemma 11 holds for any $w$, not just $w_\star$, so there is no need to explicitly plug it in there. However, to make the proof of Proposition 3 clear, we now state the following in the beginning of the proof : ``To this end, plugging in $w = w^\star$ into the inequality of Lemma 11''.

---

> > ### Author Response · Authors · 2025-11-04
> > **Response to reviewer NGmV contd.**
> >
> > **Rely to Minor Comments**
> > 1. Yes, it should be $\mathcal H$. We have fixed this in the revision.
> > 2. Yes, it should be $f_2$ and we have fixed this in the revision.
> > 3. Yes, the $\subseteq$ is more appropriate as equality can be attained, as seen in Section 5. We have fixed this in the revision.
> > 4. Yes, this was a bit repetitive we have edited this to avoid mentioning $\Theta$ twice in the revision.
> > 5. Thanks for catching this. We have removed the dot in the revision.
> > 6. Yes, we have fixed this.
> > 7. Yes that's correct, the $f_1$ in the term $\langle \nabla f_1(\tilde x^k), \tilde x^{k+1} - \tilde x^k \rangle $ should be $f_2$.
> >     We have corrected this in the revision.
> > 8. We have removed the equation numbers in the revision.
> > 9. We have reformatted according to your suggestion in the revision.

---

### Review · Reviewer_14PJ · 2025-10-26

**Summary Of Contributions:**

This paper introduces **GeNI-ADMM (Generalized Newton Inexact ADMM)**, a unified framework for designing and analyzing a broad class of approximate Alternating Direction Method of Multipliers (ADMM) algorithms for large-scale composite convex optimization. The framework synthesizes three common approximation strategies: (1) **Function approximation**, where the smooth part of the objective is replaced by a first- or second-order model; (2) **Augmented Lagrangian linearization**, which adds proximal terms to simplify subproblems; and (3) **Inexact subproblem solves**, where solutions are found only up to a specified tolerance controlled by forcing sequences.

The primary theoretical contributions are two-fold:
1.  **Sublinear Convergence (Theorem 1)**: The paper proves that under standard convexity assumptions, GeNI-ADMM achieves an ergodic $\mathcal{O}(1/t)$ convergence rate for both suboptimality and feasibility gap, matching the rate of exact ADMM despite the approximations.
2.  **Linear Convergence (Theorem 2)**: Under the additional assumption of strong convexity, the framework guarantees a linear convergence rate via a Lyapunov analysis, though this requires an additional spectral condition, $\theta_{\min}>L_f^2/\sigma_f$, on the regularization matrices (Sec. 6.1, Thm. 2).

The paper applies this framework to provide the first formal convergence guarantees for ADMM variants that use randomized numerical linear algebra (RandNLA), recovering schemes like **NysADMM** and a newly proposed **sketch-and-solve ADMM** as special cases (Sec. 7). For the latter, the analysis reveals that a specific correction term, $\gamma_k I$, chosen to dominate the Hessian approximation error $\|E_k\|$, is necessary to ensure convergence (Lemma 9). This finding is validated by numerical experiments on lasso, $\ell_1$-logistic, and elastic-net regression, which also illustrate the predicted convergence rates and transitions from sublinear to linear phases (Sec. 8.3).

**Additional Comments:**

This is a strong, well-executed theoretical paper that introduces an elegant and valuable unifying framework for approximate ADMM. The analysis is rigorous and uses the framework to resolve important open questions about the convergence of practical randomized variants like NysADMM, leading to novel insights such as the necessity of a correction term in sketch-and-solve methods. While the work represents a clear advance in the field, some of the theoretical assumptions are strong; for instance, the condition for the linear rate appears stricter than what is observed in practice. The paper does an excellent job of positioning GeNI-ADMM relative to other ADMM variants, correctly identifying the benefits of using curvature. However, it lacks comparison to non-ADMM state-of-the-art solvers for the problems tested, which would provide a more complete picture of practical performance. Additionally, the rate analysis for sketch-and-solve ADMM, while valid, could be strengthened by connecting it more explicitly to standard RandNLA rates that depend on sketch size and problem dimensions. Despite these limitations, the theoretical results are convincing, and my recommendation for acceptance is contingent on the authors addressing the requested changes.

**Audience:**

Yes

**Audience Explanation:**

A unifying analysis that covers both proximal/linearized ADMM and randomized second-order variants is relevant to modern large-scale optimization, especially with clear guidance on inexactness and sketching. The empirical section is practical but would benefit from robustness reporting.

**Broader Impact Concerns:**

No specific concerns beyond standard optimization research. No identifiable dual-use risks are apparent.
This work is foundational research in the area of convex optimization. It proposes a theoretical framework and analyzes the convergence of numerical algorithms. As such, it does not raise immediate ethical concerns or have direct societal applications that would necessitate a broader impact statement. The potential applications are general-purpose, and any societal impact would be determined by the downstream use-cases of the optimization solvers.

**Claims And Evidence:**

Yes

**Claims Explanation:**

**Yes, with caveats.**
The convex $O(1/t)$ and strong-convexity linear-rate results are sound under the stated assumptions (Theorem 1/2). However, the added condition $\theta_{\min}>L_f^2/\sigma_f$ is not enforced in experiments and should be either justified or relaxed.

**Requested Changes:**

**Critical (acceptance-blocking)**
1. **Linear-rate condition:** Either (a) remove the claim that the $\theta_{\min}>L_f^2/\sigma_f$ condition is unnecessary, or (b) strengthen Theorem 2 to avoid it (e.g., add assumptions yielding the needed lower bound) and give the exact lemma replacing the current step in Lemma 7 (Sec. 6.2–6.3).
2. **Open-question claim:** Cite specific prior work and state precisely what is closed now (which algorithms/assumptions). Add references at the end of Sec. 7 or in the Conclusion.
3. **Clarify Experimental Details**: In Section 8, please specify precisely how the inexact subproblem solves were performed. State the inner-loop algorithm used (e.g., Conjugate Gradient for linear systems, L-BFGS for logistic regression) and, most importantly, the termination criteria or tolerance schedule that implements the forcing sequences $\{\epsilon_x^k\}$ and $\{\epsilon_z^k\}$. This is essential for reproducibility.

**Major (strongly recommended)**
3. **Sketch dependence:** Add bounds translating $\|E_k\|$ in Cor. 2 to sketch size and dimension; include guidance on choosing $\gamma_k$ with computational overhead. (Sec. 7.2)
4. **Experimental robustness:** Report multiple seeds, error bars, and sensitivity to $(\rho,\eta,\gamma_k)$ and sketch sizes (Sec.
5. **Explicit Constants**: In Appendix B, provide more explicit bounds for the constants $C_x$ and $C_z$ that appear in the proofs of Lemmas 3 and 4 (and thus Theorem 1). Even if complex, showing their dependence on Lipschitz constants, operator norms, and the bound on the iterate norms ($R$ from Prop. 3) would make the theoretical results more transparent.

**Minor**
6. Clarify when NysADMM’s $\sigma I$ can be set to 0 in practice and implications for convergence. (Sec. 7.1)
7. Quantify “linear convergence” in plots (add fitted rates/slopes). (Sec. 8.2)
8.  **Add Broader Baselines**: To better contextualize the practical performance, consider adding at least one strong, non-ADMM baseline to the experiments in Section 8 (e.g., an accelerated proximal gradient method or a state-of-the-art coordinate descent solver for LASSO).
9.  **Connect Sketching Rate to Literature**: In Section 7.2, add a brief remark connecting the rate in Corollary 2 to standard RandNLA rates that depend explicitly on sketch size. For instance, you could mention how bounds on $||E^k||$ can be obtained from matrix concentration inequalities.

**Typos & Style**
8. Fix “converge converge” in Sec. 8.1.

### Comparisons & Positioning
- **To proximal/generalized ADMM (Deng & Yin, 2016):** The framework subsumes these methods; however, for **linear rates**, please state explicitly whether GeNI-ADMM’s conditions are **weaker/stronger** than Deng–Yin’s (e.g., your $\theta_{\min}$ condition vs. fixed proximal terms).
- **Randomized second-order methods:** For NysADMM and sketch-and-solve, rates should expose **sketch accuracy parameters** ($\|E_k\|$, spectrum gaps) and **sketch size**; Corollary 2 contains $\|E_k\|$ inside constants but does not translate it to sample/embedding dimension. Please add standard bounds (e.g., Nyström error vs. leverage-score sample size) to quantify the iteration **and** time complexity trade-off.
- **“Open question resolved”**: Provide concrete prior references where global convergence was unknown/unstated, and clarify which **variants** and **assumptions** your result covers.

---

> ### Author Response · Authors · 2025-11-04
> **Response to reviewer 14PJ**
>
> We thank the reviewer for their positive assessment and the detailed comments, which have helped us to improve the manuscript.
>
> **Requested Changes Critical**
> 1. > "**Linear-rate condition**: Either (a) remove the claim that the $\theta_{\min}>L_f^2/\sigma_f$ condition is unnecessary, or (b) strengthen Theorem 2 to avoid it (e.g., add assumptions yielding the needed lower bound) and give the exact lemma replacing the current step in Lemma 7 (Sec. 6.2–6.3)."
>
> We have adjusted our language in the revision, and no longer use the term unnecessary.
> What we meant to convey was that empirically we found GeNI-ADMM converges linearly without enforcing this inequality.
> We have made this point clearer in the revision. We also explain more clearly how this condition arises from the use of linearization in the $x$ subproblem.
>
> 2. > "**Open-question claim:** Cite specific prior work and state precisely what is closed now (which algorithms/assumptions). Add references at the end of Sec. 7 or in the Conclusion.
>
> We have added more discussion about our results relative to prior work.
> For NysADMM in Section 7.1 we have a discussion in paragraph 2 of how are results compare to those of Zhao et al (2022).
> Specifically, Theorem 4.2 in Zhao et al. (2022) only establishes convergence for quadratic $f$ and does not provide an explicit convergence rate.
> The authors left convergence for more general $f$ open,
> only conjecturing that if $f$ is strongly convex, then they believe linear convergence is achievable.
> Our work closes this gap by showing NysADMM obtains the same ergodic $\mathcal{O}(1/t)$-rate as standard ADMM for smooth convex $f$.
>
> In regards to Sketch-and-Solve ADMM, we discuss in Section 7.3, how the survey Buluc et al. (2021) propose using sketching to speed up solution of the $x$-subproblem for quadratic objectives in ADMM for large-scale inverse problems.
> However, they provide no formal algorithm, analysis or experiments validating this proposal.
> Unlike, with Newton's method, to the best of our knowledge, there has been no analysis of ADMM using sketching based approximations for solving the $x$-subproblem.
>
> Showing that sketch-and-solve methods can be used in the $x$-subproblem for quadratic $f$ and beyond, was another of the motivations for this paper.
> Corollary 2, shows this is the case provided that the approximation is properly regularized, e.g. $\gamma^k>0$ is properly chosen.
>
> We have also added a summary of these points to the conclusion of the paper.
>
> 3. > "**Clarify Experimental Details:** In Section 8, please specify precisely how the inexact subproblem solves were performed. State the inner-loop algorithm used (e.g., Conjugate Gradient for linear systems, L-BFGS for logistic regression) and, most importantly, the termination criteria or tolerance schedule that implements the forcing sequences $\epsilon_x^k$ and $\epsilon_z^k$. This is essential for reproducibility."
>
> To ensure reproducibility, we have added a link to an anonymous github repo:
>
> https://anonymous.4open.science/r/GeNI_ADMM_TMLR-4E37/README.md
>
> which contains all the code used to generate our experimental results.
> In our revision we have provided the details of the subproblem solvers. As the $z$-subproblems in all examples considered have simple closed-form solutions, all algorithms evaluate the $z$-subproblem exactly. Inexactness is only used for the $x$-subproblem.
> NysADMM uses Nyström PCG from Frangella et al. (2023) as in the original NysADMM paper Zhao et al. (2022).
> At each iteration, the following absolute tolerance on the PCG residual is used for determining the termination of Nyström PCG:
>
> $\varepsilon^{k}_x =$  $min (\frac{\sqrt{r_p^{k-1}r_d^{k-1}}}{k^{1.5}},1)$
>
> where $r_p^{k-1}$ and $r_d^{k-1}$ are the ADMM primal and dual residuals at the previous iteration.
>
> For Logistic Regression, where exact ADMM uses L-BFGS to solve the $x$-subproblem, we use L-BFGS's default settings in Optim.jl.
> In particular, this means the $x$-subproblem is declared solved, when $\|g\|_{\infty} \leq 10^{-8}$, where $g$ is the gradient of the $x$-subproblem.

---

> ### Author Response · Authors · 2025-11-04
> **Response to reviewer 14PJ contd.**
>
> **Major (strongly recommended)**
>
> > "**Sketch dependence:** Add bounds translating $|E_k|$
> in Cor. 2 to sketch size and dimension; include guidance on choosing $\gamma_k$ with computational overhead. (Sec. 7.2)"
>
> We have significantly revised the discussion on the convergence of Sketch-and-solve ADMM.
> We now provide a brief overview of the Newton sketch and Nystr{\"o}m approximation techniques.
> Moreover, by appealing to existing results in the sketching literature, we discuss in Section 7.4 what value of $\gamma^k$ must be used for a given sketch size $s$ to ensure convergence.
> We then discuss in Corollary 2 how the value of $\gamma^k$, which is determined by approximation quality, affects the convergence rate.
> We also highlight the clear tradeoff between cheaper iterations with a slower convergence rate versus more expensive iterations with a faster convergence rate.
> Finally, we provide concrete recommendations for when Sketch-and-solve ADMM is expected to be most useful.
>
> > **Experimental robustness:** Report multiple seeds, error bars, and sensitivity to $(\rho, \eta, \gamma_k)$ and sketch sizes.
>
> We have found the algorithm to be very stable and robust, and have not extensively tuned these parameters.
> We note that $\gamma_k$ is selected exactly as spectral norm of the error, which is in line with the theory and a practical way of setting this parameter.
> Convergence is also guaranteed for any $\rho>0$, and there exist standard methods such as residual balancing for automatically selecting this parameter. Therefore we do not view ablating this parameter to be critical.
> Our repo is now available, so it easy for people to modify these parameters and see the impact if they wish.
>
> > **Explicit Constants:** In Appendix B, provide more explicit bounds for the constants and that appear in the proofs of Lemmas 3 and 4 (and thus Theorem 1). Even if complex, showing their dependence on Lipschitz constants, operator norms, and the bound on the iterate norms (from Prop. 3) would make the theoretical results more transparent.
>
> In the revision we have provided explicit upper bounds for the constants $C_x$ and $C_z$ appearing in Lemmas 3 and 4.
> We also provide an explicit upper bound on $C_\zeta$ in Theorem 1.
> Thus, it is now clear what $C_x$, $C_z$, and $C_\mathcal E$ depend on in the worst-case.

---

> > ### Author Response · Authors · 2025-11-04
> > **Response to reviewer 14PJ contd.**
> >
> > **Minor**
> >
> > > "Clarify when NysADMM’s $\sigma I$ can be set to 0 in practice and implications for convergence. (Sec. 7.1)
> > We have clarified this point in the discussion at the end of paragraph 3 in Section 7.1."
> >
> > > "Add Broader Baselines: To better contextualize the practical performance, consider adding at least one strong, non-ADMM baseline to the experiments in Section 8 (e.g., an accelerated proximal gradient method or a state-of-the-art coordinate descent solver for LASSO)."
> >
> > One of the main motivations of this work was the paper of \cite{pmlr-v162-zhao22a}, whose NysADMM outperformed solvers such as coordinate descent, accelerated proximal gradient with restarts, SAGA, A Semismooth-Newton Augmented Lagrangian method, and a specialized IPM method.
> > However, despite the strong empirical showing, the theory developed there only showed convergence for a quadratic objective. and did not provide a rate.
> > Therefore, we do not feel the need to compare to other methods, as the primary goal of this work is provide convergence analysis showing approximate schemes like NysADMM converge, which ensures practitioners can use them without concern.
> >
> > > "Fix “converge converge” in Sec. 8.1."
> > Thanks for catching this. We have fixed this in the revision.
> >
> > **Comparisons & Positioning**
> > > To proximal/generalized ADMM (Deng \& Yin, 2016): The framework subsumes these methods; however, for linear rates, please state explicitly whether GeNI-ADMM’s conditions are weaker/stronger than Deng–Yin’s (e.g., your $\theta_{\textup{min}}$ condition vs. fixed proximal terms).
> >
> > We have added a discussion about our conditions relative to those of Deng and Yin (2016).
> > Specifically, the $\theta_{\textrm{min}}$ condition in this paper is new.
> > The condition is not present in Deng and Yin (2016) as they do not linearize $f$.
> > If $f_1 = f$ and $f_2=0$, i.e. $f$ is not linearized then this condition is not needed as the term $\langle \nabla f_2(\tilde x^{k+1})-\nabla f_2(\tilde x^k),\tilde x^{k+1}-x^\star\rangle$ in Lemma 6 is identically $0$.
> > The need for an additional condition is perhaps unsurprising, since we allow for more approximation by allowing linearization of $f$.

---

### Author Response · Authors · 2025-11-04
**Response to all reviewers.**

We thank the reviewers for their positive reception to our work.
We greatly appreciate the insightful comments and detailed feedback, which have helped us to improve the quality of the submission.
We have uploaded a revised version of the manuscript implementing the various recommendations that have been made.

Detailed responses to each reviewer's comments and questions may be found below.

---

### Decision · Action_Editor_Rf32 · 2026-01-07

**Recommendation:** Accept with minor revision

**Additional Comments:**

For the minor revision, the following changes are recommended:

- You can consider improving the presentation of Theorem 1 statement. Right now it is a rather long paragraph and difficult to read. You can consider separating some of the equations such as the definitions of the averaged iterate as display equations to break up the text a little. You can also move the constants that are defined in the beginning of the statement to the outside of the statement.

- You can consider improving the navigation in the paper by using pointers to proofs of the results, for example for Theorem 1. Right now one needs to read the proof of Theorem 1 to see which lemmas are used and so on. You can include a proof sketch in the main text to show the readers which lemmas are used where and then a pointer to where the full proof is situated.

- For the open problems that the paper is solving, it is nice that the authors are very explicit. A minor suggestion is that they can provide a clearer pointer to where the open problem is mentioned in the existing work. For example for NysADMM, they may mention that this Zhao et al., 2022 states the open problem of showing convergence for NysADMM in the second paragraph after their Thm 4.3 statement, and so on.

- Right before Asp 4 missing paranthesis between "cone" and the reference "(Rontsis et al., 2022)".

**Audience:**

Yes

**Audience Explanation:**

Both the two-block problem template with linear constraints and the ADMM algorithm are widely used in ML, so the reviewing team agrees that the results are definitely of interest to the audience of TMLR.

**Claims And Evidence:**

Yes

**Claims Explanation:**

This work focuses on the two-block, linearly constrained composite convex optimization problem and provides a general ADMM framework for solving this problem. The new framework can handle first or second order approximation of the subproblem (after also incorporating preconditioning which may help to linearize the augmented term) and can obtain the standard rates of $O(1/t)$ under convexity and linear rate of convergence under strong convexity. An interesting aspect of the results is that they can handle ADMM variants using randomized linear algebra such as NysADMM or sketch-and-solve. The convergence of these methods, as explained in the work, are mentioned as open questions in the literature. This work also solves these open problems.

All the reviewers agree that the work fills an interesting gap in the literature and that the claims are supported by accurate, convincing and clear evidence. The positioning of the work and the presentation of the ideas are executed well.

---

> ### Author Response · Authors · 2026-01-20
> **Camera Ready Uploaded**
>
> Thank you for the nice suggestions! Here's how we have incorporated the major points into the camera ready submission:
> - We have simplified the presentation of Theorem 1 and improved cross-referencing to proofs in Section 5. We also now include a roadmap at the beginning of the section, which clearly delineates the role of each subsection, as this section is lengthy.
> - We now explicitly mention where the open problems occur in existing work, e.g. specifying where in Zhao et al. 2022 the discussion about leaving convergence of NysADMM to future work occurs.